# Riemannian Federated Learning via Averaging Gradient Streams

**Zhenwei Huang**[1], **Wen Huang**[1]*, **Pratik Jawanpuria**[2], **Bamdev Mishra**[3]
[1]Xiamen University, zwhhuang@stu.xmu.edu.cn, wen.huang@xmu.edu.cn
[2]Indian Institute of Technology Bombay, pratik.jawanpuria@iitb.ac.in
[3]Microsoft India, bamdevm@microsoft.com

## Abstract

Federated learning (FL) as a distributed learning paradigm has a significant advantage in addressing large-scale machine learning tasks. In the Euclidean setting, FL algorithms have been extensively studied with both theoretical and empirical success. However, there exist few works that investigate federated learning algorithms in the Riemannian setting. In particular, critical challenges such as partial participation and data heterogeneity among agents are not explored in the Riemannian federated setting. This paper presents and analyzes a Riemannian FL algorithm, called RFedAGS, based on a new efficient server aggregation—averaging gradient streams, which can simultaneously handle partial participation and data heterogeneity. We theoretically show that the proposed RFedAGS has global convergence and sublinear convergence rate under decaying step sizes cases; and converges sublinearly/linearly to a neighborhood of a stationary point/solution under fixed step sizes cases. These analyses are based on a vital and non-trivial assumption induced by partial participation, which is shown to hold with high probability. Extensive experiments conducted on synthetic and real-world data demonstrate the good performance of RFedAGS.

## 1 Introduction

Modern learning tasks handle massive amounts of data, which are geographically distributed across heterogeneous devices. Conventional centralized algorithms, e.g., stochastic gradient descent (SGD), need to collect the data into single device for training, which consumes significant storage and computing resource. Additionally, from the perspective of privacy security, transmitting raw training data may leak data privacy. A promising distributed learning paradigm—federated learning (FL)—allows a center server to coordinate with multiple agents (e.g., mobile phones and tablets) to train a desired model parameter without raw data sharing, which is an ideal solution to the issues aforementioned.

In recent years, with the development of Riemannian optimization, many machine learning problems have data structures that can be inscribed by low-dimensional smooth manifolds, and thus they can be modeled on manifolds. There are such examples including but not limited to principal component analysis (Ye & Zhang, 2021), Fréchet mean computation (Han et al., 2021; 2023a), hyperbolic structured prediction (Xiong et al., 2022), low-rank matrix/tensor learning (Jawanpuria & Mishra, 2018; Nimishakavi et al., 2018; Mishra et al., 2019), natural language processing (Jawanpuria et al., 2019a; 2020), domain adaptation (Han et al., 2022; 2024a; Jawanpuria et al., 2025), and neural network training (Magai, 2023; Han et al., 2023b). This motives us to develop a efficient Riemannian FL algorithm.

This paper focuses on the following Riemannian federated optimization problem

$$\arg\min_{x \in \mathcal{M}} F(x) := \frac{1}{N} \sum_{i=1}^{N} f_i(x), \text{ with } f_i(x) = \mathbb{E}_{\xi \sim \mathcal{D}_i}[f_i(x; \xi)], \tag{1.1}$$

where $\mathcal{M}$ is a $d$-dimensional Riemannian manifold, $N$ is the number of agents, $F : \mathcal{M} \to \mathbb{R}$ is the global objective, and $f_i : \mathcal{M} \to \mathbb{R}$ and $\mathcal{D}_i$ are local objectives and the data distribution held by agent

---

*Corresponding author: wen.huang@xmu.edu.cn.

$i, \forall i \in [N] = \{1, 2, \ldots, N\}$. Throughout this paper, we focus on the expected minimization (1.1), but the resulting conclusions are also true for the finite sum minimization in which the local objective is defined by $f_i(x) = \frac{1}{N_i} \sum_{j=1}^{N_i} f_i(x; z_{i,j})$ with $\mathcal{D}_i = \{z_{i,1}, z_{i,2}, \ldots, z_{i,N_i}\}$ the local dataset held by agent $i$. We may not necessarily assume that $\mathcal{D}_i, \forall i \in [N]$, are the independently identical distribution (I.I.D.), i.e., the data distributions across different agents are independent but not identical (non-I.I.D.).

A well-known Euclidean FL algorithm is Federated Averaging (FedAvg) (McMahan et al., 2017), which is adapted from the local stochastic gradient descent (local SGD) method. Specifically, at the beginning, FedAvg takes an initial guess $x_1$ as input and then sends it to all agents. Subsequently, the following steps are performed alternately:

(i) agent $j$ updates its the local parameter via performing $K$-step SGD with $x_t$ being the initial guess and generates the trained local parameter $x_{t,K}^j$ (this is called "local update" or "inner iteration"), and then the local parameter $x_{t,K}^j$ is uploaded to the server;

(ii) the server at random samples a subset of size $S$ from all agents, denoted by $\mathcal{S}_t$, and then averages the received local parameters to generate the next global parameter $x_{t+1}$, i.e.,

$$x_{t+1} \leftarrow \frac{1}{S} \sum_{j \in \mathcal{S}_t} x_{t,K}^j, \tag{1.2}$$

which is called "server aggregation", and then sends $x_{t+1}$ to all agents.

The two steps above constitute a round of communication (or outer iteration).

**Related works.** Early works primarily analyzed the convergence of FedAvg and its variants in limited settings, typically relying on one or both of the following assumptions: (i) full participation (i.e., $S = N$) and (ii) I.I.D. data distributions; see, e.g., (Zhou & Cong, 2018; Stich, 2019; Yu et al., 2019; Haddadpour et al., 2019; Wang & Joshi, 2021; Gu et al., 2023) and references therein. Subsequently, numerous works have studied the convergence of FL algorithms under (iii) partial participation and (iv) non-I.I.D. data assumption; see e.g., (Li et al., 2020b;a; Rizk et al., 2022) and references therein. In these works, partial participation is implemented by random sampling—the server randomly selects a subset of agents to perform local updates in each outer iteration.

Due to heterogeneity in the computational capabilities and the environment conditions across agents, their availability and response speeds are hardly predictable. This unpredictability makes random sampling-based approaches unsuitable for such scenarios. Recent works have instead adopted an arbitrary participation model, where agents may respond to the server in a stochastic and uncontrolled manner (Gu et al., 2021; Wang & Ji, 2022; Ribero et al., 2023; Xiang et al., 2023; Yan et al., 2023; Wang & Ji, 2024; Xiang et al., 2025; Ying et al., 2025). These works can be roughly divided into three categories: (i) **time-varying statistic**, i.e. agent $i$ participates in the $t$-th outer iteration with probability $p_t^i$ varying over time (Wang & Ji, 2022; Ribero et al., 2023; Xiang et al., 2023; Wang & Ji, 2024; Xiang et al., 2025); (ii) **time-invariant statistic**, i.e., the participation probability for agent $i$ is not varying over time (meaning $p_t^i = p_i$ for all $t \geq 1$) (Wang & Ji, 2024; Ying et al., 2025); and (iii) **periodic participation**, i.e., each agent $i$ must participate in at least one communication round within a fixed iteration interval (Gu et al., 2021; Yan et al., 2023).

The FL algorithms mentioned earlier operate solely in Euclidean space and thus cannot directly handle such problems whose parameters are located in manifolds due to the inherent curvature effects of manifolds. Only a limited number of studies have explored the design and analysis of FL algorithms on Riemannian manifolds. (Li & Ma, 2023) proposed a Riemannian counterpart of (1.2) and thus developed a Riemannian FL algorithm. Their algorithm involves in exponential mapping, its inverse, and parallel transport. Nevertheless, for some manifolds, e.g., the Stiefel manifold, the inverse of the exponential mapping and parallel transport have no closed forms, and only iterative methods can be used to compute them, which brings an extra computation burden. (Huang et al., 2024) adopted a framework similar to that of (Li & Ma, 2023) but integrate differential privacy to strengthen privacy guarantees. Under the non-I.I.D. setting, most convergence results in (Li & Ma, 2023; Huang et al., 2024) are established for the case $K = 1$ and full participation, i.e., all agents just perform one step local update (notably, for $K > 1$, the convergence analyses of both algorithms further assume that only one agent participates in communication). The algorithm proposed in (Zhang et al., 2024) supports general settings where $K > 1$ and $S > 1$, but its convergence analysis relies on

the full participation assumption. Additionally, the algorithm therein involves an orthogonal projector onto the manifold and requires that this projector is a singleton. Thus, its applicability is restricted to problems on compact Riemannian submanifolds embedded in Euclidean spaces. Subsequently, Wang et al. (Wang et al., 2025) proposed a zeroth-order gradient estimator and integrate it into RFedProj, resulting a zeroth-order Rimannian FL algorithm called ZO-RFedProj. The algorithms in (Xiao et al., 2024; 2025) incorporated the Barzilai-Borwein method into the framework of (Li & Ma, 2023). Despite the efforts of some, all of the Riemannian FL algorithms above have no theoretical guarantee under both partial participation and data heterogeneity setting. See Table 1 for comprehensive comparisons of existing Riemannian FL algorithms and the proposed RFedAGS. Table 2 summarizes the computational (communication) complexity required for these methods to complete one outer iteration. The table includes the local iteration complexity per agent (LICpA), server computational complexity (SCC), communication complexity (CC), and total computational complexity (TCC), where $\text{TCC} = \text{LICpA} + \text{SCC}$.

Table 1: Summary of existing algorithms and the proposed RFedAGS.

| Algorithms | Manifold | Partial Participation | Non-I.I.D. | Retraction | Vector transport |
|---|---|---|---|---|---|
| RFedSVRG (Li & Ma, 2023) | General [1] | ✗ [2] | Conditioned [3] | Exponential mapping | Parallel transport |
| RPriFed (Huang et al., 2024) | General [1] | ✗ | Conditioned [3] | Exponential mapping | Parallel transport |
| RFedProj (Zhang et al., 2024) | Compact submanifold | ✗ | ✔ | N/A | N/A |
| ZO-RFedProj (Wang et al., 2025) | Compact submanifold | ✗ | ✔ | N/A | N/A |
| RFedSVRG-2BBS (Xiao et al., 2024) | General [1] | ✗ [2] | Conditioned [3] | Exponential mapping | Parallel transport |
| RFedSVRG-BB (Xiao et al., 2025) | General [1] | ✗ [2] | Conditioned [3] | Exponential mapping | Parallel transport |
| **RFedAGS (this paper)** | **General** | **✔** | **✔** | **General retraction** | **Bounded** |

[1] Although these methods are suitable for general manifolds, due to the usage of exponential mapping and its inverse, they may not work in some manifolds where the inverses of exponential mappings have no closed-form expressions, for example, the Stiefel manifold.

[2] These algorithms at each outer iteration compute a full gradient at current global iterate and then it is used by agents to perform local SVRG step. Hence, these algorithms are not suitable for partial participation.

[3] We highlight that these methods overcome the non-I.I.D. data challenge only when $K = 1$ and $S = N$, i.e., all agents perform one-step local update. For $K > 1$ cases, the I.I.D. and $S = 1$ assumptions are indispensable. Hence, these algorithms are suitable for the non-I.I.D. data setting conditioned on $K = 1$ and $S = N$.

Table 2: The computational complexity of RFedAvg (Li & Ma, 2023), RFedSVRG (Li & Ma, 2023), RFedProj (Zhang et al., 2024), and RFedAGS over a compact Riemannian submanifold embedded in $\mathbb{R}^{d \times p}$. Here $N$ is the number of agents, $K$ is the number of local iterations, $B$ is the batch size, $S$ is the number of local samples, and $\mathbf{r}$, $\mathbf{ir}$, $\mathbf{v}$, $\mathbf{p}$, and $\mathbf{g}$ respectively denote the flops in a retraction evaluation, an inverse evaluation of the retraction, a vector transport evaluation, a projection evaluation onto the manifold, and a gradient evaluation of single sample loss $f_i(x; z_{i,j})$.

| | LICpA | SCC [1] | CC [1] | TCC |
|---|---|---|---|---|
| RFedAvg | $\mathbf{r}K + \mathbf{g}BK + dpK$ | $(\mathbf{ir} + dp)N + \mathbf{r}$ | $2dpN$ | $(\mathbf{ir} + dp)N + \mathbf{r}(K+1) + \mathbf{g}BK + dpK$ |
| RFedSVRG | $\mathbf{r}K + \mathbf{v}K + \mathbf{g}BK + \mathbf{g}S + 3dpK$ | $(\mathbf{ir} + 2dp)N + \mathbf{r}$ | $4dpN$ | $(\mathbf{ir} + 2dp)N + \mathbf{r}(K+1) + \mathbf{v}K + \mathbf{g}(BK + S) + 3dpK$ |
| RFedProj | $\mathbf{p}(K+2) + \mathbf{g}BK + dp(4K+3)$ | $\mathbf{p} + dp(N+2)$ | $2dpN$ | $\mathbf{p}(K+3) + \mathbf{g}BK + dp(4K + N + 5)$ |
| RFedAGS | $\mathbf{r}K + \mathbf{v}(K-1) + \mathbf{g}BK + 2dpK$ | $\mathbf{r} + dpN$ | $2dpN$ | $\mathbf{r}(K+1) + \mathbf{v}(K-1) + \mathbf{g}BK + dp(2K+N)$ |

[1] Here we assume that all agents participate in communication.

**Challenges.** In this paper, we focus on investigating an FL algorithm on general Riemannian manifolds, which works under an arbitrary participation and data heterogeneity setting. In that case, the challenges of designing and analyzing such an algorithm arise mainly from (i) the curvature effects of manifolds, (ii) multiple-step local updates at each agent, (iii) stochastic error of arbitrary participation, and (iv) data heterogeneity across agents. The biggest challenge brought about by (i) and (iii) is how the server generates new global parameters based on the local update information from multiple agents, which directly affects the design of the algorithm. However, (ii) and (iv) bring local errors into the global parameter and even make algorithms diverge, which is called agent drift effects. These issues often combine and make the convergence analysis more complicated.

**Contributions.** The main contributions of this paper are summarized as follows.

1. The server aggregation (SA) proposed in (Li & Ma, 2023) is inspired by the Euclidean weighted average (1.2). Although this SA is feasible in practice, it has significant challenges in terms of theory analysis and computation efficiency. This paper present a new SA which can avoid the issues mentioned above. The idea behind the presented SA is that it does not handle local parameters but rather averages local gradient information, which retains linearity to some extent.

2. We investigate the availability of the proposed RFedAGS under arbitrary participation and non-I.I.D. data, where the arbitrary participation setting is based on the time-invariant statistic model

without requiring prior knowledge of the participation probabilities. This model encompasses many practical scenarios, including random sampling.

3. We establish the convergence guarantees of the proposed RFedAGS under the arbitrary participation and non-I.I.D data setting with the standard assumptions in FL and Riemannian optimization except Assumption 3.8 which is important and nontrivial. We also discuss the reasonability of this assumption when using the frequencies to estimate the true probabilities.

4. Extensive numerical experiments with synthetic/real-world data are conducted to demonstrate the efficacy of the proposed RFedAGS.

**Notations.** Throughout this paper, we use $\mathbb{R}, \mathbb{R}^n$, and $\mathbb{R}^{m \times n}$ to denote the real numbers, the space real vectors of dimension $n$, and the space real matrices of size $m \times n$, respectively. We use $\mathcal{M}$ to denote the Riemannian manifold and the equipped Riemannian metric is denoted by $\langle \cdot, \cdot \rangle$, whose induced norm on the tangent space $\mathrm{T}_x \mathcal{M}$ is denoted by $\| \cdot \|_x$ (omitting the subscript sometimes). $\mathrm{Exp}, \mathrm{R}, \mathcal{T}$, and $\mathrm{grad} f$ denote exponential mapping, retraction, vector transport, and the gradient of $f : \mathcal{M} \to \mathbb{R}$, respectively. Also, $\langle \cdot, \cdot \rangle_{\mathrm{F}}, \| \cdot \|_{\mathrm{F}}$, and $\nabla f$ denote the Euclidean inner product, the norm induced by the Euclidean inner product, and the Euclidean gradient of $f$.

## 2 RFedAGS: Riemannian Federated Averaging Gradient Streams

A basic background in Riemannian geometry and optimization is assumed, and the details can be found in Appendix B. The proposed RFedAGS (stated in Algorithm 1) are explained as follows.

---

**Algorithm 1** Riemannian Federated Learning via Averaging Gradient Streams: RFedAGS

**Input:** Initial global model $x_1 \in \mathcal{M}$, number of aggregations $T$, numbers of local iterations $K$, local step size sequence $\{\alpha_t\}_{t=1}^T$, global step size $\varpi$, batch size sequence $\{B_t\}_{t=1}^T$;
**Output:** $\{x_t\}_{t=1}^{T+1}$.
1: **for** $t = 1, 2, \ldots, T$ **do**
2:     The server broadcasts $x_t$ to all agents, i.e., $x_{t,0}^j \leftarrow x_t, j \in \mathcal{N}$;
3:     **for** Agent $j \in \mathcal{N}$ in parallel **do**
4:         Set $\zeta_{t,0}^j \leftarrow 0_{x_t}$;
5:         **for** $k = 0, 1, \ldots, K-1$ **do**
6:             Agent $j$ finds indices of the mini-batch sample $\mathcal{B}_{t,k}^j$ by sampling $B_t$ times;
7:             Set $\eta_{t,k}^j \leftarrow \frac{1}{B_t} \sum_{b \in \mathcal{B}_{t,k}^j} \mathrm{grad} f_j(x_{t,k}^j; \xi_{t,k,b}^j)$;
8:             Set $x_{t,k+1}^j \leftarrow \mathrm{R}_{x_{t,k}^j}(-\alpha_t \eta_{t,k}^j)$;
9:             Set $\zeta_{t,k+1}^j \leftarrow \zeta_{t,k}^j + \mathcal{T}_{\tilde{\eta}_{t,k}^j}(\alpha_t \eta_{t,k}^j)$ with $\tilde{\eta}_{t,k}^j$ satisfying $\mathrm{R}_{x_{t,k}^j}(\tilde{\eta}_{t,k}^j) = x_t$;
10:         **end for**
11:         Upload the gradient stream $\zeta_{t,K}^j$ to the server with an unknown but fixed probability $p_j$;
12:     **end for**
13:     The server computes the approximate probability $q_t^j, \forall j \in \mathcal{S}_t$;
14:     The server updates the new global model $x_{t+1}$ by (AGS-AP) with $q_t^j$ replacing $p_j$;
15: **end for**

---

**A new Riemannian SA.** Due to the curvature effects of manifolds, the addition of two points in a manifold is not valid, and thus the SA via the weighted average of local parameters (1.2) does not work in the Riemannian setting. Li & Ma (2023) proposed a SA, called tangent mean, defined by

$$x_{t+1} \leftarrow \mathrm{Exp}_{x_t} \left( \frac{1}{|\mathcal{S}_t|} \sum_{i \in \mathcal{S}_t} \mathrm{Exp}_{x_t}^{-1}(x_{t,K}^i) \right), \tag{TM}$$

which is an approximate to the weighted average of points on a manifold. On the one hand, (TM) involves the inverse of exponential mapping, which has no closed-form expression in some manifolds, e.g., the Stiefel manifold. This limits its scope of availability. Additionally, due to the curvature effects of manifolds, exponential mapping and its inverse almost are nonlinear. Hence, when agents perform multiple-step local updates, (TM) involves multiple consecutive exponential mappings, resulting in that the increment of parameters, $\mathrm{Exp}_{x_t}^{-1}(x_{t+1})$, is difficult to be bounded in analysis, which makes convergence analysis fairly challenging. In view of the discussions above, this paper

resorts to another aggregation which can not only implement SA efficiently but also analyze algorithm convergence conveniently.

Back to the Euclidean setting, the increment of parameters of FedAvg can be expanded as

$$\Delta_t = x_{t+1} - x_t = -\alpha_t \frac{1}{|\mathcal{S}_t|} \sum_{i \in \mathcal{S}_t} \sum_{k=0}^{K-1} \frac{1}{B_t} \sum_{b \in \mathcal{B}_{t,k}^i} \nabla f_i(x_{t,k}^i; \xi_{t,k,b}^i).$$

Observing the expression shows that the increment of parameters is given by the average of mini-batch gradients of active agents. We can adopt the similar idea in the Riemannian setting but require making some adaptations, since directly combining the mini-batch gradients located in different tangent spaces is not well defined. With the aid of vector transport, the combination can be defined. Specifically, we define the the Riemannian "increment of parameters" as

$$\zeta_t = \mathrm{R}_{x_t}^{-1}(x_{t+1}) = -\alpha_t \frac{1}{|\mathcal{S}_t|} \sum_{j \in \mathcal{S}_t} \sum_{k=0}^{K-1} \frac{1}{B_t} \sum_{b \in \mathcal{B}_{t,k}^j} \mathcal{T}_{\tilde{\eta}_{t,k}^j} (\mathrm{grad} f_j(x_{t,k}^k; \xi_{t,k,b}^j)).$$

Specific to agent $j$, it just need to upload $\zeta_{t,K}^j = \alpha_t \sum_{k=0}^{K-1} \frac{1}{B_t} \sum_{b \in \mathcal{B}_{t,k}^j} \mathcal{T}_{\tilde{\eta}_{t,k}^j} (\mathrm{grad} f_j(x_{t,k}^k; \xi_{t,k,b}^j))$, called gradient stream, to the server. The resulting new SA is given via averaging gradient streams:[1]

$$x_{t+1} = \mathrm{R}_{x_t}(\zeta_t) = \mathrm{R}_{x_t}\left( -\frac{1}{|\mathcal{S}_t|} \sum_{j \in \mathcal{S}_t} \zeta_{t,K}^j \right). \tag{AGS-RS}$$

It is worth noting that when the manifold reduces to a Euclidean space, (AGS-RS) is equivalent to the Euclidean SA (1.2). In our opinion, this aggregation is a more essential generalization from the Euclidean setting to the Riemannian setting.

From the perspective of geometry, tangent mean (TM) "projects" the final inner iterates $x_{t+K}^j$ back to the tangent space at $x_t$, then averages them and finally retracts the average into the manifold. While in aggregation (AGS-RS), the intermediary negative mini-batch-gradients $-\frac{1}{B_t} \sum_{b \in \mathcal{B}_{t,k}^j} \mathrm{grad} f_i(x_{t,k}^j; \xi_{t,k,b}^j)$ are transported to the tangent space at $x_t$ in some way, then averages them and finally retracts the average into the manifold. The (TM) actually is an approximation of the weighted averages of inner iterates $x_{t,K}^j$. When the degree of heterogeneity across clients are large, the inner $x_{t,K}^j$ is closer to the minimizer of local function $f_j$, and their average may be far away from the minimizer of the global function; while, the proposed (AGS-RS) leverages the gradient information drawn from clients to generate global direction and thus helps to alleviate this bias; see Figure 1(c). In particular, letting the proposed aggregation (AGS-RS) use the exponential map and parallel transport, the two aggregations coincide when (i) $\mathcal{M} = \mathbb{R}^d$; or (ii) $K = 1$. See Figure 1 for a geometric interpretation and an experimental comparison of (TM) and (AGS-RS).

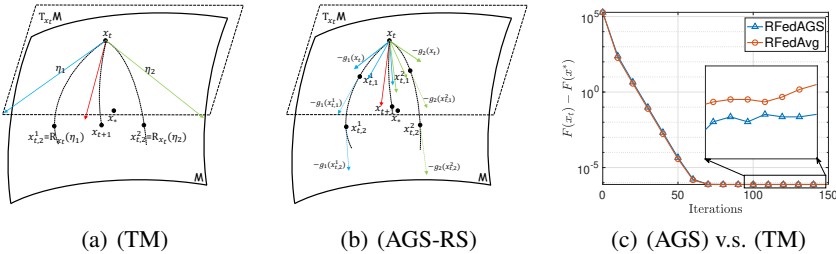

|     (a) (TM)     |     (b) (AGS-RS)     |     (c) (AGS) v.s. (TM)     |

Figure 1: (a)-(b) diagrams of (TM) and (AGS-RS) where $K = 2$, two agent participate in communication, and $g_i(x)$ denotes the local stochastic gradient of agent $i$ at $x$. (c) (AGS) v.s. (TM) on $\min_{x \in \{x \in \mathbb{R}^{50} : x^T x = 1\}} F(x) = -\frac{1}{2}\left( \frac{1}{60} \sum_{j=1}^{60} (x^T Z_{1,j} Z_{1,j}^T x + x^T Z_{2,j} Z_{2,j}^T x) \right)$.

**Arbitrary partial participation.** Now we are ready to extend (AGS-RS) to the arbitrary partial participation setting under consideration, which is formally modeled in Assumption 2.1.

---

[1]It should be highlighted that vector transport used in (AGS-RS) (subsequent (AGS-AP)) usually does not need inverse of retraction; see Line 9 in Algorithm 1.

**Assumption 2.1.** *Assume that each agent $i$ independently participates in any round of communication with probability $p_i > 0$.*

Under Assumption 2.1, when the participation probabilities are not exactly equal to each other, using (AGS-RS) simply may introduce stochastic participation errors. In that case, the next theorem points out that the algorithm equipped with (AGS-RS) may work incorrectly since it may solve another problem different from the original problem.

**Theorem 2.1** (Proved in Appendix E.1). *Under Assumption 2.1, let $\mathcal{S}_t \neq \emptyset$ denotes the set of agents who respond to the server at the $t$-th round of communication. Then, $\mathbb{E}\left[\sum_{j \in \mathcal{S}_t} \frac{1}{|\mathcal{S}_t|} \mathrm{grad} f_j(x)\right] = \sum_{i=1}^N \tilde{p}_i \mathrm{grad} f_i(x)$, with $\tilde{p}_i = p_i \int_0^1 \prod_{j \neq i}^N (1 - p_j + p_j \tau) \mathrm{d}\tau$.*

Therefore, if $p_i \neq p_j$ for some $i, j \in [N]$, then $\tilde{p}_i \neq \tilde{p}_j$, and thus there exists no $\chi > 0$ such that $\sum_{i=1}^N \tilde{p}_i \mathrm{grad} f_i(x) = \chi \mathrm{grad} F(x)$. That is, the algorithm may not solve the original problem $\min_{x \in \mathcal{M}} F(x)$ since each of its search directions leads the iterate $x_t$ to the minimizer of another problem $\min_{x \in \mathcal{M}} \tilde{F}(x) := \sum_{i=1}^N \tilde{p}_i f_i(x)$.

Back again to Assumption 2.1, at the $t$-th round of communication, note that

$$\mathbb{E}\left[\sum_{i \in \mathcal{S}_t} \frac{1}{p_i N} \mathrm{grad} f_i(x)\right] = \mathbb{E}\left[\sum_{i=1}^N \frac{1}{p_i N} \mathbb{I}_{\mathcal{S}_t}(i) \mathrm{grad} f_i(x)\right] = \sum_{i=1}^N \frac{1}{p_i N} \mathbb{E}\left[\mathbb{I}_{\mathcal{S}_t}(i) \mathrm{grad} f_i(x)\right]$$

$$= \sum_{i=1}^N \frac{1}{p_i N} \left(p_i \mathrm{grad} f_i(x)\right) = \mathrm{grad} F(x), \tag{2.1}$$

where $\mathbb{I}_{\mathcal{S}_t}(i) = 1$ if $i \in \mathcal{S}_t$ otherwise $\mathbb{I}_{\mathcal{S}_t}(i) = 0$. Hence, if the participation probabilities, $p_i$'s, are known, one of the feasible aggregation patterns can be chosen as

$$x_{t+1} \leftarrow \mathrm{R}_{x_t}\left(-\varpi \sum_{i \in \mathcal{S}_t} \frac{1}{p_i N} \zeta_{t,K}^i\right) \text{ with } \varpi > 0 \text{ the global step size,} \tag{AGS-AP}$$

which ensures that the algorithm correctly solves the original problem $\min_{x \in \mathcal{M}} F(x)$.

On the other hand, in practical applications, the server is actually unaware of the true probabilities. In this case, what the server can do is to estimate the true probabilities as possible in some ways, that is, the server computes $q_t^i$ in the $t$-th round of communication and uses it to serve as the true probability $p_i$. Summarizing above, this paper proposes a Riemannian FL algorithm, called RFedAGS, which can address the partial participation setting, as stated in Algorithm 1.

## 3 CONVERGENCE ANALYSIS

In this section, we establish the convergence properties of RFedAGS (Algorithm 1) on the partial participation and the non-I.I.D. data settings. All of the proofs can be found in Appendix D.

### 3.1 ASSUMPTIONS

We first present a set of assumptions as follows that are necessary for the convergence analysis. All assumptions except Assumption 3.8 have been used in e.g., (Bonnabel, 2013; Tripuraneni et al., 2018; Sato et al., 2019; Han & Gao, 2021), and their reasonability is discussed in Appendix C.

**Assumption 3.1.** *The retraction $\mathrm{R}$ is such that its restriction to $\mathrm{T}_x\mathcal{M}$ for all $x \in \mathcal{M}$, $\mathrm{R}_x$, is of class $C^2$, and the associated vector transport $\mathcal{T}$ is continuous and bounded in the sense that there exists a constant $\Upsilon > 0$ such that for any $x \in \mathcal{M}$, $\zeta_x, \eta_x \in \mathrm{T}_x\mathcal{M}$, it holds that $\|\mathcal{T}_{\eta_x}(\zeta_x)\| \leq \Upsilon \|\zeta_x\|$.*

**Assumption 3.2.** *For a sequence of the outer iterates $\{x_t\}_{t \geq 1}$ and a sequence of the inner iterates $\{\{\{x_{t,k}^j\}_{j=1}^N\}_{k=0}^{K-1}\}_{t \geq 1}$ generated by Algorithm 1, there exists a $W$-totally retractive set $\mathcal{W} \subset \mathcal{M}$ such that $\{x_t\}_{t \geq 1} \subset \mathcal{W}$ and $\{\{\{x_{t,k}^j\}_{j=1}^N\}_{k=0}^{K-1}\}_{t \geq 1} \subset \mathcal{W}$. The minimizers of Problem (1.1) are inside $\mathcal{W}$. Additionally, there exists a compact and connected set $\mathcal{X} \subset \mathcal{M}$ such that $\mathcal{W} \subset \mathcal{X}$.*

**Assumption 3.3.** *The cost function $F$ is continuously differentiable in $\mathcal{W}$, the local cost functions $f_1, \ldots, f_N$ are continuously differentiable in $\mathcal{W}$, and their components $f_j(\cdot, \xi)$ for $\xi \sim \mathcal{D}_j$ with $j \in [N]$ are continuously differentiable in $\mathcal{W}$.*

**Assumption 3.4.** *The local objective functions $f_j$, $j \in [N]$, are $L_f$-Lipschitz continuously differentiable in $\mathcal{W}$ with the retraction $\mathrm{R}$ and the vector transport $\mathcal{T}$ (see Definition B.1), implying that $F$ is also $L_f$-Lipschitz continuously differentiable.*

**Assumption 3.5.** *$F$ is $L_g$-retraction smooth over $\mathcal{W}$ with respect to $\mathrm{R}$ (see Definition B.2).*

**Assumption 3.6.** *For any parameter $x \in \mathcal{M}$, the Riemannian stochastic gradient $\mathrm{grad} f_j(x; \xi^j)$ is an unbiased estimator of the gradient $\mathrm{grad} f_j(x)$, i.e., $\mathbb{E}_{\xi^j}[\mathrm{grad} f_j(x; \xi^j)] = \mathrm{grad} f_j(x)$, $\forall j \in [N]$.*

**Assumption 3.7.** *For any fixed parameter $x \in \mathcal{M}$, there exists a positive constant $\sigma_L$ such that for all $j \in [N]$, it holds that $\mathbb{E}[\|\frac{1}{B}\sum_{b \in \mathcal{B}^j} \mathrm{grad} f_j(x; \xi_b^j) - \mathrm{grad} f_j(x)\|^2] \leq \frac{\sigma_L^2}{B}$ with $|\mathcal{B}^j| = B$.*

The method estimating the probabilities is discussed in Section 3.3. Now we just make an assumption requiring that the approximate probability $q_t^i$ in each round of communication is not far away from the true probability $p_i$, formally stated in Assumption 3.8.

**Assumption 3.8.** *There exist constants $q_{\min}, q_{\max} \in (0, 1]$ and $G \geq 0$ independent of $t \geq 1$ and $i \in [N]$, such that the approximate probabilities $q_t^i$'s satisfy $\left|\frac{1}{q_t^i} - \frac{1}{p_i}\right| \leq \sqrt{G\alpha_t}$, that $q_{\min} \leq q_t^i \leq q_{\max}$, and that $q_t^i$ is independent of $\mathcal{S}_t$, $\forall t \geq 1, i \in [N]$, where $\alpha_t$ is the local step size in the $t$-th round of communication.*

Note that the constant $G$ controls the accuracy of the approximate probabilities and when the true probabilities are available to the server, $G$ can take exactly zero. In Section 3.3, we discuss the reasonability of Assumption 3.8.

**Remark 3.1.** *In (Wang & Ji, 2024), the authors imposed the following bound on the approximate probabilities: $\sum_{i=1}^N p_i^2 \left(\frac{1}{q_t^i} - \frac{1}{p_i}\right)^2 \leq \frac{N}{81}$. This bound essentially requires that $|\frac{1}{q_t^i} - \frac{1}{p_i}|$ is less than some constant, which is consistent with Assumption 3.8 in fixed step size cases. Note that this assumption is considered in (Wang & Ji, 2024) only for fixed step size cases, but Assumption 3.8 considers another situation where the bound varies over time $t$ when decaying step sizes are used.*

### 3.2 CONVERGENCE PROPERTIES

In this section, we establish the convergence properties of the proposed RFedAGS.

**Theorem 3.1.** *Let Assumptions 3.1-3.8 hold. Suppose Algorithm 1 is run with a fixed global step size $\varpi > 0$ and a decaying local step size sequence $\{\alpha_t\}$ satisfying Conditions*

$$\sum_{t=1}^{\infty} \alpha_t = \infty, \sum_{t=1}^{\infty} \alpha_t^2 < \infty. \tag{3.1}$$

*Then, $\liminf_{t \to \infty} \mathbb{E}[\|\mathrm{grad} F(x_t)\|^2] = 0$.*

In what follows, we further characterize the nonasymptotic convergence.

**Theorem 3.2.** *Under the same conditions as Theorem 3.1 except that the local step size sequence $\{\alpha_t\}$ is determined by $\alpha_t = \frac{\alpha_0}{(\beta+t)^p}$ with constants $\alpha_0, \beta > 0$ and $p \in (1/2, 1]$ satisfying $\varpi \alpha_1 K L_g \leq 1$, the weighted average norm of the squared gradients satisfy, with $A_T = \sum_{t=1}^T \alpha_t$,*

$$\frac{1}{A_T} \sum_{t=1}^T \alpha_t \mathbb{E}[\|\mathrm{grad} F(x_t)\|^2] \leq \begin{cases} \mathcal{O}(\frac{1}{\ln(\beta+T)}) & p = 1, \\ \mathcal{O}(\frac{1}{(\beta+T)^{1-p}}) & p \in (1/2, 1). \end{cases}$$

**Remark 3.2.** *In particular, if the full agent participate in any round of communication and agents use the full local gradient in local update, i.e., $G = 0$ and $\sigma_L = 0$, one can relax the step sizes to $\alpha_t = \frac{\alpha_0}{(\beta+t)^p}$ where $p = 1/3 + a$ with $a \in (0, 2/3)$. In this case, for large $T$, the upper bound can be improved to $\frac{1}{A_T} \sum_{t=1}^T \alpha_t \mathbb{E}[\|\mathrm{grad} F(x_t)\|^2] \leq \mathcal{O}(\frac{1}{(\beta+T)^{2/3-a}})$ (see Appendix D.3).*

**Theorem 3.3.** *Under Assumptions 3.1-3.8, suppose that $F$ satisfies RPL condition, i.e., there exists a constant $\mu > 0$, such that for all $x \in \mathcal{W}$, it holds that $F(x) - F(x^*) \leq \frac{1}{2\mu}\|\mathrm{grad} F(x)\|^2$. If we run Algorithm 1 with the batch size $B_t \in [B_{\mathrm{low}}, B_{\mathrm{up}}]$ and the step sizes satisfying $\alpha_t = \frac{\beta}{\gamma+t}$ for some $\gamma > 0$ and $\beta > \frac{1}{\mu\varpi K}$ such that $\alpha_1 \varpi K L_g \leq 1$, then the iterates $\{x_t\}_{t \geq 1}$ satisfy*

$$\mathbb{E}[F(x_t)] - F(x^*) \leq \frac{\nu}{\gamma+t}, \quad and \quad \mathbb{E}[\|\mathrm{grad} F(x_t)\|^2] \leq \frac{2L_g\nu}{\gamma+t}, \tag{3.2}$$

*where $\nu = \max\left\{\frac{\varpi K\beta^2 Q(K,B_{\mathrm{low}},\alpha_1,\varpi)}{\beta\mu\varpi K-1}, (\gamma+1)\Theta(x_1)\right\}$, $\Theta(x_1) = F(x_1) - F(x^*)$, and $Q(K, B_t, \alpha_t, \varpi) = (2K-1)(K-1)L_f^2\delta_1^2 P^2(J^2 + \alpha_t^2 P^2 H^2)\alpha_t/6 + GP^2\delta_2^2 + \Upsilon^2 P^2\delta_4^2 KL_g\varpi +$*

$\frac{L_g \delta_3^2 \sigma_L^2 \Upsilon^2 \varpi}{2B_t}$ *with $P, J$, and $H$ being three constants depended on the problem, manifold and the retraction and $\delta_1, \delta_2, \delta_3, \delta_4$ being constants depended on $q_t^i, p_i, \forall i \in [N]$. That is, Algorithm 1 converges sublinearly to the minimizer in expectation.*

Theorems 3.1-3.3 provide the global convergence of Algorithm 1. Under mild assumptions, the first theorem states that Algorithm 1 has global convergence in expectation for general objectives while the other theorems further provide the convergence rate of Algorithm 1. However, all of these theorems require the usage of the decaying step sizes. When decaying step sizes are used, a large number of iteration are required for Algorithm 1 to converge. A compromise is to use a fixed step size of moderate size, the advantage of which is that the convergence rate is sublinear (even linear) while the disadvantage of which is that it may not converge to the minimizers but to an $\epsilon$-stationary point/solution (see Definition B.4); see Theorems 3.4 and 3.5.

**Theorem 3.4.** *Suppose that Assumptions 3.1-3.8 hold. We run Algorithm 1 with a fixed global step size $\varpi$, a fixed batch size $B$, and a fixed number of local updates $K$.*

*1. If the fixed step sizes $\alpha$ and $\varpi$ satisfy $\alpha \varpi K L_g \leq 1$, then*

$$\frac{1}{T} \sum_{t=1}^{T} \mathbb{E}[\|\mathrm{grad} F(x_t)\|^2] \leq \frac{2\Theta(x_1)}{\varpi \alpha K T} + 2\alpha Q(K, B, \alpha, \varpi). \tag{3.3}$$

*2. If the true probabilities are known, meaning $G = 0$, and one takes local and global step sizes $\alpha$ and $\varpi$ such that $\alpha \varpi = \sqrt{\frac{\Theta(x_1)B}{(\delta_3^2 \sigma_L^2 + 2P^2 \delta_4^2 KB)\Upsilon^2 L_g KT}}$ with $T$ satisfying $T \geq$*

$\max\left\{ \frac{KL_g \Theta(x_1)B}{(\delta_3^2 \sigma_L^2 + 2P^2 \delta_4^2 KB)\Upsilon^2}, \frac{\Theta(x_1)(2K-1)^2(K-1)^2 L_f^4 \delta_1^4 P^4 (L_g^2 \varpi^2 J^2 K^2 + P^2 H^2)^2 B^3}{9(\delta_3^2 \sigma_L^2 + 2P^2 \delta_4^2 KB)^3 \Upsilon^6 L_g^7 \varpi^6 K^5} \right\}$, *then*

$$\frac{1}{T} \sum_{t=1}^{T} \mathbb{E}[\|\mathrm{grad} F(x_t)\|^2] \leq 4\Upsilon \sqrt{L_g \Theta(x_1)\left(\frac{\delta_3^2 \sigma_L^2}{KTB} + \frac{2P^2 \delta_4^2}{T}\right)}.$$

**Remark 3.3.** *If the probabilities $p_i$ are known, i.e., $q_t^i = p_i$, and $p_{\min} = \min_i\{p_i\}$ is not too small and not fairly far away from $p_{\max} = \max_i\{p_i\}$, such that the constants $\delta_1^2, \delta_2^2, \delta_3^2, \delta_4^2$ are $\delta_1^2 = \frac{1}{N}\sum_{j=1}^{N}\left(\frac{p_j}{q_t^j}\right) = 1, \delta_2^2 = \sum_{j=1}^{N}\frac{p_j^2}{N} \leq 1, \delta_3^2 = \frac{1}{N^2}\sum_{j=1}^{N}\frac{p_j}{(q_t^j)^2} \leq \frac{1}{Np_{\min}}, \delta_4^2 = \frac{1}{N^2}\sum_{j=1}^{N}\frac{(1-p_j)}{p_j} \leq \frac{1}{Np_{\min}}$, then, Item 2 gives the upper bound as $\mathcal{O}(\frac{1}{\sqrt{p_{\min}NKTB}}) + \mathcal{O}(\frac{1}{\sqrt{p_{\min}NT}})$.*
*In particular, if the probabilities are the same across agents, e.g., $p_i = \frac{S}{N}$ with $S \leq N$, then $\delta_3^2 = \frac{1}{S}$, and $\delta_4^2 = \frac{N-S}{NS} \leq \frac{1}{S}$. It follows that Item 2 gives the upper bounds as $\mathcal{O}(\frac{1}{\sqrt{SKTB}}) + \mathcal{O}(\frac{1}{\sqrt{ST}})$. The bound of $\mathcal{O}(\frac{1}{\sqrt{ST}})$ matches with the existing result for FedAvg given in (Karimireddy et al., 2020, Theorem 1) and improves by $\frac{1}{\sqrt{K}}$ over that given in (Yang et al., 2021, Corollary 2).*

**Theorem 3.5.** *Under Assumptions 3.1-3.8, suppose that $F$ satisfies RPL condition with a constant $\mu > 0$. If we run Algorithm 1 with batch size $B_t \in [B_{\mathrm{low}}, B_{\mathrm{up}}]$ and step sizes $\alpha_t = \alpha$ and $\varpi$ satisfying $\alpha \varpi K \leq \min\{1/L_g, 1/\mu\}$, then the resulting iterates $\{x_t\}_{t=1}^{T}$ satisfy*

$$\mathbb{E}[F(x_T)] - F(x^*) \leq (1 - \mu \varpi K \alpha)^{T-1}\Theta(x_1) + \frac{\alpha}{\mu}Q(K, B_{\mathrm{low}}, \alpha, \varpi) \xrightarrow{T \to \infty} \frac{\alpha}{\mu}Q(K, B_{\mathrm{low}}, \alpha, \varpi). \tag{3.4}$$

From Theorem 3.5, if one lets $T \to \infty$, then the expected optimality gaps $\{\mathbb{E}[F(x_T)] - F(x^*)\}$ are bounded from above by $\frac{\alpha}{\mu}Q(K, B_{\mathrm{low}}, \alpha, \varpi)$, which implies that any accumulation point of the sequence of iterates $\{x_t\}$ generated by Algorithm 1 is a $\epsilon$-solution if taking $\alpha \leq \frac{\epsilon \mu}{Q(K, B_{\mathrm{low}}, \alpha, \varpi)}$. Smaller $\alpha$ means smaller upper bound as well as slower convergence speed.

**Remark 3.4.** *Similar to the Euclidean setting, the RPL property is weaker than the strong retraction-convexity. In fact, if the objective $f : \mathcal{M} \to \mathbb{R}$ is $\mu$-strongly retraction-convex, then it also satisfies the RPL property with parameter $\mu$ (proved in Appendix E.2). Therefore, Theorems 3.3 and 3.5 also hold under strong retraction-convexity.*

### 3.3 ESTIMATING THE PARTICIPATION PROBABILITIES

At the $t$-th round of communication, let $\mathcal{S}_t$ denote the set of participating agents. Then, under Assumption 2.1, $\mathbb{I}_{\mathcal{S}_t}(i)$ follows the Bernoulli distribution, i.e., $\mathbb{I}_{\mathcal{S}_t}(i) \sim \mathrm{Bernoulli}(p_i)$. At each round of communication, for each agent $i$, whether it participates in communication can be regarded

as a Bernoulli trial. Therefore, by Bernoulli's Large Number Theorem, the frequency of agent $i$ participating in communication goes closely to the true probability $p_i$ as the growth of $t$, the number of communications. Formally, let $\mathsf{q}_t^i = \sum_{\tau=1}^{t-1} \mathbb{I}_{\mathcal{S}_\tau}(i)$, and compute the approximate probability by $q_t^j = \mathsf{q}_t^j/(t-1)$, ensuring the independency of $q_t^j$ and $\mathcal{S}_t$. Then we have $\lim_{t\to\infty} \mathbb{P}\{|q_t^i - p_i| \le \epsilon\} = 1$ for any small $\epsilon > 0$. This justifies the use of frequencies to estimate probabilities. The next theorem shows that Assumption 3.8 holds with high probability when the step size takes the form of $\alpha_t = \mathcal{O}(t^{-a})$ with $a \in (1/2, 1] \cup \{0\}$.

**Theorem 3.6** (Proved in Appendix D.7). *Under Assumption 2.1, for each agent $i$, we have*

$$\mathbb{P}\left\{\left|\frac{1}{q_t^i} - \frac{1}{p_i}\right| \le \mathcal{G}t^{-\frac{a}{2}}\right\} \ge 1 - \min\left\{2e^{-\frac{\hat{t}p_i^2}{2}}, \frac{4(1-p_i)}{\hat{t}p_i}\right\} - \min\left\{2e^{-\frac{\mathcal{G}^2 p_i^4}{2}\hat{t}t^{-a}}, \frac{4(1-p_i)}{\mathcal{G}^2 p_i^3 \hat{t}t^{-a}}\right\}, \quad (3.5)$$

*where $q_t^i = \sum_{\tau=1}^{\hat{t}} \mathbb{I}_{\mathcal{S}_\tau}(i)/\hat{t}$, $\hat{t} = t - 1$, and $\mathcal{G}$ and $a$ are nonnegative constants.*

In practice, taking $a = 0$ leads to fixed step size cases or $a \in (1/2, 1]$ to decaying step size cases. Therefore, it follows from Theorem 3.6 that Assumption 3.8 holds with probability not less than $1 - \min\{2e^{-\frac{\hat{t}p_i^2}{2}}, \frac{4(1-p_i)}{\hat{t}p_i}\} - \min\{2e^{-\frac{\mathcal{G}^2 p_i^4}{2}\hat{t}t^{-a}}, \frac{4(1-p_i)}{\mathcal{G}^2 p_i^3 \hat{t}t^{-a}}\}$ with a proper constant $\mathcal{G} \ge 0$. Large enough $t$ and properly chosen $\mathcal{G}$ make the probability high.

## 4 EXPERIMENTS

Here we conduct numerical experiments on principal component analysis (PCA) over the Stiefel manifold, hyperbolic structured prediction (HSP) over the hyperbolic manifold, and the Fréchet mean computation (FMC) over the SPD manifold such that we can compare RFedAGS [2] with existing RFL algorithms, including RFedAvg (Li & Ma, 2023), RFedSVRG (Li & Ma, 2023), RFedProj (Zhang et al., 2024) (used in PCA), and ZO-RFedProj (Wang et al., 2025) (used in PCA). **Additionally, we still conduct two experiments on principal eigenvector computation and low-rank matrix completion shown in Appendices A.1-A.2.** The first one tests the comprehensive performance of RFedAGS, while the second compares RFedAGS with some existing centralized algorithms showing the comparable availability of RFedAGS with those. **The experiment settings in this section can be found in Appendix A.3**.

**PCA.** The PCA problem has the form of $\min_{X \in \mathrm{St}(r,d)} F(X) := \frac{1}{N}\sum_{i=1}^N f_i(X)$, with $f_i(X) = -\frac{1}{S}\sum_{j=1}^S \mathrm{tr}(X^T(Z_{ij}Z_{ij}^T)X)$, where $\mathrm{St}(r,d)$ is the Stiefel manifold, $Z_{ij}Z_{ij}^T$ is the covariance matrix of local datum $Z_{ij} \in \mathbb{R}^{d \times p}$. We generate $\mathcal{D}_i = \{Z_{ij}\}_{j=1}^S$ in two ways: (i) synthetic data by sampling from the Gaussian distribution $\mathcal{N}(0, \frac{i}{N})$ such that $\mathcal{D}_i$ are non-I.I.D; (ii) real-world data from CIFAR10 [3] dataset. We can observe from Figure 2 that our proposed RFedAGS outperforms the existing three RFL algorithms under the arbitrary participation setting in terms of accuracy of solutions and consumed time. This justifies the efficacy of the proposed RFedAGS.

It should be noted that the tools used in our RFedAGS are fairly general (as stated in Assumption 3.1), however RFedAvg and RFedSVRG require more strict tools (the inverse of exponential and parallel transport), and RFedProj and ZO-RFedProj require the orthogonal projector onto the manifold. These requirements limit the application scope of RFedAvg, RFedSVRG, RFedProj and ZO-RFedProj. For instance, RFedProj and ZO-RFedProj can not be used in the HSP and FMC problems below.

**HSP.** Given a set of training pairs $\mathcal{D} = \{\mathcal{D}_i\}_{i=1}^N = \{\{(w_{i,j}, y_{i,j})\}_{j=1}^S\}_{i=1}^N$, where $w_{i,j} \in \mathbb{R}^r$ is the feature and $y_{i,j} \in \mathcal{H}^d$ is the hyperbolic embedding of the class of $w_{i,j}$. Then for a test sample $w$, the task of HPS is to predict its hyperbolic embeddings by solving the following problem $\arg\min_{x \in \mathcal{H}^d} F(x) := \frac{1}{N}\sum_{i=1}^N f_i(x)$, with $f_i(x) = \frac{1}{S}\sum_{j=1}^S a_{i,j}(\omega)\mathrm{dist}^2(x, y_{i,j})$ where the hyperbolic manifold $\mathcal{H}^d$ is characterized via the Lorentz hyperbolic model, $[a_1(w), \ldots, a_N(w)]^T \in \mathbb{R}^{N \times S}$ is a parameter matrix. We use the WordNet [4] dataset to test RFedAGS, RFedAvg, and RFedSVRG. From the reported Figure 3, we can observe that the proposed RFedAGS outperforms both RFedAGS and RFedSVRG in terms of distance to the true point. Figure 3(c) directly demonstrates this advantage of RFedAGS.

---

[2] Our code is available at https://github.com/zhenwei-huang/RFedAGS.git.

[3] See https://www.cs.toronto.edu/ kriz/cifar.html.

[4] See https://wordnet.princeton.edu/.

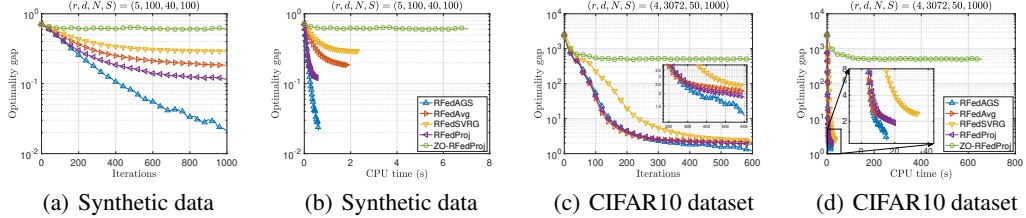

(a) Synthetic data   (b) Synthetic data   (c) CIFAR10 dataset   (d) CIFAR10 dataset

Figure 2: PCA: RFedAGS consistently performs better than the competing methods across both synthetic and real datasets.

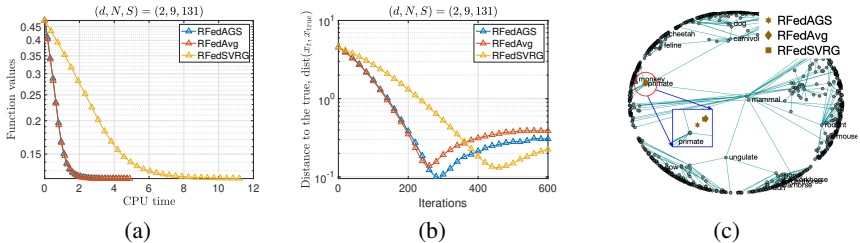

(a)   (b)   (c)

Figure 3: HSP with WordNet dataset. Here "primate" is the test sample (true point).

**FMC.** Given a set of SPD matrices, $\mathcal{D} = \{\{X_{i,j}\}_{j=1}^{S}\}_{i=1}^{N}$, the FMC of these SPD matrices is the solution to the problem $\arg\min_{X \in \mathcal{S}_{++}^{n}} F(X) := \frac{1}{N}\sum_{i=1}^{N} f_i(X)$ with $f_i(X) = \frac{1}{S}\sum_{j=1}^{S} \text{dist}^2(X, X_{i,j})$, where $\text{dist}(\cdot, \cdot)$ is the Riemannian distance. We use the PATHMNIST [5] dataset to test the algorithms. From Figure 4, we still observe that RFedAGS outperforms RFedAvg and RFedSVRG.

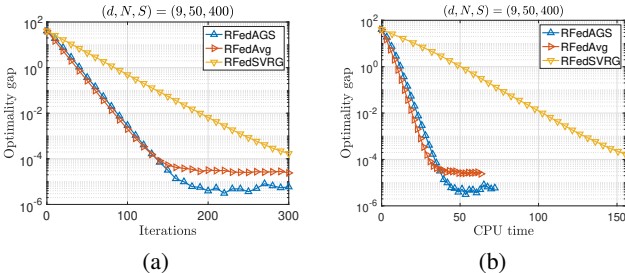

(a)   (b)

Figure 4: FMC with PATHMNIST dataset: RFedAGS consistently performs better than RFedAvg and RFedSVRG.

## 5   CONCLUSIONS

In this work, we propose a Riemannian FL algorithm, called RFedAGS, that addresses critical challenges caused by curvature effects of manifolds, the partial participation, and the heterogeneity data. Unlike the commonly studied random sampling setting, RFedAGS accommodates a more practical and challenging scenario where agents' participation statistics may be unknown. Theoretically, we prove that the proposed RFedAGS, under decaying step sizes, achieves global convergence and provide sublinear convergence rate. When using a fixed step size, it attains sublinear—or even linear—convergence near a neighborhood of a stationary point/solution. Numerical experiments we conducted have confirmed the efficacy of RFedAGS and in particular, it outperforms existing RFL algorithms methods on PCA, HSP, and FMC with synthetic and real-world data.

Current analyses on partial participation rely on time-invariant statistical assumptions. An important direction for future research is to analyze more realistic and complex scenarios, such as settings with time-varying participation probabilities.

---

[5]See https://medmnist.com/.

ACKNOWLEDGMENTS

WH was partially supported by the National Natural Science Foundation of China (No. 12371311), the Natural Science Foundation of Fujian Province (No. 2023J06004), and the Fundamental Research Funds for the Central Universities (No. 20720240151).

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

# Appendix

## A    EXPERIMENT SETTINGS AND ADDITIONAL EXPERIMENT RESULTS

In this section, we supplement the numerical experiments conducted to demonstrate the performance of RFedAGS (Algorithm 1) on non-I.I.D. data setting. We focus on the finite sum minimization of (1.1).

The decaying local step size is determined by the following formula

$$\alpha_t = \begin{cases} \alpha_0 & \text{if } t = 0, \\ \frac{\alpha_0}{\beta + c_t} & \text{if } t \geq 1, \end{cases} \quad \text{with } c_t = \begin{cases} 0 & \text{if } t = 0, \\ c_{t-1} + 1 & \text{if } \mathrm{mod}(t, \mathrm{d}) = 0, \\ c_{t-1} & \text{otherwise}, \end{cases}$$

where $\alpha_0$ and $\beta$ are two positive constants, and $\mathrm{d}$ is a positive constant integer, which results in the step size decaying once after each $\mathrm{d}$ iterations. Optimality gap defined as $F(x_t) - F(x^*)$ with $x^* \in \arg\min_{x \in \mathcal{W}} F(x)$ is a commonly-used measure to evaluate the performance of algorithms. In all experiments, the global step size is set as $1$. The CPU time consists of the server computation time and the local computation time of active agents, without the communication time between the server and agents. Unless otherwise specified, frequencies are used in Algorithm 1 to estimate the true probabilities. All of algorithms involved in our experiments are implemented built on Manopt (Boumal et al., 2014). All of the experiments are conducted under Windows 11 and MATLAB R2024b running on a laptop (Intel(R) Core(TM) i7-1165G7 CPU @2.80GHz, 16.0G RAM).

### A.1    COMPREHENSIVE TESTS

Consider the principal eigenvector computation (PEC) problem over the sphere manifold, formulated as follows

$$\min_{x \in \mathbb{S}^{n-1}} F(x) := \frac{1}{N} \sum_{i=1}^{N} f_i(x), \text{ with } f_i(x) = -\frac{1}{S} \sum_{j=1}^{S} x^T z_{i,j} z_{i,j}^T x, \tag{A.1}$$

where $\mathbb{S}^{n-1} = \{x \in \mathbb{R}^n : x^T x = 1\}$ is the sphere manifold, $\mathcal{D}_i = \{z_{i,1}, \ldots, z_{i,S}\}$ is the local samples held by agent $i$. Problem (A.1) is in the form of finite sum minimization of (1.1).

The sphere manifold $\mathbb{S}^{n-1}$ is viewed as a Riemannian embedded submanifold of $\mathbb{R}^n$, that is, the Riemannian metric is induced by the Euclidean metric: $\langle \xi, \eta \rangle_x = \xi^T \eta$ for all $\xi, \eta \in \mathrm{T}_x \mathbb{S}^{n-1}$. The exponential mapping is chosen as the Retraction and the parallel transport along the geodesic correspondingly is selected as the isometric vector transport. The MNIST dataset (Deng, 2012) [6] consists of 60000 hand-written gray images of size $28 \times 28$ each of which is associated with a label taking values from $0$ to $9$. In our experiments, each image is concatenated into a 784-dimensional column vector by column. In addition, to test the effectiveness of the proposed RFedAGS under the heterogeneity data setting, according to the FL setting, the MNIST dataset is shuffled into different levels of heterogeneity following the way in (McMahan et al., 2017). Figure 5 demonstrates histograms of the MNIST dataset with three different levels of heterogeneity.

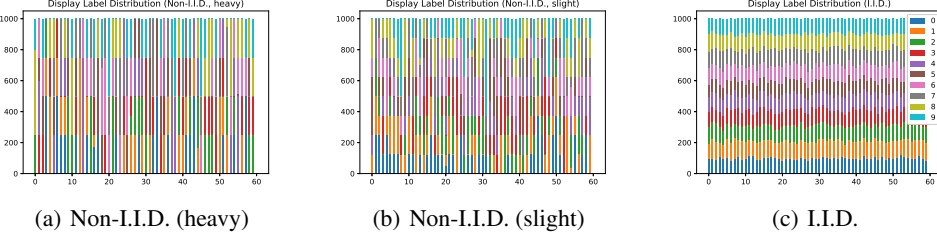

(a) Non-I.I.D. (heavy)    (b) Non-I.I.D. (slight)    (c) I.I.D.

Figure 5: Sample distributions across different agents on MNIST dataset. $x$-axis is the ID of each agents and $y$-axis is the number of local samples.

---

[6]See https://yann.lecun.com/exdb/mnist/.

### A.1.1 COMPARISON OF TWO AGGREGATION PATTERNS

First we demonstrate the importance of the aggregation pattern (AGS-AP). As shown in (2.1), the aggregation of RFedAGS in Line 14 of Algorithm 1 actually is unbiased in the sense of $\mathbb{E}\left[\sum_{i \in \mathcal{S}_t} \frac{1}{p_i N} \mathrm{grad} f_i(x)\right] = \mathrm{grad} F(x)$. Nevertheless, if the participation probabilities are not considered and the usual aggregation, $x_{t+1} \leftarrow \mathrm{R}_{x_t}\left(-\varpi \sum_{j \in \mathcal{S}_t} \frac{1}{|\mathcal{S}_t|} \zeta_{t,K}^j\right)$, is used, then the output of the algorithm equipped with this aggregation will tend towards a minimizer of another objective function different from the original objective when there exist $i, j \in [N]$ such that $p_i \neq p_j$, which exactly is what Theorem 2.1 points out.

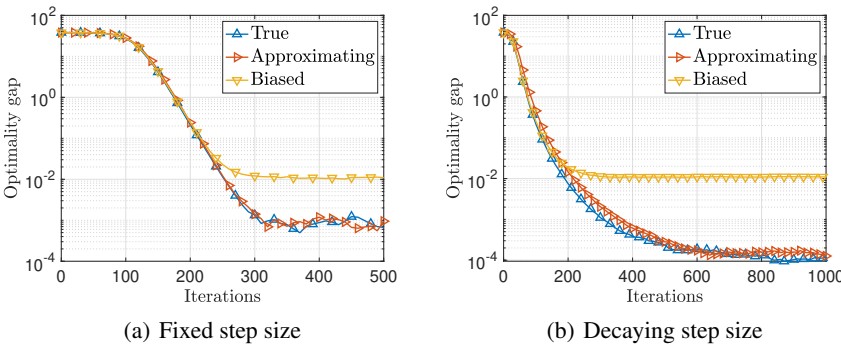

(a) Fixed step size          (b) Decaying step size

Figure 6: PEC with non-I.I.D. (slight) MNIST dataset: comparisons of the two aggregations patterns (AGS-RS) and (AGS-AP).

Figure 6 reports the experiment results, where the two curves "True" and "Approximating" adopt the aggregation pattern (AGS-AP), the curve "Approximating" uses the frequency to estimate the true probability, and the curve "Biased" uses the usual aggregation (AGS-RS). Besides, the participation probabilities $p_i$'s are uniformly and randomly generated (i.e., $p_i$, $i \in [N]$, follows the uniform distribution $\mathrm{U}(0,1)$), the fixed step size is set as $\alpha = 8.0 \times 10^{-5}$, the parameters for decaying steps sizes are set as $(\alpha_0, \beta, \mathrm{d}) = (3.5 \times 10^{-4}, 0.1, 20)$, batch size is $B = 0.5S$, and the number of local updates is set as $K = 5$. It is observed from Figure 6 that RFedAGS equipped with the aggregation pattern (AGS-AP) gives a better solution to Problem (A.1) than that generated by RFedAGS equipped with the usual aggregation pattern (AGS-RS). The reason lies on that the usual aggregation pattern (AGS-RS) leads the iterates to the minimizer of $\tilde{F} := \sum_{i=1}^N \tilde{p}_i f_i$ with $\tilde{p}_i = p_i \int_0^1 \prod_{j \neq i}^N (1 - p_j + p_j t) \mathrm{d}t$, as stated by Theorem 2.1. Meanwhile, due to $p_i \neq p_j$ for some $i, j \in [N]$, it follows that there exists no $\chi > 0$ such that $\tilde{F} = \chi \cdot F$. Hence, the minimizers of $\tilde{F}$ may be not consistent with those of $F$.

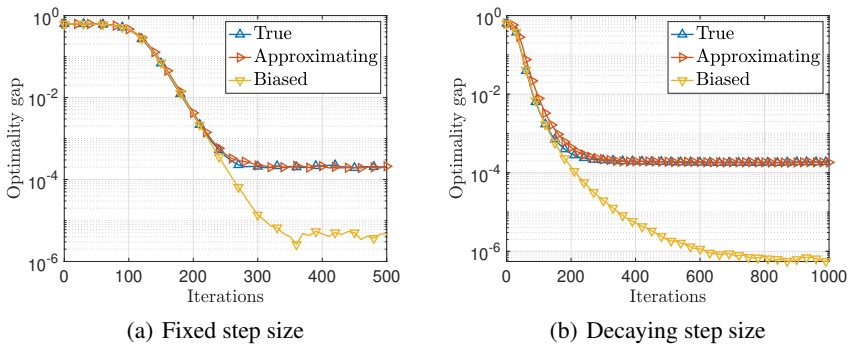

(a) Fixed step size          (b) Decaying step size

Figure 7: PEC with non-I.I.D. (slight) MNIST dataset: RFedAGS with the two aggregations solve the re-weighted problem $\arg\min_{x \in \mathcal{M}} \tilde{F}(x)$.

Furthermore, Figure 7 shows the curves of optimality gap v.s. iterations for the re-weighted objective $\tilde{F}$ valued at the iterates given in Figure 6. Combining Figures 6 and 7, we conclude that RFedAGS equipped with the aggregation pattern (AGS-RS) does solve the re-weighted problem $\arg\min_{x \in \mathcal{M}} \tilde{F}(x)$ rather than the original problem.

### A.1.2 Comparisons of different participation schemes

Here we consider the special case where each agents participates in any round of communication with the same participation probability, i.e., $p_i = p_j$ with $i, j \in [N]$. In this case, the random sampling scheme is denoted by Scheme I, while our arbitrary participation scheme is denoted by Scheme II, where we use frequencies to estimate the true probabilities. For Scheme I, the sampling rate (the ratio of the number of sampled agents to the number of total agents) is as $\rho = 0.3$ (0.5, or 0.7). For Scheme II, the participation probability agent $i$ is respectively set as $p_i = 0.3$ (0.5, or 0.7) for all $i \in [N]$ such that the number of participating agents in Scheme II is equivalent to that of Scheme I in expectation, which means $\sum_{i=1}^{N} p_i = \rho N$. The fixed step size is set $\alpha = 8 \times 10^{-5}$, the parameters for decaying step sizes are set as $(\alpha_0, \beta, d) = (3.5 \times 10^{-4}, 0.1, 20)$, batch size is $B = 0.5S$, and the number of local updates is set as $K = 5$. As demonstrated in Figure 8, the performance of two participation schemes are extremely the same. This indicates that Scheme I can be viewed as a special case of our participation scheme and that using frequencies to estimate the true probabilities is sufficient to ensure convergence.

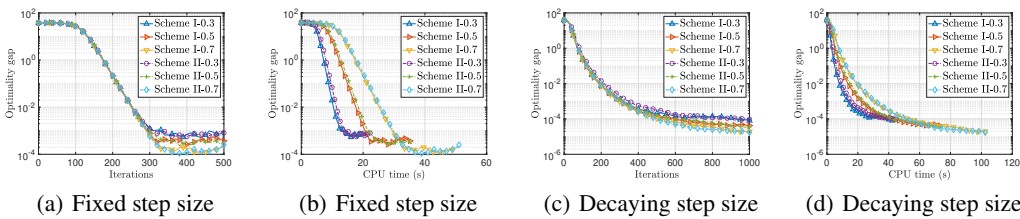

(a) Fixed step size     (b) Fixed step size     (c) Decaying step size     (d) Decaying step size

Figure 8: PEC with non-I.I.D. (slight) MNIST dataset: comparisons of the two participation schemes.

Next, we simulate the scenario of straggling agent participation. Suppose that the first three agents are stragglers and make their local computation time become 10 times as much as that under normal conditions. Specifically, for Scheme I, if one of the three stragglers are chosen, then its local computational time becomes 10 times as much as that under normal conditions; for Scheme II, setting the stragglers' participation probabilities as $0.05$ ensures that they rarely participate in local updates, and when one of the stragglers responds to the server, its local computational also becomes 10 times as much as that under normal conditions. The participation probabilities of the other agents are properly set such that $\sum_{i=1}^{N} p_i \approx \rho N$. The fixed step size is set $\alpha = 8 \times 10^{-5}$, the parameters for decaying step sizes are set as $(\alpha_0, \beta, d) = (2.8 \times 10^{-4}, 0.1, 20)$, batch size is $B = 0.5S$, and the number of local updates is set as $K = 5$. The experiment results are shown in Figure 9.

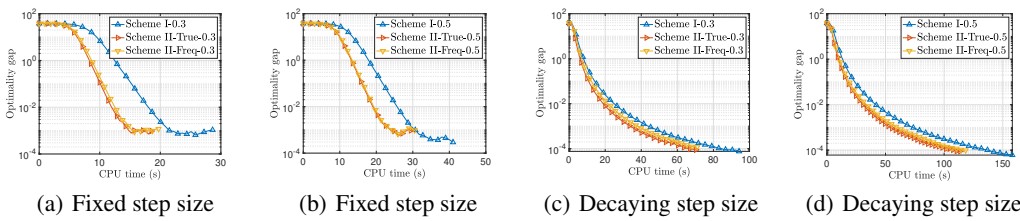

(a) Fixed step size     (b) Fixed step size     (c) Decaying step size     (d) Decaying step size

Figure 9: PEC with non-I.I.D. (slight) MNIST dataset: the situation where the FL system has three stragglers. Here in the legends, Scheme-II-True (or Scheme-II-Freq) means that the Scheme II is equipped with the true probabilities (or frequencies serving as the true probabilities).

By the definition of Scheme I, each agent is sampled with probability $\rho$ (e.g., 0.3 and 0.5 in our experiments), which is much greater than 0.05 in Scheme II for the three stragglers. Hence, the number of stragglers participating local updates of Scheme I is greater than the one of Scheme II, leading to the CPU time of Scheme I are greater than the one of Scheme II. The results in Figure 9 is consistent with our analysis. Meanwhile we note that the performance of using the true probabilities and frequencies is extremely the same, which indicates again the validity of using frequencies serving as the true probabilities.

It should be noted that in a practical situation, if some agents do not respond to the server in a certain round of communication, then scheme I may not work in this case, because one of these agents may be sampled by the server, but it will not respond to the server. This will cause the algorithm to stagnate. Nevertheless, Scheme II does not encounter this issue since the server does not choose the agents which do not respond.

### A.1.3 Influence of the level of data heterogeneity on performance

Next we test the impact of the heterogeneity level of the MNIST dataset on the performance of RFedAGS. Here the participation probabilities $p_i$'s are uniformly and randomly generated, that is, $p_i \sim \mathrm{U}(0,1)$ for $i \in [N]$. The fixed step size is set $\alpha = 8 \times 10^{-5}$, the parameters for decaying step sizes are set as $(\alpha_0, \beta, \mathrm{d}) = (2.8 \times 10^{-4}, 0.1, 20)$, batch size is $B = 0.5S$, and the number of local updates is set as $K = 5$. The experiment results are reported in Figure 10, where we observe that the quality of the solution generated by Algorithm 1 gets worse as the growth of the levels of heterogeneity of the training data across agents. Additionally, Theorems 3.1 and 3.2 point out that if decaying step sizes satisfying equation 3.1 are used, Algorithm 1 has global convergence. Hence, it is expected that the higher-quality solutions may be found when using decaying step sizes and running more rounds of communication compared with the case using a fixed step size. This is consistent with the experiment results as shown in Figure 8-Figure 10.

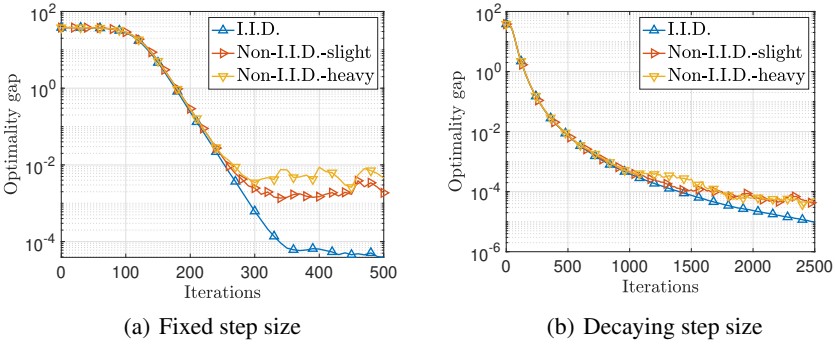

(a) Fixed step size            (b) Decaying step size

Figure 10: PEC with different non-I.I.D. datasets: impact of heterogeneity level.

### A.1.4 Effect of local multiple-step update

In addition, we test the impact of different number of local updates $K$ on the performance of Algorithm 1. The participation probabilities $p_i$'s are uniformly and randomly generated, that is, $p_i \sim \mathrm{U}(0,1)$ for $i \in [N]$. The fixed step size is set $\alpha = 8 \times 10^{-5}$ and batch size is $B = 0.5S$. The experiment results are shown in Figure 11.

When using a fixed step size, Item 1 of Theorem 3.4 states that the convergence upper bound consists of two terms: a decaying term $\frac{2\Theta(x_1)}{\varpi \alpha K T}$ as $K$ (or $T$) increases, and a increasing (or constant) term $2\alpha Q(K, B, \alpha, \varpi)$ with respect to $K$ (or $T$). The initial guess $x_1$ is usually generated at random such that $\Theta(x_1)$ is relatively large, and thus the first term dominates at the initial stage. As a result, at the initial stage, the convergence speed is accelerated when using larger $K$. Subsequently, due to the growth of $T$, the second term begins to dominate and thus when using larger $K$ the error of solution generated by Algorithm 1 to the minimizer get larger. This analysis is verified by Figure 11. Additionally, we note that in fixed step size cases, Algorithm 1 numerically demonstrates linear convergence as seen in Figures 6-11.

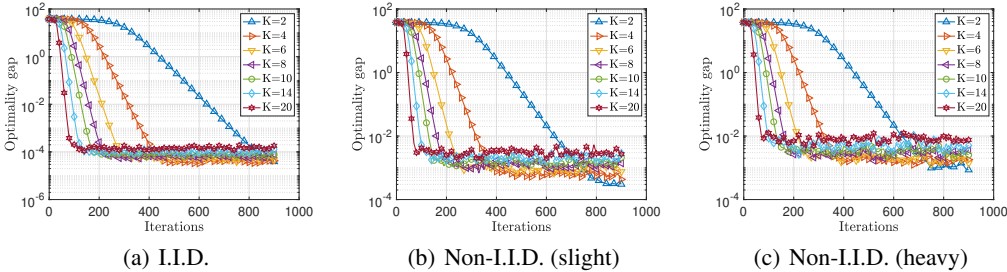

|(a) I.I.D.|(b) Non-I.I.D. (slight)|(c) Non-I.I.D. (heavy)|

Figure 11: PEC with non-I.I.D. (slight) MNIST dataset: impact of number of local updates.

### A.1.5 COMPARISONS WITH EXISTING RIEMANNIAN FL ALGORITHMS

Here we test the performance of RFedAGS, RFedAvg, RFedSVRG, and RFedProj as (i) data distributions diverge or (ii) participation becomes sparse.

(i) for the first purpose, we use the MNIST dataset partitioned as three different levels of heterogeneity; see Figure 5 for the sample distributions. Figures 12(a)-12(b) show the results, where the participation probabilities $p_i$'s are uniformly and randomly generated (i.e., $p_i \sim \mathcal{U}(0,1)$ for $i \in [N]$), the fixed step size is set as $\alpha = 8 \times 10^{-5}$, and batch size is $B = 0.5S$. We can observe from Figure 12 that as expected, for all of algorithms, as data distributions diverge, the performance becomes poor. Besides, at the same level of data heterogeneity, RFedAGS consistently outperforms compared to the other algorithms.

(ii) for the second purpose, we use the non-I.I.D. (slight) MNIST dataset (see Figure 5(b)). The experimental results are reported in Figures 12(c)-12(d), where $0.5, 0.4, 0.3$ in the legends denote the expected participation ratios, i.e., $\frac{1}{N}\sum_{i=1}^{N}\mathbb{E}[p_i] = 0.5, 0.4, 0.3$. Specifically, for participation ratio $0.5$, we set the participation probabilities as $p_i \sim \mathrm{U}(0,1)$; next, for participation ratio $0.4$ (or, $0.3$), we let $p_i' = 0.8 \times p_i$ (or, $p_i' = 0.6 \times p_i$). It follows from Figures 12(c)-12(d) that as participation becomes sparse, the performance of all algorithms becomes poor. On the other hand, at the same participation ratio, our RFedAGS consistently performs compared to other algorithms.

### A.2 COMPARISONS WITH SOME CENTRALIZED ALGORITHMS

Low-rank matrix completion (LRMC) aims to recover the missing entries of an unknown matrix from a small account of accesible entries with low-rank constraint for the matrix. Mishra et al. (Mishra et al., 2019) formulate LRMC in the form of finite sum, which can be extended to the FL setting with finite sum minimization as follows:

$$
\min_{\mathcal{U} \in \mathrm{Gr}(r,m)} F(\mathcal{U}) := \frac{1}{N}\sum_{i=1}^{N} f_i(\mathcal{U}), \text{with } f_i(\mathcal{U}) = \frac{1}{S}\sum_{j=1}^{S} 0.5\|\mathcal{P}_{\Omega_{ij}}(\mathbf{U}\mathbf{W}_{ij\mathbf{U}}^T) - \mathcal{P}_{\Omega_{ij}}(\mathbf{Y}_{ij}^*)\|_F^2
$$
$$
+ \lambda\|\mathbf{U}\mathbf{W}_{ij\mathbf{U}}^T - \mathcal{P}_{\Omega_{ij}}(\mathbf{U}\mathbf{W}_{ij\mathbf{U}}^T)\|_F^2
$$
(A.2)

where $\mathrm{Gr}(r,m)$ is the Grassmann manifold, i.e, the set of all the $r$-dimension subspaces of $\mathbb{R}^m$, $\mathbf{U} \in \mathrm{St}(r,m)$ is the matrix characterization of $\mathcal{U} \in \mathrm{Gr}(r,m)$, $\mathbf{W}_{ij\mathbf{U}} \in \mathbb{R}^{n_{ij} \times r}$ with $\sum_{i=1}^{N}\sum_{j=1}^{S} n_{ij} = n$ is the least-squares solution to $\arg\min_{\mathbf{W}_{ij} \in \mathbb{R}^{n_{ij} \times r}} 0.5\|\mathcal{P}_{\Omega_{ij}}(\mathbf{U}\mathbf{W}_{ij}^T) - \mathcal{P}_{\Omega_{ij}}(\mathbf{Y}_{ij}^*)\|_F^2 + \lambda\|\mathbf{U}\mathbf{W}_{ij}^T - \mathcal{P}_{\Omega_{ij}}(\mathbf{U}\mathbf{W}_{ij}^T)\|_F^2$, $\mathbf{Y}^* \in \mathbb{R}^{m \times n}$ is the known matrix and is partitioned into $\mathbf{Y}^* = [\mathbf{Y}_{1,1}^*, \ldots, \mathbf{Y}_{1,S}^*, \ldots, \mathbf{Y}_{N,1}^*, \ldots, \mathbf{Y}_{N,S}^*]$ with $\mathbf{Y}_{ij}^* \in \mathbb{R}^{m \times n_{ij}}$, $\Omega$ is the indices set of elements of $\mathbf{Y}^*$: the $(l,k)$-element of $\mathbf{Y}^*$ is nonzero if and only if its index belongs to $\Omega$ and is also partitioned similar to the way of $\mathbf{Y}$: $\Omega = \{\Omega_{1,1}, \ldots, \Omega_{1,S}, \ldots, \Omega_{N,1}, \ldots, \Omega_{N,S}\}$, and operator $\mathcal{P}_{\Omega_{ij}}$ is the orthogonal sampling operator defined by $[\mathcal{P}_{\Omega_{ij}}(\mathbf{Y})]_{lk} =$ the $(l,k)$-element of $\mathbf{Y}$ if $(l,k) \in \Omega_{i,j}$ and $[\mathcal{P}_{\Omega_{ij}}(\mathbf{Y})]_{lk} = 0$ otherwise. It is worthy mentioned that Problem (A.2) is defined on $\mathrm{Gr}(r,m)$ but the computation can be implemented with matrices $\mathbf{U}$ in $\mathrm{St}(r,m)$. The over-sampling ratio (OS) is the ratio of number of entries of $\Omega$ and the freedom degree of $\mathbf{Y}^*$, i.e., $OS = |\Omega|/((m+n-r)r)$.

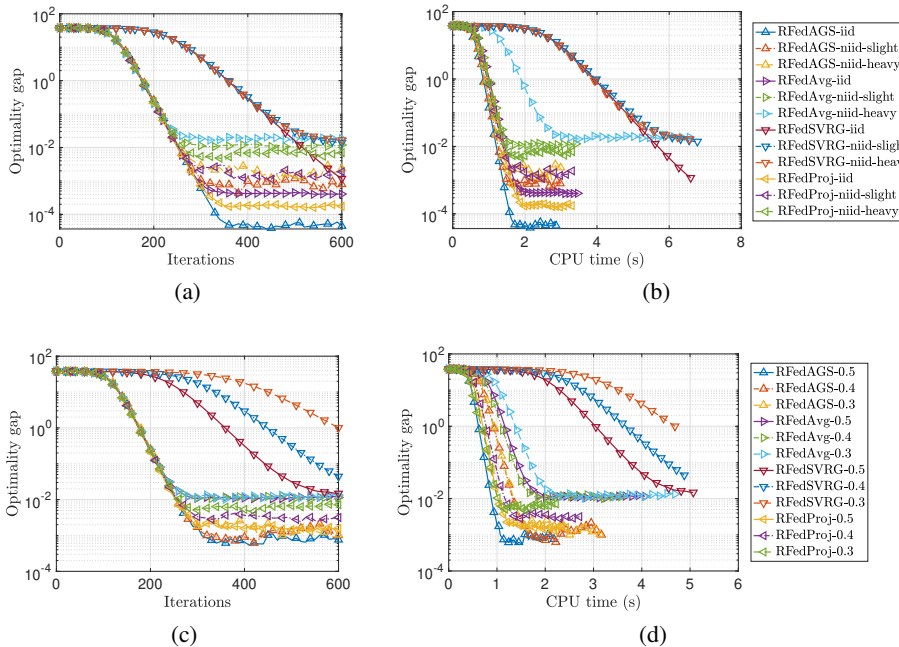

Figure 12: PEC with the MNIST dataset. (a)-(b): performance as data distributions diverge. (c)-(d): performance as participation becomes sparse.

The Grasssmann manifold $\mathrm{Gr}(r, m)$ is equipped with the quotient structure $\mathrm{Gr}(r, m) = \mathrm{St}(r, m)/\mathrm{O}(r) = \{[\mathbf{U}] : \mathbf{U} \in \mathrm{St}(r, m)\}$ with $\mathrm{O}(r)$ the orthogonal group of the order $r$. The Riemannian metric on $\mathrm{Gr}(r, m)$ is induced by the inner product, i.e., $\langle \eta_{\mathcal{U}}, \xi_{\mathcal{U}} \rangle_{\mathcal{U}} = \mathrm{trace}(\eta_{\mathcal{U}_\uparrow}^T \xi_{\mathcal{U}_\uparrow})$, where $\xi_{\mathcal{U}_\uparrow}$ is the horizontal lift of $\xi_{\mathcal{U}}$. The retraction via Cayley transform (CT) (Zhu & Sato, 2021) is given by

$$\mathrm{R}_{\mathcal{U}}^{\mathrm{Cay}}(\xi_{\mathcal{U}}) = \left[ \mathbf{U} + \xi_{\mathcal{U}_\uparrow} - \left( \frac{1}{2}\mathbf{U} + \frac{1}{4}\xi_{\mathcal{U}_\uparrow} \right) \left( I_r + \frac{1}{4}\xi_{\mathcal{U}_\uparrow}^T \xi_{\mathcal{U}_\uparrow} \right)^{-1} \xi_{\mathcal{U}_\uparrow}^T \xi_{\mathcal{U}_\uparrow} \right],$$

and the inverse of $\mathrm{R}^{\mathrm{Cay}}$ (Zhu & Sato, 2021) is computed by

$$\left( \left( \mathrm{R}_{\mathcal{U}}^{\mathrm{Cay}} \right)^{-1} (\mathcal{V}) \right)_{\mathcal{U}_\uparrow} = 2(\mathbf{V} - \mathbf{U}\mathbf{U}^T\mathbf{V})(I_r + \mathbf{U}^T\mathbf{V})^{-1}.$$

Correspondingly, the isometric vector transport associated with $\mathrm{R}^{\mathrm{Cay}}$ (Zhu & Sato, 2021) is given by

$$\left( \mathcal{T}_{\eta_{\mathcal{U}}}^{\mathrm{Cay}}(\xi_{\mathcal{U}}) \right)_{\mathcal{V}_\uparrow} = \xi_{\mathcal{U}_\uparrow} - \left( \mathbf{U} + \frac{1}{2}\eta_{\mathcal{U}_\uparrow} \right) \left( I_r + \frac{1}{4}\eta_{\mathcal{U}_\uparrow}^T \eta_{\mathcal{U}_\uparrow} \right)^{-1} \eta_{\mathcal{U}_\uparrow}^T \xi_{\mathcal{U}_\uparrow}$$

with $\mathcal{V} = \mathrm{R}_{\mathcal{U}}^{\mathrm{Cay}}(\eta_{\mathcal{U}})$. We point out that Algorithm 1 does not require the usage of the inverse of retraction. Here, what we use the inverse of retraction is just to assist in the implementation of the vector transport. Moreover, if one uses the vector transport by projection, then the inverse of retraction does not need.

### A.2.1 SYNTHETIC CASE

Sample at random two matrices $\mathbf{A} \in \mathbb{R}^{m \times r}$ and $\mathbf{B} \in \mathbb{R}^{n \times r}$. Let $\mathbf{Y}^* = \mathbf{A}\mathbf{B}^T$. $mn - |\Omega|$ entries are randomly removed with uniform probability. Each of the rest entries is perturbed by noise obeying the Gaussian distribution with mean zero and standard deviation $10^{-6}$. In the experiment, the rank is set as $r = 5$, the OS is set as $OS = 6$, and $(m, n) = (100, 2000)$. The other parameters are set as $\lambda = 0, (N, S) = (20, 100), p_i \sim \mathrm{U}(0, 1), \forall i \in [N], B = 0.5S$, and $\alpha = 2 \times 10^{-3}$,

Let $\tilde{\mathbf{U}}$ be the solution given by Algorithm 1. Then $\mathbf{W}_{\tilde{\mathbf{U}}} = [\mathbf{W}_{11\tilde{\mathbf{U}}}, \ldots, \mathbf{W}_{1S\tilde{\mathbf{U}}}, \ldots, \mathbf{W}_{N1\tilde{\mathbf{U}}}, \ldots, \mathbf{W}_{NS\tilde{\mathbf{U}}}]$, and thus the approximation to $\mathbf{Y}^*$ is given by $\tilde{\mathbf{Y}} = \tilde{\mathbf{U}}\mathbf{W}_{\tilde{\mathbf{U}}}^T$. Relative error (lower is better) between $\tilde{\mathbf{Y}}$ and $\mathbf{Y}^*$, computed by

$$\text{rel\_err}(\tilde{\mathbf{Y}}) = \frac{\|\tilde{\mathbf{Y}} - \mathbf{Y}^*\|_F}{\|\mathbf{Y}^*\|_F},$$

is used to measure the performance of Algorithm 1. From Figure 13, we also observe a similar result: the number of inner iterations significantly affects the convergence. It is worth mentioning that the results demonstrate Algorithm 1 has a linear convergence rate.

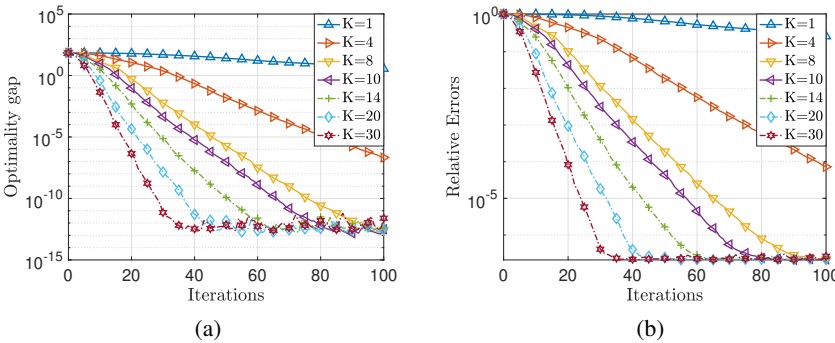

Figure 13: LRMC with synthetic data: performance of RFedAGS with different $K$.

### A.2.2 A REAL-WORLD APPLICATION

We use MovieLens 1M [7] dataset which consists of 1000209 ratings with 6040 users rating 3952 movies. In LRMC setting, $\mathbf{Y}^* \in \mathbb{R}^{m \times n}$, with $m = 3952$, $n = 6040$, and $|\Omega| = 1000209$, whose nonzero elements are the ratings. We randomly sample $80\%$ ratings for each column of $\mathbf{Y}^*$ as the training samples, denoted by $\mathbf{Y}^{\text{tr}}$, and the testing dataset, denoted by $\mathbf{Y}^{\text{te}}$, is consisted of the remainder. In terms of the FL setting, $\mathbf{Y}^{\text{tr}}$ is equally divided into $N = 40$ agents by column at order, i.e., $\mathbf{Y}^{\text{tr}} = [\mathbf{Y}_1^{\text{tr}}, \ldots, \mathbf{Y}_N^{\text{tr}}]$, and each agent has $S = 151$ columns, i.e., $\mathbf{Y}_i^{\text{tr}} = [\mathbf{Y}_{i,1}^{\text{tr}}, \ldots, \mathbf{Y}_{i,S}^{\text{tr}}]$ where $\mathbf{Y}_{i,j}^{\text{tr}} \in \mathbb{R}^m$. The other parameters are set as $\lambda = 10^{-2}$, $p_i \sim \text{U}(0,1), \forall i \in [N]$, $B = 0.5S$, and $\alpha = 6 \times 10^{-4}$.

In order to evaluate the performance of those methods, the root mean square error (RMSE) is used and is computed by

$$\text{RMSE}(\tilde{\mathbf{Y}}) = \sqrt{\frac{1}{|\Omega^{\text{te}}|} \sum_{(i,j) \in \Omega^{\text{te}}} |\tilde{\mathbf{Y}}_{ij} - \mathbf{Y}_{ij}^{\text{te}}|^2}$$

with $\tilde{\mathbf{Y}}$, $\mathbf{Y}^{\text{te}}$, and $\Omega^{\text{te}}$ being the approximation to $\mathbf{Y}^{\text{te}}$, the testing matrix, and the indices set of known entries of $\mathbf{Y}^{\text{te}}$, respectively. We observe in Figure 14 and Table 3 that the proposed RFedAGS is comparable to these centralized methods in solving LRMC in terms of RMSE when choosing an appropriate $K$.

---

[7]See https://grouplens.org/datasets/movielens/1m/.

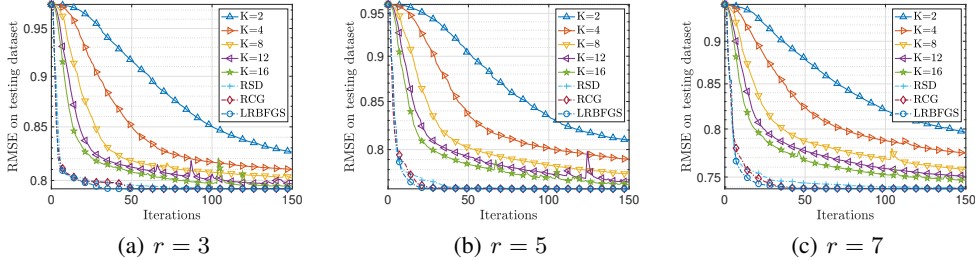

Figure 14: LRMC with MovieLens 1M dataset: comparisons of RFedAGS (with different $K$) with RSD, RCG, and LRBFGS.

Table 3: The best RMSE scores (lower is better) on testing set for different subspace dimension $r$ and different number of local update $K$. Here the scalar $a.bcd_k$ denotes $a.bcd \times 10^k$.

| | RFedAGS | | | | | RSD | RCG | LRBFGS |
|---|---|---|---|---|---|---|---|---|
| | $K = 2$ | $K = 4$ | $K = 8$ | $K = 12$ | $K = 16$ | | | |
| $r = 3$ | $8.260_{-1}$ | $8.101_{-1}$ | $8.023_{-1}$ | $7.968_{-1}$ | $7.948_{-1}$ | $7.925_{-1}$ | $7.925_{-1}$ | $7.925_{-1}$ |
| $r = 5$ | $8.095_{-1}$ | $7.902_{-1}$ | $7.757_{-1}$ | $7.679_{-1}$ | $7.654_{-1}$ | $7.616_{-1}$ | $7.614_{-1}$ | $7.614_{-1}$ |
| $r = 7$ | $7.966_{-1}$ | $7.743_{-1}$ | $7.577_{-1}$ | $7.507_{-1}$ | $7.468_{-1}$ | $7.392_{-1}$ | $7.384_{-1}$ | $7.382_{-1}$ |

### A.3 THE DETAILS OF EXPERIMENT SETTINGS IN SECTION 4

In this section, we detail the experiment settings in Section 4.

#### A.3.1 PCA.

We restate the PCA problem as follows for convenience:

$$\min_{X \in \mathrm{St}(r,d)} F(X) := \frac{1}{N} \sum_{i=1}^{N} f_i(X), \quad \text{with } f_i(X) = -\frac{1}{S} \sum_{j=1}^{S} \mathrm{tr}(X^T (Z_{ij} Z_{ij}^T) X), \tag{A.3}$$

where $\mathrm{St}(r,d) = \{X \in \mathbb{R}^{d \times r} : X^T X = I_r\}$ is the Stiefel manifold, $\mathcal{D}_i = \{Z_{i1}, \ldots, Z_{iS}\} \subseteq \mathbb{R}^{d \times p}$ is the local dataset held by agent $i$, $\forall i \in [N]$.

For the Stiefel manifold $\mathrm{St}(r,d)$, we view it as a Riemannian manifold embedded in $\mathbb{R}^{d \times r}$. Thus the Riemannian metric is chosen as $\langle U, V \rangle_X = \langle U, V \rangle_{\mathrm{F}}$ for all $X \in \mathrm{St}(r,d)$ and $U, V \in \mathrm{T}_X \mathrm{St}(r,d)$. The retraction is the qr-retraction (Absil et al., 2008) and the vector transport is given via the projection, i.e., $\mathcal{T}_V U = \mathcal{P}_{\mathrm{R}_X^{\mathrm{qr}}(V)}(U)$. In theory, RFedAvg and RFedSVRG (Li & Ma, 2023) require the exponential mapping, its inverse, and parallel transport. But on the Stiefel manifold, the last two operators have no closed-form expressions. Thus we use retraction, its inverse, and vector transport to replace them.

**Setup details corresponding to Figures 2(a)-2(b).** For the synthetic data, we set $p = 1$ and generate the local datesets by setting $[Z_{i1}, \ldots, Z_{iS}] = Z_i$ drawn from the Gaussian distribution $Z_i \sim \mathcal{N}(0, \frac{i}{N})$. In experiment, all parameters are set as $(r, d) = (5, 100)$, $(N, S) = (40, 100)$, $\alpha = 6 \times 10^{-3}$, $B = 0.5S$, $K = 5$, and $p_i \sim \mathrm{U}(0, 1)$.

**Setup details corresponding to Figures 2(c)-2(d).** For CIFAR10 dataset, whose training dataset contains 50000 RGB images with size $32 \times 32$ of each channel, it is also shuffled following the way of McMahan et al. (2017) such that the local datasets are non-I.I.D. (see Figure 15 below). In experiment, we flatten each image into a vector in $\mathbb{R}^{3072}$, and thus each local data point $Z_{ij}$ is inside $\mathbb{R}^{3072}$. The other parameters are set as $(r, d) = (4, 3072)$, $(N, S) = (50, 1000)$, $\alpha = 3 \times 10^{-5}$, $B = 0.5S$, $K = 5$, and $p_i \sim \mathrm{U}(0, 1)$.

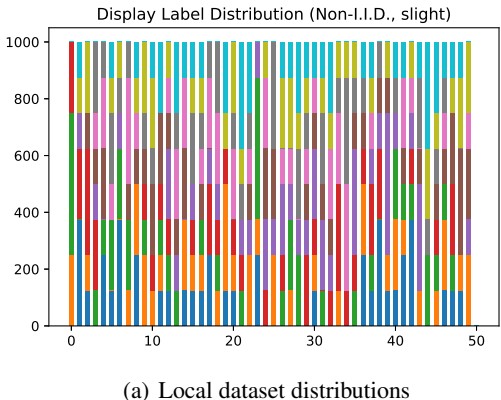

(a) Local dataset distributions

Figure 15: Local dataset distributions of the CIFAR10 dataset

**Scalability of RFedAGS on PCA.** Here we conduct additional experiments to empirically explore the scalability of RFedAGS. The results are reported in Figure 16, where the local update step is set as $K = 5$, batch size is $B = 0.5S$. In the first column, we fix the local dataset size and the manifold dimension and enlarge the number of agents. In the second column, we enlarge local dataset size and fix the other two factors. In the last column, we enlarge the manifold dimension and fix the other two factors. In summary, it can be observed from Figure 16 that the RFedAGS can all solve these problems of such scale, showing the scalability of RFedAGS. We would like to point out that as shown in Table 2, number of agents, local dataset size, and manifold dimension have a linear relationship with the total computation complexity, so their increase will not cause the total computation complexity to increase sharply.

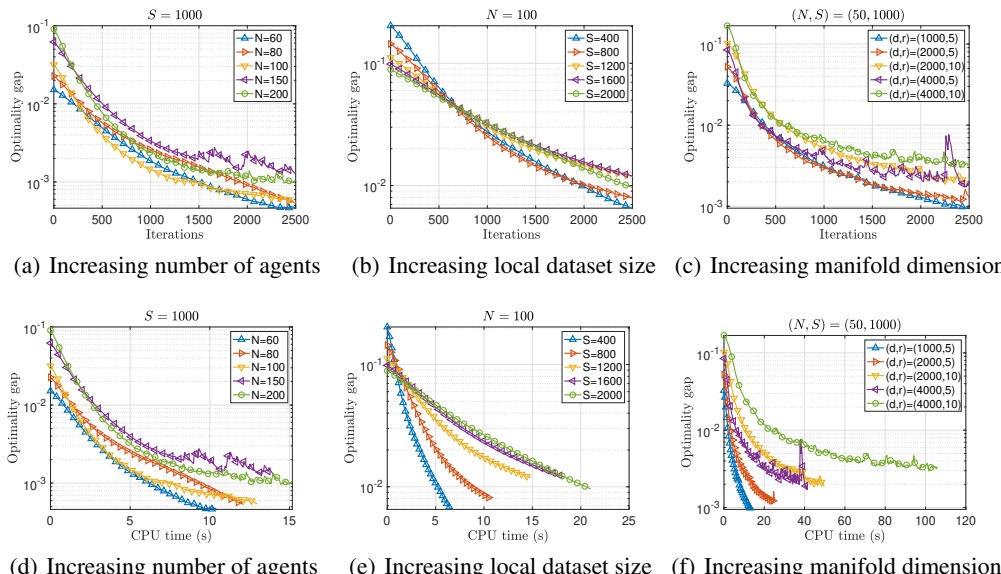

(a) Increasing number of agents    (b) Increasing local dataset size    (c) Increasing manifold dimension

(d) Increasing number of agents    (e) Increasing local dataset size    (f) Increasing manifold dimension

Figure 16: PCA with synthetic data: scalability of RFedAGS.

### A.3.2  HSP.

Given a set of training pairs $\mathcal{D} = \{\mathcal{D}_i\}_{i=1}^N = \{\{(w_{i,j}, y_{i,j})\}_{j=1}^S\}_{i=1}^N$, where $w_{i,j} \in \mathbb{R}^r$ is the feature and $y_{i,j} \in \mathcal{H}^d$ is the hyperbolic embedding of the class of $w_{i,j}$. Then for a test sample $w$, the task of

HPS is to predict its hyperbolic embeddings by solving the following problem

$$\arg\min_{x \in \mathcal{H}^d} F(x) := \frac{1}{N} \sum_{i=1}^N f_i(x), \text{ with } f_i(x) = \frac{1}{S} \sum_{j=1}^S a_{i,j}(\omega) \text{dist}^2(x, y_{i,j})$$

where the hyperbolic manifold $\mathcal{H}^d$ is characterized via the Lorentz hyperbolic model $\mathcal{H}^d := \{x \in \mathbb{R}^{d+1} : \langle x, x \rangle_{\mathcal{L}} = -1\}$ with $\langle x, y \rangle_{\mathcal{L}} = x^T y - 2x_1 y_1$, $a_1(w)^T = (a_{i,1}(w), \ldots, a_{i,S}(w))^T \in \mathbb{R}^S$ is a pre-given constant vector related to $w$, and $\text{dist}(\cdot, \cdot) : \mathcal{M} \times \mathcal{M} \to \mathbb{R}$ is the Riemannian distance. A commonly used option of $a_i(w)$ is computed by $a_i(w) = (K_i + \gamma I)^{-1} K_{i,w}$, where $\gamma$ is the regularization parameter, and $K_i \in \mathbb{R}^{S \times S}$ and $K_{i,w} \in \mathbb{R}^S$ are given by $(K_i)_{l,h} = k(w_{i,l}, w_{i,h})$ and $(K_{i,w})_j = k(w_{i,j}, w)$ for a raial basis function (RBF) kernel $k(w, w') = \exp(-\|w - w'\|_2^2/(2\nu)^2)$ with a constant $\nu > 0$.

**Setup details corresponding to Figure 3.** The WordNet dataset (Miller, 1995) is used to conduct the experiment of inferring hyperbolic embeddings (Nickel & Kiela, 2018; Jawanpuria et al., 2019b). Following (Nickel & Kiela, 2017), the pretrained hyperbolic embeddings on $\mathcal{H}^2$ of the mammals subtree with the transitive closure containing $n = 1180$ nodes (words) and 6540 edges (hierarchies) are used.[8] The features are stemmed from Laplacian eigenmap (Belkin & Niyogi, 2003) to dimension $r = 3$ of the adjacency matrix formed by the edges. In other words, we obtained $\{(w_i, y_i)\}_{i=1}^n \subset \mathbb{R}^3 \times \mathcal{H}^2$. This setting is in line with the work in (Han et al., 2024b). In the experiments, the word "primate" is selected as the test sample, and the remainder is used to train. Therefore, the hyperbolic embedding of the word "primate" is known and is viewed as the true embedding, i.e., $x_{\text{true}}$. For other parameters, they are set as $(N, S) = (9, 131)$, $\alpha = 6 \times 10^{-2}$, $B = 0.5S$, $K = 5$, $p_i \sim \text{U}(0, 1)$, and $(\gamma, \nu) = (10^{-5}, 0.3)$.

### A.3.3 FMC.

Given a set of training SPD matrices $\mathcal{D} := \{\mathcal{D}_i\}_{i=1}^N = \{\{X_{i,j}\}_{j=1}^S\}_{i=1}^N$, where $\{X_{i,j}\}_{j=1}^S \subseteq \mathcal{S}_{++}^N := \{X \in \mathbb{R}^{N \times N} : X^T = X, X \succ 0\}$, the FMC of these SPD matrices is the solution to the following problem

$$\arg\min_{X \in \mathcal{S}_{++}^N} F(X) := \frac{1}{N} \sum_{i=1}^N f_i(X), \text{ with } f_i(X) = \frac{1}{S} \sum_{j=1}^S \text{dist}^2(X, X_{i,j}),$$

where $\text{dist}(X, Y) = \|\text{logm}(X^{-1/2} X_{i,j} X^{-1/2})\|_F$ with $\text{logm}(\cdot)$ the principal matrix logarithm is the Riemannian distance.

**Setup details corresponding to Figures 4.** The PATHMNIST dataset (Yang et al., 2023) consists of 89996 RGB images and we transform each image into a $9 \times 9$ SPD matrix by the covariance descriptor (Tuzel et al., 2006). In the experiment, we randomly selects 20000 images to construct the training dataset. The parameters are set as $(N, S) = (50, 400)$, $\alpha = 0.01$, $B = 0.5S$, $K = 5$, and $p_i \sim \text{U}(0, 1)$.

## B PRELIMINARIES ON RIEMANNIAN OPTIMIZATION

In this section, we briefly review the basic ingredients for Riemannian optimization, which are drawn from the standard literature, e.g., (Boothby, 1975; Absil et al., 2008). Let $\mathcal{M}$ be a $d$-dimensional Riemannian manifold equipped with a Riemannian metric $\langle \cdot, \cdot \rangle : (\eta_x, \zeta_x) \mapsto \langle \eta_x, \zeta_x \rangle_x \in \mathbb{R}$ for any $x \in \mathcal{M}, \eta_x, \zeta_x \in T_x\mathcal{M}$ (when it is clear in the context, we omit the subscript and write $\langle \eta, \zeta \rangle$ for short). For all $x \in \mathcal{M}$, the tangent space $T_x\mathcal{M}$ is a $d$-dimensional linear space. The norm induced by the Riemannian metric in the tangent space $T_x\mathcal{M}$ is $\|\eta\| = \sqrt{\langle \eta, \eta \rangle}$ for all $\eta \in T_x\mathcal{M}$. An open ball centered at $\eta \in T_x\mathcal{M}$ with radius $r$ in $T_x\mathcal{M}$ is denoted by $\mathbb{B}(\eta, r) = \{\zeta \in T_x\mathcal{M} : \|\zeta - \eta\| < r\}$. The union of all tangent spaces is tangent bundle, denoted by $T\mathcal{M}$. A vector field is a mapping which maps from $\mathcal{M}$ to $T\mathcal{M}$, formally defined by $\eta : \mathcal{M} \to T\mathcal{M} : x \mapsto \eta_x \in T_x\mathcal{M}$. Given a differentiable function $f : \mathcal{M} \to \mathbb{R}$, the Riemannian gradient of $f$, denoted by $\text{grad} f$, is a vector

---

[8]It is referred to website https://github.com/facebookresearch/poincare-embeddings.

field such that for any $x \in \mathcal{M}$, $\mathrm{grad} f(x)$ is the unique vector satisfying $\mathrm{D} f(x)[\eta] = \langle \mathrm{grad} f(x), \eta \rangle$ for any $\eta \in \mathrm{T}_x \mathcal{M}$, where $\mathrm{D} f(x)[\eta]$ is the directional derivative of $f$ at $x$ along $\eta$.

A critical concept in Riemannian optimization is retraction, which defines a smooth mapping, denoted by $\mathrm{R}$, from the tangent bundle to the manifold, i.e., $\mathrm{R} : \mathrm{T}\mathcal{M} \to \mathcal{M}$, satisfying

1. $\mathrm{R}(0_x) = x$ for all $x \in \mathcal{M}$, where $0_x$ is the origin of $\mathrm{T}_x \mathcal{M}$;
2. $\mathrm{DR}(0_x)[\eta] = \eta$ for all $\eta \in \mathrm{T}_x \mathcal{M}$, which implies that $\mathrm{DR}(0_x) = \mathrm{id}_{\mathrm{T}_x \mathcal{M}}$ with $\mathrm{id}_{\mathrm{T}_x \mathcal{M}}$ being the identity in $\mathrm{T}_x \mathcal{M}$.

When restricted to $\mathrm{T}_x \mathcal{M}$, we denote $\mathrm{R}$ by $\mathrm{R}_x$, i.e., $\mathrm{R}_x = \mathrm{R} \mid_{\mathrm{T}_x \mathcal{M}}$. Note that the domain of $\mathrm{R}$ does not need to be the whole tangent bundle. In practice, it is usually the case. In this paper, we always assume that $\mathrm{R}$ is well-defined whenever needed. A special retraction is the exponential mapping, dented by $\mathrm{Exp}$, satisfying $\mathrm{Exp}_x(\eta_x) = \gamma(1)$ where $\gamma$ is the geodesic such that $\gamma(0) = x$ and $\gamma'(0) = \eta_x$. Geodesic is the generalization of straight line in the Euclidean setting to the Riemannian setting, and naturally the exponential mapping is the generalization of addition to the Riemannian setting. Additionally, retraction is a first-order approximation to the exponential mapping. A $r$-totally retractive set $\mathcal{W}$ is a subset of $\mathcal{M}$ such that for any $y \in \mathcal{W}$, it holds that $\mathcal{W} \subseteq \mathrm{R}_y(\mathbb{B}(0_y, r))$ and $\mathrm{R}_y$ is a diffeomorphism on $\mathbb{B}(0_y, r)$. Hence, $\mathrm{R}_x^{-1}(y)$ is well-defined, whenever $x, y \in \mathcal{W}$.

For our RFedAGS, another essential concept is vector transport, denoted by $\mathcal{T}$, which is usually associated with a retraction $\mathrm{R}$. Given a retraction $\mathrm{R}$, a vector transport associated with $\mathrm{R}$ maps from $\mathrm{T}\mathcal{M} \oplus \mathrm{T}\mathcal{M}$, the Whitney sum, to $\mathrm{T}\mathcal{M}$, i.e., $\mathcal{T} : \mathrm{T}\mathcal{M} \oplus \mathrm{T}\mathcal{M} \to \mathrm{T}\mathcal{M}$, and satisfies that for any $(x, \eta_x) \in \mathrm{domain}(\mathrm{R})$ and all $\zeta_x \in \mathrm{T}_x \mathcal{M}$, the followings hold that

1. $\mathcal{T}_{\eta_x}(\zeta_x) \in \mathrm{T}_{\mathrm{R}(\eta_x)}\mathcal{M}$;
2. $\mathcal{T}_{0_x}\zeta_x = \zeta_x$;
3. $\mathcal{T}_{\eta_x}$ is linear, i.e., for all $a_1, a_2 \in \mathbb{R}$ and $\xi_x, \zeta_x \in \mathrm{T}_x \mathcal{M}$, it holds that $\mathcal{T}_{\eta_x}(a_1 \xi_x + a_2 \zeta_x) = a_1 \mathcal{T}_{\eta_x}(\xi_x) + a_2 \mathcal{T}_{\eta_x}(\zeta_x)$.

We say $\mathcal{T}$ is isometric if for any $(x, \eta_x) \in \mathrm{domain}(\mathrm{R})$, $\xi_x, \zeta_x \in \mathrm{T}_x \mathcal{M}$, it satisfies $\langle \mathcal{T}_{\eta_x}(\xi_x), \mathcal{T}_{\eta_x}(\zeta_x) \rangle_{\mathrm{R}(\eta_x)} = \langle \xi_x, \zeta_x \rangle_x$, which implies that $\|\mathcal{T}_{\eta_x}(\zeta_x)\| = \|\zeta_x\|$. An important vector transport is the parallel transport, which is isometric; refer to (Absil et al., 2008; Boumal, 2023) for the rigorous definition.

In the Euclidean setting, the convergence analyses of FedAvg are established under the assumption that $F$ is $L$-smooth, where a continuously differentiable function $f : \mathbb{R}^n \to \mathbb{R}$ is said $L$-smooth if

$$\|\nabla f(x) - \nabla f(y)\| \leq L\|x - y\| \ \forall x, y \in \mathbb{R}^n,$$

in which case we have

$$f(y) \leq f(x) + \langle \nabla f(x), y - x \rangle + \frac{L}{2}\|y - x\|^2.$$

Both properties above are critical in the analyses of FedAvg. Similar assumptions are made in the Riemannian setting for the analysis of the proposed RFedAGS; see Definitions B.1 (Huang et al., 2018) and B.1 (Huang & Wei, 2022). The first one is called $L$-Lipschitz continuously differentiable (Definitions B.1) and the second one is called $L$-retraction-smooth (Definitions B.2).

**Definition B.1** ($L$-Lipschitz continuous differentiability). *Let $\mathcal{T}$ be a vector transport associated with a retraction $\mathrm{R}$. A function $f : \mathcal{M} \to \mathbb{R}$ is said $L$-Lipschitz continuous differentiable with respect to $\mathcal{T}$ on $\mathcal{U} \subseteq \mathcal{M}$ if there exists a constant $L > 0$ such that*

$$\|\mathrm{grad} f(y) - \mathcal{T}_\eta(\mathrm{grad} f(x))\| \leq L\|\eta\|$$

*for all $x \in \mathcal{U}$ and $\eta \in \mathrm{T}_x \mathcal{M}$ satisfying $y = \mathrm{R}_x(\eta)$.*

**Definition B.2** ($L$-retracton-smoothness). *A function $f : \mathcal{M} \to \mathbb{R}$ is called $L$-retraction-smooth with respect to a retraction $\mathbb{R}$ in $\mathcal{N} \subseteq \mathcal{M}$ if for any $x \in \mathcal{N}$ and any $\mathcal{N}_x \subseteq \mathrm{T}_x \mathcal{M}$ satisfying $\mathrm{R}_x(\mathcal{N}_x) \subseteq \mathcal{N}$, it holds that*

$$f(\mathrm{R}_x(\eta)) \leq f(x) + \langle \mathrm{grad} f(x), \eta \rangle + \frac{L}{2}\|\eta\|^2,$$

*for all $\eta \in \mathcal{N}_x$.*

A function which is $L$-Lipschitz continuously differentiable is not necessarily $L$-retraction-smoooth, however it is the case in the Euclidean setting. It should be highlighted that there exist some cases where $L$-Lipschitz continuous differentiability implies also $L$-retraction smoothness (Huang et al., 2018; Boumal et al., 2019; Boumal, 2023).

Next we review convexity and strong convexity in the Riemannian setting (Huang & Wei, 2022).

**Definition B.3** (Strongly retraction-convex, retraction-convex). *A function $f : \mathcal{M} \to \mathbb{R}$ is called $\mu$-strongly retraction-convex with respect to a retraction $R$ in $\mathcal{N} \subseteq \mathcal{M}$ if for any $x \in \mathcal{N}$ and any $\mathcal{N}_x \subseteq T_x\mathcal{M}$ satisfying $R_x(\mathcal{N}_x) \subseteq \mathcal{N}$, there exist a constant $\mu > 0$ and a tangent vector $\zeta \in T_x\mathcal{M}$ such that $f_x = f \circ R_x$ satisfies*

$$f_x(\eta) \geq f_x(\xi) + \langle \zeta, \eta - \xi \rangle + \frac{\mu}{2}\|\eta - \xi\|^2 \ \forall \eta, \xi \in \mathcal{N}_x.$$

*In particular, if $\mu = 0$, we call $f$ retraction-convex with respect to $R$ in $\mathcal{N}$.*

Note that $\zeta = \operatorname{grad} f_x(\xi)$ if $f$ is differentiable; otherwise, $\zeta$ is any Riemannian subgradient of $f_x$ at $\xi$. In literature, convexity has been studied based on geodesic; see, e.g., (Ferreira & Oliveira, 2002; Zhang & Sra, 2016), in which case a function $f : \mathcal{M} \to \mathbb{R}$ is called geodesic convex, if for any $x, y \in \mathcal{M}$, there exists a tangent vector $\zeta_x \in T_x\mathcal{M}$ such that $f(y) \geq f(x) + \langle \zeta_x, \operatorname{Exp}_x^{-1}(y) \rangle$. It can be verified if taking $\xi = 0$ and exponential mapping as the retraction in Definition B.3, then retraction-convexity reduces to geodesic convexity.

We end this section with an introduction to the concept of $\epsilon$-stationary points/solutions.

**Definition B.4.** *We say that $x_T \in \mathcal{M}$, the output from Algorithm 1, is an $\epsilon$-stationary point of Problem (1.1) if it holds that $\mathbb{E}[\|\operatorname{grad} F(x_T)\|^2] \leq \epsilon$, or is an $\epsilon$-solution if it holds that $\mathbb{E}[F(x_T)] - F(x^*) \leq \epsilon$, where $x^* \in \arg\min_{x \in \mathcal{M}} F(x)$.*

## C ADDITIONAL DISCUSSIONS

### C.1 DISCUSSIONS FOR ASSUMPTIONS

Assumptions 3.1-3.7 are standard for Riemannian stochastic gradient-based methods. Assumptions 3.1 imposes requirements for the retraction under consideration to be $C^2$ and the vector transport under consideration to be continuous and bounded from above. These requirements are fairly standard in Riemannian optimization. Note that the boundedness for vector transport can be achieved by requiring isometricness, in which case we have $\|\mathcal{T}_{\eta_x}(\zeta_x)\| = \|\zeta_x\|$, implying $\Upsilon = 1$. In fact, a lots of papers do have such requirements, e.g., (Sato et al., 2019; Li & Ma, 2023). Additionally, if the Riemannian manifold $\mathcal{M}$ is a submanifold embedded in a Euclidean space and equipped with the inner product as its Riemannian metric, then an option for vector transport is based on the orthogonal operation onto the tangent space, i.e., $\mathcal{T}_{\eta_x}(\zeta_x) = \mathcal{P}_{R(\eta_x)}(\zeta_x)$ with $\mathcal{P}_x(u) = \arg\min_{v \in T_x\mathcal{M}} \|v - u\|_F^2$, in which case by the nonexpansivity of the orthogonal projection we have $\|\mathcal{T}_{\eta_x}(\zeta_x)\| \leq \|\zeta_x\|$, also implying $\Upsilon = 1$.

In the deterministic optimization, the compactness of the sublevel set of the objective function is required to ensure that the iterates generated by the algorithms which are monotonically decreasing are still located in that compact set. However, in the stochastic setting, it is difficult to ensure that the iterates generated by the algorithms all fall within the sublevel set since the algorithms are not necessarily monotonically decreasing, and thus, it is not sufficient to require the sublevel set to be compact under stochastic optimization. In this case, Assumption 3.2 becomes a commonly used choice in Riemannian stochastic optimization; see, e.g., (Bonnabel, 2013; Zhang & Sra, 2016; Tripuraneni et al., 2018; Sato et al., 2019; Kasai et al., 2019; Han & Gao, 2021; Li & Ma, 2023). For some manifolds that are compact themself, e.g., the Stiefel manifold and the Grassmann manifold, the compactness assumption naturally holds. Moreover, in all experiments we conducted, it is observed that the generated iterates $x_t$, with $t \geq 1$, fall into the sublevel set $\{x \in \mathcal{M} : f(x) \leq f(x_1)\}$.

Assumptions 3.6 and 3.7 impose requirements on the first- and second-order moments for the local stochastic gradient estimator, which are necessary for Riemannian/Euclidean stochastic gradient-based methods. In the analyses for Euclidean federated learning algorithms, majority of works make extra assumptions for addressing the heterogeneity data. These assumptions essentially require that

the divergence between local and global gradients is bounded, i.e., there exists a constant $\sigma > 0$ such that for all $x$,

$$\|\nabla f_i(x) - \nabla F(x)\|^2 \leq \sigma^2.$$

In our analyses for the proposed RFedAGS, we do not explicitly make the similar assumption, since Assumption 3.2 implies the counterpart requirement. Indeed, under Assumption 3.2, there exists a constant $P > 0$, such that $\|\mathrm{grad} f_i(x)\| \leq P$ and $\|\mathrm{grad} F(x)\| \leq P$ for all $i \in [N]$ and $x \in \mathcal{W}$. Hence, it holds that

$$\|\mathrm{grad} f_i(x) - \mathrm{grad} F(x)\|^2 \leq 2\|\mathrm{grad} f_i(x)\|^2 + 2\|\mathrm{grad} F(x)\|^2 \leq 4P^2.$$

Assumption 3.8 imposes the requirement that the approximate probabilities are how close to the true probabilities. As discussed in Section 3.3, when using frequencies as the approximation, this assumption holds with high probability. Numerically, the reported results show that the performance using frequencies is comparable to the case using true probabilities. We note that in the fixed step size case, existing work (Wang & Ji, 2024) also makes an equivalent assumption. The difference lies in that the assumption in (Wang & Ji, 2024) only considers fixed step size cases, but Assumption 3.8 more finely encompasses the cases of decaying step sizes.

In summary, except Assumption 3.8 that aims to address the arbitrary partial participation, there exists no assumption beyond those made for Riemannian (stochastic) optimization and federated learning. In theory, the proposed RFedAGS is the first algorithm that can simultaneously address the challenges caused by the partial participation and the heterogeneity data settings. The partial participation under consideration allows arbitrary participation which is more practical than the commonly-countered participation scheme based on random sampling. Even without the Riemannian manifold constraint, i.e., $\mathcal{M} = \mathbb{R}^n$, the proposed RFedAGS can reduce to one proposed in (Wang & Ji, 2024). This paper establishes the convergence propoerties of RFedAGS under both the decaying (see Theorems 3.1, 3.2, and 3.3) and fixed (see Theorems 3.4 and 3.5) step size cases. Under the decaying step size case, global convergence is guaranteed. These analyses depend on a vital and non-trivial observation (see Assumption 3.8). However, (Wang & Ji, 2024) only considered the assumption of the fixed step case, and thus only established convergence under the fixed step size case, which does not ensure global convergence rather only converges to a $\epsilon$-stationary point.

## C.2 Discussions for Implementations

In Algortihm 1, there exists a scenario (called NA) where in certain round of communication no agent participates in communication. We emphasize that this scenario happens with fairly low probability. For example, considering a FL system where 20 agents participate in communication with probability $p_i = 0.1$, $i = 1, 2, \ldots, 20$, and 5 agents participate with probability $p_i = 0.5$, $i = 21, 22, \ldots, 25$. Then the scenario NA happens only with probability not greater than $0.38\%$. For the purpose of robustness, when the scenario NA happens, one option is set $x_{t+1} \leftarrow x_t$ to restart the next round of local updates.

## D Proofs of Theorems in Section 3

### D.1 Supporting lemmas

If the objective $F$ in Problem (1.1) is $L_g$-retraction smooth (Assumption 3.5), under Assumption 3.1, it follows that

$$\mathbb{E}_t[F(x_{t+1})] - F(x_t) \leq \mathbb{E}_t[\langle \mathrm{grad} F(x_t), \mathrm{R}_{x_t}^{-1}(x_{t+1})\rangle] + \frac{L_g}{2}\mathbb{E}_t[\|\mathrm{R}_{x_t}^{-1}(x_{t+1})\|^2]. \tag{D.1}$$

Without considering the arbitrary participation, recalling (TM) and (AGS-RS), we have

$$\mathrm{Exp}_{x_t}^{-1}(x_{t+1}) = \frac{1}{S}\sum_{j \in \mathcal{S}_t} \mathrm{Exp}_{x_t}^{-1}(x_{t,K}^j), \text{ and} \tag{D.2}$$

$$\mathrm{R}_{x_t}^{-1}(x_{t+1}) = -\alpha_t \frac{1}{S}\sum_{j \in \mathcal{S}_t}\sum_{k=0}^{K-1} \mathcal{T}_{\tilde{\eta}_{t,k}^j}\left(\frac{1}{B_t}\sum_{b \in \mathcal{B}_{t,k}^j} f_j(x_{t,k}^j; \xi_{t,k,b}^j)\right). \tag{D.3}$$

When $K > 1$ and $S > 1$, from the increment of parameters of (TM) it follows that analyzing the upper bounds of the two terms in the right-hand side of (D.1) is fairly challenging, since the nonlinearity of exponential and its inverse leads to difficulty to expand (D.2) into the desired one involved gradient information. However, the form of (D.3) is very similar to the Euclidean version and thus significantly address the issue.

Lemmas D.1 together with D.3 have provided an upper bound for the first term in the right-hand side of (D.1).

**Lemma D.1.** *Under Assumptions 3.1-3.5, at the $t$-th outer iteration of Algorithm 1 with a stepsize $\alpha_t$ and a batchsize $B_t$, we have that*

$$\mathbb{E}_t[\langle \mathrm{grad}F(x_t), \mathrm{R}_{x_t}^{-1}(x_{t+1})\rangle] \leq \varpi\alpha_t L_f^2 \delta_1^2 \max_j \left\{ \sum_{k=0}^{K-1} \mathbb{E}_t[\|\mathrm{R}_{x_t}^{-1}(x_{t,k}^j)\|^2] \right\} + \varpi\alpha_t^2 KGP^2\delta_2^2$$

$$- \frac{\varpi\alpha_t K}{2}\|\mathrm{grad}F(x_t)\|^2 - \frac{\varpi\alpha_t}{2}\sum_{k=0}^{K-1}\mathbb{E}\left[\left\|\sum_{j=1}^{N}\frac{p_j}{q_t^j N}\mathcal{T}_{\tilde{\eta}_{t,k}^j}(\mathrm{grad}f_j(x_{t,k}^j))\right\|^2\right], \tag{D.4}$$

*where $\delta_1^2 = \max_{t\geq 1}\left\{\frac{1}{N}\sum_{j=1}^{N}\left(\frac{p_j}{q_t^j}\right)^2\right\}$ and $\delta_2^2 = \sum_{j=1}^{N}\frac{p_j^2}{N}$.*

*Proof of Lemma D.1.* On the one hand, we have

$$\mathbb{E}_t[\langle \mathrm{grad}F(x_t), \mathrm{R}_{x_t}^{-1}(x_{t+1})\rangle]$$
$$= \mathbb{E}_t[\langle \mathrm{grad}F(x_t), \mathrm{R}_{x_t}^{-1}(x_{t+1}) + \varpi\alpha_t K\mathrm{grad}F(x_t) - \varpi\alpha_t K\mathrm{grad}F(x_t)\rangle]$$
$$= -\varpi\alpha_t K\|\mathrm{grad}F(x_t)\|^2 + \mathbb{E}_t[\langle \mathrm{grad}F(x_t), \mathrm{R}_{x_t}^{-1}(x_{t+1}) + \varpi\alpha_t K\mathrm{grad}F(x_t)\rangle], \tag{D.5}$$

where for the second term of the equality on the right-hand side, we have

$$\mathbb{E}_t[\langle \mathrm{grad}F(x_t), \mathrm{R}_{x_t}^{-1}(x_{t+1}) + \varpi\alpha_t K\mathrm{grad}F(x_t)\rangle]$$

$$= \mathbb{E}_t\left[\left\langle \mathrm{grad}F(x_t), -\sum_{j\in\mathcal{S}_t}\frac{\varpi\alpha_t}{q_t^j N}\sum_{k=0}^{K-1}\frac{1}{B_t}\sum_{b\in\mathcal{B}_{t,k}^j}\mathcal{T}_{\tilde{\eta}_{t,k}^j}(\mathrm{grad}f_j(x_{t,k}^j;\xi_{t,k,b}^j)) + \varpi\alpha_t K\mathrm{grad}F(x_t)\right\rangle\right]$$

$$= \mathbb{E}_t\left[\left\langle \mathrm{grad}F(x_t), -\sum_{k=0}^{K-1}\varpi\alpha_t\left(\sum_{j\in\mathcal{S}_t}\frac{1}{q_t^j NB_t}\sum_{b\in\mathcal{B}_{t,k}^j}\mathcal{T}_{\tilde{\eta}_{t,k}^j}(\mathrm{grad}f_j(x_{t,k}^j;\xi_{t,k,b}^j)) + \mathrm{grad}F(x_t)\right)\right\rangle\right]$$

$$= \sum_{k=0}^{K-1}\mathbb{E}_t\left[\left\langle \mathrm{grad}F(x_t), -\varpi\alpha_t\sum_{j\in\mathcal{S}_t}\left(\frac{1}{q_t^j N}\mathcal{T}_{\tilde{\eta}_{t,k}^j}(\mathrm{grad}f_j(x_{t,k}^j)) - \frac{1}{p_j N}\mathrm{grad}f_j(x_t)\right)\right\rangle\right]$$

$$= \sum_{k=0}^{K-1}\mathbb{E}_t\left[\left\langle \sqrt{\varpi\alpha_t}\mathrm{grad}F(x_t), -\sum_{j\in\mathcal{S}_t}\frac{\sqrt{\varpi\alpha_t}}{N}\left(\frac{1}{q_t^j}\mathcal{T}_{\tilde{\eta}_{t,k}^j}(\mathrm{grad}f_j(x_{t,k}^j)) - \frac{1}{p_j}\mathrm{grad}f_j(x_t)\right)\right\rangle\right]$$

$$= \sum_{k=0}^{K-1}\mathbb{E}_t\left[\left\langle \sqrt{\varpi\alpha_t}\mathrm{grad}F(x_t), -\sum_{j=1}^{N}\mathbb{I}_{\mathcal{S}_t}(j)\frac{\sqrt{\varpi\alpha_t}}{N}\left(\frac{1}{q_t^j}\mathcal{T}_{\tilde{\eta}_{t,k}^j}(\mathrm{grad}f_j(x_{t,k}^j)) - \frac{1}{p_j}\mathrm{grad}f_j(x_t)\right)\right\rangle\right]$$

$$= \sum_{k=0}^{K-1}\mathbb{E}_t\left[\left\langle \sqrt{\varpi\alpha_t}\mathrm{grad}F(x_t), -\sum_{j=1}^{N}\frac{p_j\sqrt{\varpi\alpha_t}}{N}\left(\frac{1}{q_t^j}\mathcal{T}_{\tilde{\eta}_{t,k}^j}(\mathrm{grad}f_j(x_{t,k}^j)) - \frac{1}{p_j}\mathrm{grad}f_j(x_t)\right)\right\rangle\right]$$

$$= \frac{\varpi\alpha_t K}{2}\|\mathrm{grad}F(x_t)\|^2 + \frac{\varpi\alpha_t}{2}\sum_{k=0}^{K-1}\mathbb{E}_t\left[\left\|\sum_{j=1}^{N}\frac{p_j}{N}\left(\frac{1}{q_t^j}\mathcal{T}_{\tilde{\eta}_{t,k}^j}(\mathrm{grad}f_j(x_{t,k}^j)) - \frac{1}{p_j}\mathrm{grad}f_j(x_t)\right)\right\|^2\right]$$

$$- \frac{\varpi\alpha_t}{2}\sum_{k=0}^{K-1}\mathbb{E}_t\left[\left\|\sum_{j=1}^{N}\frac{p_j}{q_t^j N}\mathcal{T}_{\tilde{\eta}_{t,k}^j}(\mathrm{grad}f_j(x_{t,k}^j))\right\|^2\right], \tag{D.6}$$

where the third equality follows (2.1), the sixth equality follows $\mathbb{E}[\mathbb{I}_{\mathcal{S}_t}(j)] = p_j$, and the last equality is due to $\langle u, v \rangle = \frac{1}{2}(\|u\|^2 + \|v\|^2 - \|u - v\|^2)$. Moreover, we note that

$$
\sum_{k=0}^{K-1} \mathbb{E}_t \left[ \left\| \sum_{j=1}^{N} \frac{p_j}{N} \left( \frac{1}{q_t^j} \mathcal{T}_{\tilde{\eta}_{t,k}^j}(\mathrm{grad} f_j(x_{t,k}^j)) - \frac{1}{p_j} \mathrm{grad} f_j(x_t) \right) \right\|^2 \right]
$$

$$
= \sum_{k=0}^{K-1} \mathbb{E}_t \left[ \left\| \sum_{j=1}^{N} \frac{p_j}{N} \left( \frac{1}{q_t^j} \left( \mathcal{T}_{\tilde{\eta}_{t,k}^j}(\mathrm{grad} f_j(x_{t,k}^j)) - \mathrm{grad} f_j(x_t) \right) + \left( \frac{1}{q_t^j} - \frac{1}{p_j} \right) \mathrm{grad} f_j(x_t) \right) \right\|^2 \right]
$$

$$
\leq 2 \sum_{k=0}^{K-1} \mathbb{E}_t \left[ \left\| \sum_{j=1}^{N} \frac{p_j}{q_t^j N} \left( \mathcal{T}_{\tilde{\eta}_{t,k}^j}(\mathrm{grad} f(x_{t,k}^j)) - \mathrm{grad} f_j(x_t) \right) \right\|^2 \right]
$$

$$
+ 2 \sum_{k=0}^{K-1} \mathbb{E}_t \left[ \left\| \sum_{j=1}^{N} \frac{p_j}{N} \left( \frac{1}{q_t^j} - \frac{1}{p_j} \right) \mathrm{grad} f_j(x_t) \right\|^2 \right]
$$

$$
\leq \frac{2L_f^2}{N} \sum_{j=1}^{N} \left( \frac{p_j}{q_t^j} \right)^2 \sum_{k=0}^{K-1} \mathbb{E}_t[\|\mathrm{R}_{x_t}^{-1}(x_{t,k}^j)\|^2] + 2\alpha_t K G P^2 \sum_{j=1}^{N} \frac{p_j^2}{N}
$$

$$
\leq 2\delta_1^2 L_f^2 \max_j \left\{ \sum_{k=0}^{K-1} \mathbb{E}_t[\|\mathrm{R}_{x_t}^{-1}(x_{t,k}^j)\|^2] \right\} + 2\alpha_t K G P^2 \delta_2^2, \tag{D.7}
$$

where the first inequality follows $\|u + v\|^2 \leq 2\|u\|^2 + 2\|v\|^2$, the second inequality is due to the $L_f$-retraction smoothness of $\mathrm{grad} f_j$ for $j = 1, 2, \ldots, N$, Assumption 3.2 (which implies that there exists $P > 0$ such that $\|\mathrm{grad} f_i(x_t)\| \leq P$), 3.4, and 3.8, and the third inequality follows that $\delta_1^2 = \max_{t \geq 1} \left\{ \frac{1}{N} \sum_{i=1}^{N} \left( \frac{p_i}{q_t^i} \right)^2 \right\}$ and $\delta_2^2 = \sum_{i=1}^{N} \frac{p_i^2}{N}$. Combining (D.5), (D.6), and (D.7) yields the desired result. $\qquad \square$

In order to further bound $\mathbb{E}_t[\langle \mathrm{grad} F(x_t), \mathrm{R}_{x_t}^{-1}(x_{t+1}) \rangle]$ for $K > 1$, from Lemma D.1, it is necessary to estimate the bounds for $\mathbb{E}_t[\|\mathrm{R}_{x_t}^{-1}(x_{t,k}^j)\|^2]$, as theoretically discussed in Lemma D.3 which states that for agent $j$, the "distance" between the $k$-th local update $x_{t,k}^j$ and the the $t$-th outer iterate $x_t$ are controlled by the sum of squared step sizes. Intuitively, the "distance" increases as the number of local iterations grows, which is shown in Lemma D.4. Meanwhile, it also reflects the drift between an agent's local update parameter $x_{t,k}^j$ and the global parameter $x_t$. A general result is provided in Lemma D.2.

**Lemma D.2.** *Under Assumptions 3.1-3.3, let $F : \mathcal{M} \to \mathbb{R}$ be a smooth function. If consider the following update formulation*

$$
x_{t,k+1} = \mathrm{R}_{x_{t,k}}(-\alpha_{t,k} \mathcal{G}_F(x_{t,k})),
$$

*where $\mathcal{G}_F(x_{t,k})$ is an estimator of $\mathrm{grad} F(x_{t,k})$, $x_t = x_{t,0}$, and $\alpha_{t,\tau}$ is the step size, then it follows that*

$$
\|\mathrm{R}_{x_t}^{-1}(x_{t,k})\|^2 \leq 2k \sum_{\tau=0}^{k-1} \alpha_{t,\tau}^2 (J^2 + \alpha_{t,\tau}^2 H^2 \|\mathcal{G}_F(x_{t,\tau})\|^2) \|\mathcal{G}_F(x_{t,\tau})\|^2,
$$

*where $J$ and $H$ are two positive constants related with the manifold and retraction.*

The proof of Lemma D.2 needs the following inverse function theorem on manifolds.

**Theorem D.1** (Inverse function theorem). *Given a smooth mapping $P : \mathcal{M} \to \mathcal{M}'$ defined between two manifolds, if $\mathrm{D}P(x)$ is invertible at some point $x \in \mathcal{M}$, then there exist neighborhoods $\mathcal{U}_x \subseteq \mathcal{M}$ of $x$ and $\mathcal{V}_{P(x)} \subseteq \mathcal{M}'$ of $P(x)$ such that $P|_{\mathcal{U}_x} : \mathcal{U}_x \to \mathcal{V}_{P(x)}$ is a diffeomorphism. Meanwhile, if $P^{-1}$ is the inverse of $P$ in $\mathcal{U}_x$, then we have $(\mathrm{D}P(x))^{-1} = \mathrm{D}P^{-1}(P(x))$.*

Now we are ready to prove Lemma D.2.

*Proof of Lemma D.2.* For two points $x, y \in \mathcal{W}$, consider the map $P_{x,y} = \mathrm{R}_y^{-1} \circ \mathrm{R}_x : \mathrm{T}_x\mathcal{M} \to \mathrm{T}_y\mathcal{M} : \eta_x \mapsto \mathrm{R}_y^{-1}(\mathrm{R}_x(\eta_x))$, which is defined between two vector spaces. According to the chain rule for the differential of a map and the first-order property of the retraction, i.e., $\mathrm{DR}_x(0_x) = \mathrm{I}_{\mathrm{T}_x\mathcal{M}}$, we have

$$\mathrm{D}P_{x,y}(0_x) = \mathrm{D}(\mathrm{R}_y^{-1} \circ \mathrm{R}_x)(0_x) = \mathrm{DR}_y^{-1}(\mathrm{R}_x(0_x)) \circ \mathrm{DR}_x(0_x)$$
$$= (\mathrm{DR}_y(\mathrm{R}_y^{-1}(R_x(0_x))))^{-1} \circ \mathrm{I}_{\mathrm{T}_x\mathcal{M}} = (\mathrm{DR}_y(\mathrm{R}_y^{-1}(x)))^{-1} = (\Lambda_y^x)^{-1},$$

where the third equality is due to the inverse function Theorem D.1. Noting that the map $P_{\cdot,\cdot}(\cdot)$ is defined in $\mathrm{T}_\mathcal{W} = \{(x, y, \eta) : x, y \in \mathcal{W}, \eta \in \mathrm{R}_x^{-1}(\mathcal{W})\}$, which is inside a compact set, according to Assumption 3.2, thus, smoothness of the retraction implies that the Jacobin and Hessian of $P_{\cdot,\cdot}(\cdot)$ with respect to the third variable is uniformly bounded in norm on the compact set. We, thus, use $C_2, C_3 > 0$ to denote bounds on the operator norms of the Jacobin and Hessian of $P_{\cdot,\cdot}(\cdot)$ with respect to the third variable in the compact set. Noting that

$$P_{x_{t,k-1}^j, x_t}(\eta_{x_{t,k-1}^j}) = \mathrm{R}_{x_t}^{-1}(\mathrm{R}_{x_{t,k-1}^j}(\eta_{x_{t,k-1}^j})) = \mathrm{R}_{x_t}^{-1}(x_{t,k}^j), \text{ and}$$
$$P_{x_{t,k-1}^j, x_t}(0) = \mathrm{R}_{x_t}^{-1}(\mathrm{R}_{x_{t,k-1}^j}(0)) = \mathrm{R}_{x_t}^{-1}(x_{t,k-1}^j)$$

with $\eta_{x_{t,k-1}^j} = -\alpha_{t,k-1}\mathcal{G}_F(x_{t,k-1}^j)$, using a Taylor expansion for $P_{x,y}$ yields

$$\mathrm{R}_{x_t}^{-1}(x_{t,k}^j) = P_{x_{t,k-1}^j, x_t}(-\alpha_{t,k-1}\mathcal{G}_F(x_{t,k-1}^j))$$
$$= P_{x_{t,k-1}^j, x_t}(0) + \mathrm{D}P_{x_{t,k-1}^j, x_t}(0)(-\alpha_{t,k-1}\mathcal{G}_F(x_{t,k-1}^j)) + \alpha_{t,k-1}e_{t,k-1}^j$$
$$= \mathrm{R}_{x_t}^{-1}(x_{t,k-1}^j) - \alpha_{t,k-1}(\Lambda_{x_t}^{x_{t,k-1}^j})^{-1}(\mathcal{G}_F(x_{t,k-1}^j)) + \alpha_{t,k-1}e_{t,k-1}^j,$$

where $\|e_{t,k-1}^j\| \leq \alpha_{t,k-1}C_3\|\mathcal{G}_F(x_{t,k-1}^j)\|^2$. Hence, repeatedly, we have

$$\mathrm{R}_{x_t}^{-1}(x_{t,k}^j) = -\sum_{\tau=0}^{k-1} \alpha_{t,\tau}(\Lambda_{x_t}^{x_{t,\tau}^j})^{-1}(\mathcal{G}_F(x_{t,\tau}^j)) + \sum_{\tau=0}^{k-1} \alpha_{t,\tau}e_{t,\tau}^j, \tag{D.8}$$

where we used $\mathrm{R}_{x_t}^{-1}(x_t) = 0_{x_t}$. Combining (D.8), $\|(\Lambda_{x_t}^{x_{t,k-1}^j})^{-1}(\mathcal{G}_F(x_{t,k-1}^j))\| \leq C_2\|\mathcal{G}_F(x_{t,k-1}^j)\|$ (for all $t = 1, 2, \ldots, T-1$ and $k = 1, 2, \ldots, K-1$), and $\|\sum_{i=1}^n u_i\|^2 \leq n\sum_{i=1}^n \|u_i\|^2$ yields the desired result. □

When $\mathcal{M}$ reduces into a Euclidean space, e.g., $\mathcal{M} = \mathbb{R}^d$, the constants in Lemma D.2 will become $C_2 = 1$ and $C_3 = 0$. In this case, the results correspondingly becomes $\|x_t - x_{t,k}^j\|^2 \leq k\sum_{\tau=0}^{k-1} \alpha_{t,\tau}^2\|\mathcal{G}_F(x_{t,\tau}^j)\|^2$. In Lemma D.2, if one uses $\frac{1}{B_t}\sum_{b\in\mathcal{B}_{t,k}^j} \mathrm{grad}f_j(x_{t,k}^j; \xi_{t,k,b}^j)$ to replace $\mathcal{G}_F(x_{t,k}^j)$, then the desired result is obtained in Lemma D.3.

**Lemma D.3.** *Under Assumptions 3.1-3.3, at the $k$-th inner iteration of the $t$-th outer iteration of Algorithm 1, for each agent $j \in \mathcal{S}_t$ and $k = 1, 2, \ldots, K-1$, we have*

$$\|\mathrm{R}_{x_t}^{-1}(x_{t,k}^j)\|^2 \leq 2k^2\alpha_t^2 P^2(J^2 + \alpha_t^2 P^2 H^2), \tag{D.9}$$

*where $P$ is a positive constant such that for all $x \in \mathcal{W}$, $j = 1, 2, \ldots, N$ and $\xi \sim \mathcal{D}_j$, it holds that $\|\mathrm{grad}F(x)\| \leq P$, $\|\mathrm{grad}f_j(x)\| \leq P$ and $\|\mathrm{grad}f_j(x; \xi)\| \leq P$ by Assumption 3.2.*

*Proof of Lemma D.3.* From Algorithm 1, letting $\mathcal{G}_F(x_{t,k}^j) = -\frac{1}{B_t}\sum_{b\in\mathcal{B}_{t,k}^j} \mathrm{grad}f_j(x_{t,k}^j; \xi_{t,k,b}^j)$, then, we have

$$\|\mathcal{G}_F(x_{t,k}^j)\| = \left\| -\frac{1}{B_t}\sum_{b\in\mathcal{B}_{t,k}^j} \mathrm{grad}f_j(x_{t,k}^j; \xi_{t,k,b}^j) \right\| \leq \frac{1}{B_t}\sum_{b\in\mathcal{B}_{t,k}^j} \|\mathrm{grad}f_j(x_{t,k}^j; \xi_{t,k,b}^j)\| \leq P.$$

Hence, combining the inequality above and Lemma D.2 gives rise to the desired result (D.9). □

Under the same conditions as Lemma D.1, plugging (D.9) into (D.4) yields

$$
\mathbb{E}_t[\langle \mathrm{grad} F(x_t), \mathrm{R}_{x_t}^{-1}(x_{t+1})\rangle] \leq -\frac{\varpi \alpha_t}{2} \sum_{k=0}^{K-1} \mathbb{E}_t \left[ \left\| \sum_{j=1}^{N} \frac{p_j}{q_t^j N} \mathcal{T}_{\tilde{\eta}_{t,k}^j}(\mathrm{grad} f_j(x_{t,k}^j)) \right\|^2 \right]
$$

$$
-\frac{\varpi \alpha_t K}{2} \|\mathrm{grad} F(x_t)\|^2 + \frac{1}{6}(2K-1)K(K-1)L_f^2 \delta_1^2 P^2 (J^2 + \alpha_t^2 P^2 H^2)\varpi \alpha_t^3
$$

$$
+ \varpi \alpha_t^2 K G P^2 \delta_2^2 \tag{D.10}
$$

The next is to bound the second term $\mathbb{E}_t[\|\mathrm{R}_{x_t}^{-1}(x_{t+1})\|^2]$.

**Lemma D.4.** *Under Assumptions 3.1-3.8, the iterates $\{x_t\}_{t=1}^{T}$ generated by Algorithm 1 with fixed stepsize $\alpha_t$ and fixed batchsize $B_t$ within parallel inner iterations satisfies*

$$
\mathbb{E}_t[\|\mathrm{R}_{x_t}^{-1}(x_{t+1})\|^2] \leq \frac{\varpi^2 \alpha_t^2 \Upsilon^2 \sigma_L^2 \delta_3^2 K}{B_t} + \varpi^2 \alpha_t^2 K \sum_{k=0}^{K-1} \mathbb{E}_t \left[ \left\| \sum_{j=1}^{N} \frac{p_j}{q_t^j N} \mathcal{T}_{\tilde{\eta}_{t,k}^j}(\mathrm{grad} f_j(x_{t,k}^j)) \right\|^2 \right]
$$

$$
+ \varpi^2 \alpha_t^2 P^2 K^2 \delta_4^2 \tag{D.11}
$$

*where $\delta_3^2 = \frac{1}{N^2} \sum_{j=1}^{N} \frac{p_j}{(q_t^j)^2}$ and $\delta_4^2 = \frac{1}{N^2} \sum_{j=1}^{N} \frac{p_j(1-p_j)}{(q_t^j)^2}$.*

*Proof of Lemma D.4.* Let $x_t$ denote the $t$-th aggregation by the server. Then,

$$
\mathbb{E}_t[\|\mathrm{R}_{x_t}^{-1}(x_{t,k}^j)\|^2] = \varpi^2 \alpha_t^2 \mathbb{E}_t \left[ \left\| \sum_{j \in \mathcal{S}_t} \frac{1}{q_t^j N} \sum_{k=0}^{K-1} \mathcal{T}_{\tilde{\eta}_{t,k}^j} \left( \frac{1}{B_t} \sum_{b \in \mathcal{B}_{t,k}^j} \mathrm{grad} f_j(x_{t,k}^j; \xi_{t,k,b}^j) \right) \right\|^2 \right]
$$

$$
= \varpi^2 \alpha_t^2 \mathbb{E}_t \left[ \left\| \sum_{j=1}^{N} \mathbb{I}_{\mathcal{S}_t}(j) \frac{1}{q_t^j N} \sum_{k=0}^{K-1} \mathcal{T}_{\tilde{\eta}_{t,k}^j} \left( \frac{1}{B_t} \sum_{b \in \mathcal{B}_{t,k}^j} \mathrm{grad} f_j(x_{t,k}^j; \xi_{t,k,b}^j) \right) \right\|^2 \right]
$$

$$
= \varpi^2 \alpha_t^2 \mathbb{E}_t \left[ \left\| \sum_{j=1}^{N} \mathbb{I}_{\mathcal{S}_t}(j) \frac{1}{q_t^j N} \sum_{k=0}^{K-1} \mathcal{T}_{\tilde{\eta}_{t,k}^j} \left( \frac{1}{B_t} \sum_{b \in \mathcal{B}_{t,k}^j} \mathrm{grad} f_j(x_{t,k}^j; \xi_{t,k,b}^j) - \mathrm{grad} f_j(x_{t,k}^j) \right.\right.\right.
$$

$$
\left.\left.\left. + \mathrm{grad} f_j(x_{t,k}^j) \right) \right\|^2 \right]
$$

$$
= \varpi^2 \alpha_t^2 \mathbb{E}_t \left[ \left\| \sum_{j=1}^{N} \frac{\mathbb{I}_{\mathcal{S}_t}(j)}{q_t^j N} \sum_{k=0}^{K-1} \mathcal{T}_{\tilde{\eta}_{t,k}^j} \left( \frac{1}{B_t} \sum_{b \in \mathcal{B}_{t,k}^j} \mathrm{grad} f_j(x_{t,k}^j; \xi_{t,k,b}^j) - \mathrm{grad} f_j(x_{t,k}^j) \right) \right\|^2 \right]
$$

$$
+ \varpi^2 \alpha_t^2 \mathbb{E}_t \left[ \left\| \sum_{j=1}^{N} \mathbb{I}_{\mathcal{S}_t}(j) \frac{1}{q_t^j N} \sum_{k=0}^{K-1} \mathcal{T}_{\tilde{\eta}_{t,k}^j} \left( \mathrm{grad} f_j(x_{t,k}^j) \right) \right\|^2 \right]
$$

$$
= \varpi^2 \alpha_t^2 \mathbb{E}_t \left[ \left\| \sum_{j=1}^{N} \frac{\mathbb{I}_{\mathcal{S}_t}(j)}{q_t^j N} \sum_{k=0}^{K-1} \mathcal{T}_{\tilde{\eta}_{t,k}^j} \left( \frac{1}{B_t} \sum_{b \in \mathcal{B}_{t,k}^j} \mathrm{grad} f_j(x_{t,k}^j; \xi_{t,k,b}^j) - \mathrm{grad} f_j(x_{t,k}^j) \right) \right\|^2 \right]
$$

$$
+ \varpi^2 \alpha_t^2 \mathbb{E}_t \left[ \left\| \sum_{j=1}^{N} (\mathbb{I}_{\mathcal{S}_t}(j) - p_j + p_j) \frac{1}{q_t^j N} \sum_{k=0}^{K-1} \mathcal{T}_{\tilde{\eta}_{t,k}^j} \left( \mathrm{grad} f_j(x_{t,k}^j) \right) \right\|^2 \right]
$$

$$
\leq \frac{\varpi^2 \alpha_t^2 \Upsilon^2 \sigma_L^2 K}{N^2 B_t} \sum_{j=1}^{N} \frac{p_j}{(q_t^j)^2} + \varpi^2 \alpha_t^2 \mathbb{E}_t \left[ \left\| \sum_{j=1}^{N} (\mathbb{I}_{\mathcal{S}_t}(j) - p_j) \frac{1}{q_t^j N} \sum_{k=0}^{K-1} \mathcal{T}_{\tilde{\eta}_{t,k}^j}(\mathrm{grad} f_j(x_{t,k}^j)) \right\|^2 \right]
$$

$$
+ \varpi^2 \alpha_t^2 \mathbb{E}_t \left[ \left\| \sum_{j=1}^{N} \frac{p_j}{q_t^j N} \sum_{k=0}^{K-1} \mathcal{T}_{\tilde{\eta}_{t,k}^j}(\mathrm{grad} f_j(x_{t,k}^j)) \right\|^2 \right]
$$

$$= \frac{\varpi^2 \alpha_t^2 \Upsilon^2 \sigma_L^2 K}{N^2 B_t} \sum_{j=1}^{N} \frac{p_j}{(q_t^j)^2} + \frac{\varpi^2 \alpha_t^2}{N^2} \sum_{j=1}^{N} \frac{p_j(1-p_j)}{(q_t^j)^2} \mathbb{E}_t \left[ \left\| \sum_{k=0}^{K-1} \mathcal{T}_{\tilde{\eta}_{t,k}^j} (\mathrm{grad} f_j(x_{t,k}^j)) \right\|^2 \right]$$

$$+ \varpi^2 \alpha_t^2 \mathbb{E}_t \left[ \left\| \sum_{j=1}^{N} \frac{p_j}{q_t^j N} \sum_{k=0}^{K-1} \mathcal{T}_{\tilde{\eta}_{t,k}^j} (\mathrm{grad} f_j(x_{t,k}^j)) \right\|^2 \right]$$

$$\leq \frac{\varpi^2 \alpha_t^2 \Upsilon^2 \sigma_L^2 \delta_3^2 K}{B_t} + \varpi^2 \alpha_t^2 \Upsilon^2 P^2 K^2 \delta_4^2 + \varpi^2 \alpha_t^2 K \sum_{k=0}^{K-1} \mathbb{E}_t \left[ \left\| \sum_{j=1}^{N} \frac{p_j}{q_t^j N} \mathcal{T}_{\tilde{\eta}_{t,k}^j} (\mathrm{grad} f_j(x_{t,k}^j)) \right\|^2 \right]$$

where the fourth equality follows that

$$\mathbb{E} \left[ \sum_{j=1}^{N} \sum_{k=0}^{K-1} \frac{\mathbb{I}_{\mathcal{S}_t}(j)}{q_t^j N} \mathcal{T}_{\tilde{\eta}_{t,k}^j} \left( \frac{1}{B_t} \sum_{b \in \mathcal{B}_{t,k}^j} \mathrm{grad} f_j(x_{t,k}^j; \xi_{t,k,b}^j) \right) \right] = \sum_{j=1}^{N} \sum_{k=0}^{K-1} \frac{\mathbb{I}_{\mathcal{S}_t}(j)}{q_t^j N} \mathcal{T}_{\tilde{\eta}_{t,k}^j} (\mathrm{grad} f_j(x_{t,k}^j))$$

and that $\mathbb{E}[\|u\|^2] = \mathbb{E}[\|u - \mathbb{E}[u]\|^2] + \|\mathbb{E}[u]\|^2$, the first inequality follows that

$$\mathbb{E} \left[ \sum_{j=1}^{N} \sum_{k=0}^{K-1} \frac{\mathbb{I}_{\mathcal{S}_t}(j)}{q_t^j N} \mathcal{T}_{\tilde{\eta}_{t,k}^j} \left( \frac{1}{B_t} \sum_{b \in \mathcal{B}_{t,k}^j} \mathrm{grad} f_j(x_{t,k}^j; \xi_{t,k,b}^j) - \mathrm{grad} f_j(x_{t,k}^j) \right) \right] = 0$$

and that $\mathbb{E}[\| \sum_{i=1}^{n} u_i\|^2] = \sum_{i=1}^{n} \mathbb{E}[\|u_i\|^2]$ with $u_i$ being independent and having zero mean, that $\|\mathcal{T}_\eta(\zeta)\| \leq \Upsilon$ (Assumption 3.1), and Assumption 3.6, the sixth equality follows that

$$\mathbb{E} \left[ \sum_{j=1}^{N} (\mathbb{I}_{\mathcal{S}_t}(j) - p_j) \frac{1}{q_t^j N} \sum_{k=0}^{K-1} \mathcal{T}_{\tilde{\eta}_{t,k}^j} (\mathrm{grad} f_j(x_{t,k}^j)) \right] = 0,$$

and that $\mathbb{E}[(\mathbb{I}_{\mathcal{S}_t}(j) - p_j)^2] = p_j(1 - p_j)$, and the last inequality follows that $\delta_3^2 = \max_{t \geq 1} \left\{ \frac{1}{N^2} \sum_{j=1}^{N} \frac{p_j}{(q_t^j)^2} \right\}$, $\delta_4^2 = \max_{t \geq 1} \left\{ \frac{1}{N^2} \sum_{j=1}^{N} \frac{p_j(1-p_j)}{(q_t^j)^2} \right\}$, and $\|\sum_{i=1}^{n} u_i\|^2 \leq n \sum_{i=1}^{n} \|u_i\|^2$. $\square$

Now we can formally state the descent lemma in the Riemannian FL setting.

**Lemma D.5.** *Under Assumptions 3.1-3.8, we run Algorithm 1 with batch size $B_t$ and step sizes $\varpi > 0$ and $\{\alpha_t\}$ satisfying*

$$1 \geq KL_g \varpi \alpha_t. \tag{D.12}$$

*Then, we have*

$$\mathbb{E}_t[F(x_{t+1})] - F(x_t) \leq -\frac{\varpi \alpha_t K}{2} \|\mathrm{grad} F(x_t)\|^2 + \varpi \alpha_t^2 K Q(K, B_t, \alpha_t, \varpi), \tag{D.13}$$

*where* $Q(K, B_t, \alpha_t, \varpi) = (2K - 1)(K - 1)L_f^2 \delta_1^2 P^2 (J^2 + \alpha_t^2 P^2 H^2)\alpha_t/6 + GP^2 \delta_2^2 + \Upsilon^2 P^2 \delta_4^2 K L_g \varpi + \frac{L_g \delta_3^2 \sigma_L^2 \Upsilon^2 \varpi}{2B_t}$.

*Proof of Lemma D.5.* By the $L_g$-retraction smoothness of $F$, it follows for $t \geq 1$ that

$$F(x_{t+1}) \leq F(x_t) + \langle \mathrm{grad} F(x), \mathrm{R}_{x_t}^{-1}(x_{t+1}) \rangle + \frac{L_g}{2} \|\mathrm{R}_{x_t}^{-1}(x_{t+1})\|^2,$$

where the existence of $\mathrm{R}_{x_t}^{-1}(x_{t+1})$ is guaranteed by Assumption 3.2. Taking expectation on both sides over the randomness over the $t$-th outer iteration yields

$$\mathbb{E}_t[F(x_{t+1})] \leq F(x_t) + \mathbb{E}_t[\langle \mathrm{grad} F(x_t), \mathrm{R}_{x_t}^{-1}(x_{t+1}) \rangle] + \frac{L_g}{2} \mathbb{E}[\|\mathrm{R}_{x_t}^{-1}(x_{t+1})\|^2]. \tag{D.14}$$

Inequality equation D.14 together with Lemmas D.1, D.3, and D.4 give rise to

$$\mathbb{E}_t[F(x_{t+1})] - F(x_t) \leq -\frac{\varpi \alpha_t K}{2} \|\mathrm{grad} F(x_t)\|^2 + L_g \varpi^2 \alpha_t^2 \Upsilon^2 P^2 K^2 \delta_4^2 + \frac{KL_g \sigma_L^2 \Upsilon^2 \delta_3^2 \varpi^2 \alpha_t^2}{2B_t}$$

$$+ \varpi \alpha_t^2 KGP^2\delta_2^2 + \frac{(2K-1)K(K-1)}{6}L_f^2\delta_1^2 P^2(J^2 + \alpha_t^2 P^2 H^2)\varpi\alpha_t^3$$

$$- \frac{\varpi\alpha_t}{2}(1 - KL_g\varpi\alpha_t)\sum_{k=0}^{K-1}\mathbb{E}_t\left[\left\|\left\|\sum_{j=1}^{N}\frac{p_j}{q_t^j N}\mathcal{T}_{\tilde{\eta}_{t,k}^j}(\mathrm{grad}f_j(x_{t,k}^j))\right\|\right\|^2\right]. \tag{D.15}$$

Under Condition (D.12), the third term on the right-hand side of (D.15) can be discarded and then we obtain

$$\mathbb{E}_t[F(x_{t+1})] - F(x_t) \leq -\frac{\varpi\alpha_t K}{2}\|\mathrm{grad}F(x_t)\|^2 + L_g\varpi^2\alpha_t^2\Upsilon^2 P^2 K^2\delta_4^2 + \frac{KL_g\sigma_L^2\Upsilon^2\delta_3^2\varpi^2\alpha_t^2}{2B_t}$$

$$+ \varpi\alpha_t^2 KGP^2\delta_2^2 + \frac{1}{6}(2K-1)K(K-1)L_f^2\delta_1^2 P^2(J^2 + \alpha_t^2 P^2 H^2)\varpi\alpha_t^3$$

$$= -\frac{\varpi\alpha_t K}{2}\|\mathrm{grad}F(x_t)\|^2 + \varpi\alpha_t^2 KQ(K, B_t, \alpha_t, \varpi)$$

where $Q(K, B_t, \alpha_t, \varpi) = (2K - 1)(K - 1)L_f^2\delta_1^2 P^2(J^2 + \alpha_t^2 P^2 H^2)\alpha_t/6 + GP^2\delta_2^2 + \Upsilon^2 P^2\delta_4^2 KL_g\varpi + \frac{L_g\delta_3^2\sigma_L^2\Upsilon^2\varpi}{2B_t}$. $\qquad\square$

Note that $Q(K, B_t, \alpha_t, \varpi)$ in (D.13) consists of four error terms: the first one resulted from the agent drift effect and non-I.I.D. setting, the second one brought by the probability approximating, the third one caused by partial participation, and the fourth one caused by the local stochastic gradient.

## D.2  PROOF OF THEOREM 3.1

Now we are ready to prove Theorem 3.1.

*Theorem 3.2.* The second condition in (3.1) ensure $\{\alpha_t\} \to 0$, and thus, without loss of generality, we may assume that $L_g K\varpi\alpha_t \leq 1$ for all $t \in \mathbb{N}_+$. Then, it follows from D.5 that

$$\alpha_t\|\mathrm{grad}F(x_t)\|^2 \leq \frac{2(F(x_{t+1}) - \mathbb{E}_t[F(x_t)])}{K\varpi} + \alpha_t^2 Q(K, B_t, \alpha_t, \varpi).$$

Summing the inequality above over $t = 1, 2, \ldots, T$ and taking total expectation yields

$$\sum_{t=1}^{T}\alpha_t\mathbb{E}[\|\mathrm{grad}F(x_t)\|^2] \leq \frac{2\mathbb{E}[F(x_0) - F(x_{T+1})]}{K\varpi} + \sum_{t=1}^{T}\alpha_t^2 Q(K, B_t, \alpha_t, \varpi)$$

$$\leq \frac{2(F(x_0) - F(x^*))}{K\varpi} + \sum_{t=1}^{T}\alpha_t^2 Q(K, B_t, \alpha_t, \varpi).$$

Dividing the both side by $A_T = \sum_{t=1}^{T}\alpha_t$ results in the bound for the weighted average norm of the squared gradients as follows

$$\frac{1}{A_T}\sum_{t=1}^{T}\alpha_t\mathbb{E}[\|\mathrm{grad}F(x_t)\|^2] \leq \frac{2(F(x_0) - F(x^*))}{K\varpi A_T} + \frac{1}{A_T}\sum_{t=1}^{T}\alpha_t^2 Q(K, B_t, \alpha_t, \varpi), \tag{D.16}$$

which, under Conditions (3.1), implies that

$$\lim_{T\to\infty}\frac{1}{A_T}\sum_{t=1}^{T}\alpha_t\mathbb{E}[\|\mathrm{grad}F(x_t)\|^2] = 0.$$

The desired result follows the fact above. $\qquad\square$

## D.3 PROOF OF THEOREM 3.2

*Theorem 3.2.* By the definition of $\alpha_t$, there exists a positive constant $M > 0$ such that $\sum_{t=1}^{T} \alpha_t^2, \sum_{t=1}^{T} \alpha_t^3, \sum_{t=1}^{T} \alpha_t^4, \sum_{t=1}^{T} \alpha_t^5 \leq M$ for all $T \geq 1$. Then,

$$\sum_{t=1}^{T} \alpha_t^2 Q(K, B_{\text{low}}, \alpha_t, \varpi) \leq \frac{1}{6}(2K-1)(K-1)L_f^2 \delta_1^2 P^2 (J^2 + P^2 H^2) M$$

$$+ GP^2 \delta_2^2 M + P^2 \delta_4^2 K L_g \varpi M + \frac{L_g \delta_3^2 \sigma_L^2 \Upsilon^2 \varpi}{2B_{\text{low}}} M. \qquad (\text{D.17})$$

On the other hand,

$$A_T = \sum_{t=1}^{T} \frac{\alpha_0}{(\beta+t)^p} \geq \int_{t=1}^{T+1} \frac{\alpha_0}{(\beta+t)^p} \mathrm{d}t = \begin{cases} \alpha_0(\ln(T+1+\beta) - \ln(\beta+1)) & p = 1, \\ \frac{\alpha_0}{1-p}((T+1+\beta)^{1-p} - (b+1)^{1-p}) & p \in (1/2, 1), \end{cases}$$

which gives

$$\frac{1}{A_T} \leq \begin{cases} \frac{1}{\alpha_0(\ln(T+1+\beta) - \ln(\beta+1))} & p = 1, \\ \frac{1-p}{\alpha_0((T+1+\beta)^{1-p} - (b+1)^{1-p})} & p \in (1/2, 1). \end{cases} \qquad (\text{D.18})$$

Plugging (D.17) and (D.18) into (D.16) ensures the desired result. □

In particular, if full agents participate in any round of communication and agents use local full gradient in local updates, implying $G = 0$, $\delta_4^2 = 0$, and $\sigma_L^2 = 0$, then we have

$$\sum_{t=1}^{T} \alpha_t^2 Q(K, B_{\text{low}}, \alpha_t, \varpi) = \frac{1}{6}(2K-1)(K-1)L_f^2 \delta_1^2 P^2 (J^2 \sum_{t=1}^{T} \alpha_t^3 + P^2 H^2 \sum_{t=1}^{T} \alpha_t^4).$$

Hence, we can relax the condition for $\alpha_t$ as $\sum_{t=1}^{\infty} \alpha_t = \infty$ and $\sum_{t=1}^{\infty} \alpha_t^3 < \infty$. If one takes $\alpha_t = \frac{\alpha_0}{(\beta+t)^p}$ with constants $\alpha_0, \beta, p = 1/3 + a$ and $a \in (0, 2/3)$ properly small, it follows that

$$\frac{1}{A_T} \sum_{t=1}^{T} \alpha_t \mathbb{E}_t[\|\mathrm{grad}F(x_t)\|^2] \leq \frac{M(a)}{(\beta+T)^{2/3-a}},$$

where $M(a)$ is a constant depended on $a$. The smaller $a$ the larger $M(a)$.

## D.4 PROOF OF THEOREM 3.3

*Theorem 3.3.* By Lemma D.5 and the RPL condition, we have

$$\mathbb{E}_t[F(x_{t+1})] - F(x^*) + (F(x^*) - F(x_t)) \leq -\mu \varpi K \alpha_t (F(x_t) - F(x^*)) + \varpi \alpha_t^2 K Q(K, B_{\text{low}}, \alpha_t, \varpi).$$

Rearranging this inequality yields

$$\mathbb{E}[F(x_{t+1})] - F(x^*) \leq (1 - \mu \varpi K \alpha_t)(\mathbb{E}[F(x_t)] - F(x^*)) + \varpi \alpha_t^2 K Q(K, B_{\text{low}}, \alpha_1, \varpi), \quad (\text{D.19})$$

where we take the total expectation on both sides. Subsequently, we prove the desired result by induction. For $t = 1$, it follows from the definition of $\nu$. Now assume that (3.2) holds for $t \geq 1$. Then, from (D.19), it follows that

$$\begin{aligned} \mathbb{E}[F(x_{t+1})] - F(x^*) &\leq \left(1 - \frac{\beta \mu \varpi K}{t}\right) \frac{\nu}{t} + \frac{\varpi K \beta^2}{t^2} Q(K, B_{\text{low}}, \alpha_1, \varpi) \\ &= \left(\frac{t - \beta \mu \varpi K}{t^2}\right) \nu + \frac{\varpi K \beta^2}{t^2} Q(K, B_{\text{low}}, \alpha_1, \varpi) \\ &= \left(\frac{t-1}{t^2}\right) \nu - \left(\frac{\beta \mu \varpi K - 1}{t^2}\right) \nu + \frac{\varpi K \beta^2}{t^2} Q(K, B_{\text{low}}, \alpha_1, \varpi) \\ &\leq \frac{\nu}{t+1}, \end{aligned} \qquad (\text{D.20})$$

where $\mathfrak{t} = \gamma + t$, the last inequality is due to $-\left(\frac{\beta\mu\kappa K - 1}{\mathfrak{t}^2}\right)\nu + \frac{\varpi K\beta^2}{\mathfrak{t}^2}Q(K, B_{\text{low}}) \leq 0$ by the definition of $\nu$ and $\mathfrak{t}^2 \geq (\mathfrak{t} - 1)(\mathfrak{t} + 1)$.

On the other hand, for any two points $x, y \in \mathcal{W}$, it follows from the $L_g$-smoothness of $F$ that

$$F(y) \leq F(x) + \left\langle \text{grad}F(x), \text{R}_x^{-1}(y) \right\rangle + \frac{L_g}{2}\|\text{R}_x^{-1}(y)\|^2.$$

Plugging $y = \text{R}_x(-\frac{1}{L_g}\text{grad}F(x))$ into the inequality above yields

$$F(x^*) \leq F(y) \leq F(x) - \frac{1}{2L_g}\|\text{grad}F(x)\|^2,$$

which gives $\frac{1}{2L_g}\|\text{grad}F(x)\| \leq F(x) - F(x^*)$. Replacing $x$ with $x_t$ and plugging the replaced inequality into Inequality (D.20) yields

$$\mathbb{E}[\|\text{grad}F(x_t)\|^2] \leq \frac{2L_g\nu}{\gamma + t},$$

which completes the proof. $\qquad\square$

### D.5 PROOF OF THEOREM 3.4

Here we rewrite Theorem 3.5 as the following more complete statement.

**Theorem D.2.** *Suppose that Assumptions 3.1-3.8 hold. We run Algorithm 1 with a fixed global step size $\varpi$, a fixed batch size $B$, and a fixed number of local updates $K$.*

*1. If the fixed step sizes $\alpha$ and $\varpi$ satisfy $\alpha\varpi K L_g \leq 1$, then*

$$\frac{1}{T}\sum_{t=1}^{T}\mathbb{E}[\|\text{grad}F(x_t)\|^2] \leq \frac{2\Theta(x_1)}{\varpi\alpha KT} + 2\alpha Q(K, B, \alpha, \varpi). \tag{D.21}$$

*2. If local full gradient descent step is performed in local updates, i.e., $\sigma_L = 0$, and one takes a local fixed step size $\alpha > 0$ such that $\alpha = \sqrt{\frac{\Theta(x_1)}{2\varpi P^2(G\delta_2^2 + \Upsilon^2\delta_4^2 K L_g\varpi)KT}}$ with $T$ satisfying*

$$T \geq \max\left\{\frac{\varpi K L_g^2\Theta(x_1)}{2P^2(G\delta_2^2 + K L_g\varpi\Upsilon^2\delta_4^2)}, \frac{\Theta(x_1)(2K-1)^2(K-1)^2 L_f^4\delta_1^4(\varpi^2 L_g^2 J^2 K^2 + P^2 H^2)^2}{72P^2 L_g^4 K^4\varpi^5(G\delta_2^2 + K L_g\varpi\Upsilon^2\delta_4^2)^3}\right\}, \text{ then}$$

$$\frac{1}{T}\sum_{t=1}^{T}\mathbb{E}[\|\text{grad}F(x_t)\|^2] \leq 4P\sqrt{2\Theta(x_1)\left(\frac{G\delta_2^2}{\varpi KT} + \frac{L_g\Upsilon^2\delta_4^2}{T}\right)}.$$

*3. If the true probabilities are known, meaning $G = 0$, and one takes local and global step sizes $\alpha$ and $\varpi$ such that $\alpha\varpi = \sqrt{\frac{\Theta(x_1)B}{(\delta_3^2\sigma_L^2 + 2P^2\delta_4^2 KB)\Upsilon^2 L_g KT}}$ with $T$ satisfying $T \geq$*

$$\max\left\{\frac{K L_g\Theta(x_1)B}{(\delta_3^2\sigma_L^2 + 2P^2\delta_4^2 KB)\Upsilon^2}, \frac{\Theta(x_1)(2K-1)^2(K-1)^2 L_f^4\delta_1^4 P^4(L_g^2\varpi^2 J^2 K^2 + P^2 H^2)^2 B^3}{9(\delta_3^2\sigma_L^2 + 2P^2\delta_4^2 KB)^3\Upsilon^6 L_g^7\varpi^6 K^5}\right\}, \text{ then}$$

$$\frac{1}{T}\sum_{t=1}^{T}\mathbb{E}[\|\text{grad}F(x_t)\|^2] \leq 4\Upsilon\sqrt{L_g\Theta(x_1)\left(\frac{\delta_3^2\sigma_L^2}{KTB} + \frac{2P^2\delta_4^2}{T}\right)}.$$

*Proof.* Item 1. Using $\alpha_t = \alpha$ and $B_t = B$ in Lemma D.5, we have

$$\mathbb{E}[\|\text{grad}F(x_t)\|^2] \leq \frac{2\mathbb{E}[F(x_t) - F(x_{t+1})]}{\varpi\alpha K} + 2\alpha Q(K, B, \alpha_t, \varpi).$$

Summing the inequality above over $t = 1, 2, \ldots, T$ gives rise to

$$\frac{1}{T}\sum_{t=1}^{T}\mathbb{E}[\|\text{grad}F(x_t)\|^2] \leq \frac{2\mathbb{E}[F(x_0) - F(x_{T+1})]}{\varpi\alpha KT} + 2\alpha Q(K, B, \alpha, \varpi)$$

$$\leq \frac{2(F(x_0) - F(x^*))}{\varpi\alpha KT} + 2\alpha Q(K, B, \alpha, \varpi),$$

where the last inequality follows $F(x^*) \le F(x_{T+1})$.

Item 2. In particular, suppose that let $\alpha$ and $\varpi$ satisfy

$$\frac{1}{6}(2K-1)(K-1)L_f^2\delta_1^2 P^2(J^2 + \alpha^2 P^2 H^2)\alpha \le GP^2\delta_2^2 + \Upsilon^2 P^2\delta_4^2 KL_g\varpi. \tag{D.22}$$

Define $h(\alpha) = \frac{2\Theta(x_1)}{\varpi\alpha KT} + 4\alpha GP^2\delta_2^2 + 4\Upsilon^2 P^2\delta_4^2 KL_g\varpi\alpha$. Solving $\alpha^* = \arg\min_{\alpha>0} h(\alpha)$ results in

$$\alpha^* = \sqrt{\frac{\Theta(x_1)}{2\varpi P^2(G\delta_2^2 + \Upsilon^2\delta_4^2 KL_g\varpi)KT}}, \text{ and } h(\alpha^*) = 4P\sqrt{2\Theta(x_1)\left(\frac{G\delta_2^2}{\varpi KT} + \frac{L_g\Upsilon^2\delta_4^2}{T}\right)}.$$

Taking

$$T \ge \max\left\{\frac{\varpi KL_g^2\Theta(x_1)}{2P^2(G\delta_2^2 + KL_g\varpi\Upsilon^2\delta_4^2)}, \frac{\Theta(x_1)(2K-1)^2(K-1)^2 L_f^4\delta_1^4(\varpi^2 L_g^2 J^2 K^2 + P^2 H^2)^2}{72P^2 L_g^4 K^4\varpi^5(G\delta_2^2 + KL_g\varpi\Upsilon^2\delta_4^2)^3}\right\}$$

can ensure that $\alpha^*\varpi KL_g \le 1$ and that (D.22) holds. Hence, the left-hand side of (D.21) is not greater than $h(\alpha^*)$. The proof for Item 3 is similar to that for Item 2. $\qquad\square$

**Remark D.1.** *Continuing with Remark 3.3, if the probabilities $p_i$ are known, i.e., $q_t^i = p_i$, and $p_{\min} = \min_i\{p_i\}$ is not too small and not fairly far away from $p_{\max} = \max_i\{p_i\}$, Item 2 gives the upper bound as $\mathcal{O}(\frac{1}{\sqrt{\varpi KT}}) + \mathcal{O}(\frac{1}{\sqrt{p_{\min}NT}})$. In particular, if $p_i = \frac{S}{N}$ with $S \le N$, the upper bound becomes $\mathcal{O}(\frac{1}{\sqrt{\varpi KT}}) + \mathcal{O}(\frac{1}{\sqrt{ST}})$.*

### D.6  PROOF OF THEOREM 3.5

*Theorem 3.5.* Using a fixed stepsize $\alpha_t = \alpha \le 1/(\mu\varpi K)$ satisfying Condition (D.12) and batchsize $B_{t,k} \in [B_{\text{low}}, B_{\text{up}}]$, it follows from (D.19) that

$$\mathbb{E}[F(x_{t+1})] - F(x^*) \le (1 - \mu\varpi K\alpha)\mathbb{E}[F(x_t)] - F(x^*) + \varpi\alpha^2 KQ(K, S, B_{\text{low}}, \alpha, \varpi),$$

which implies that

$$\mathbb{E}[F(x_T)] - F(x^*) \le (1 - \mu\varpi K\alpha)\mathbb{E}[F(x_{T-1})] - F(x^*) + \varpi\alpha^2 KQ(K, B_{\text{low}}, \alpha, \varpi)$$

$$\le (1 - \mu\varpi K\alpha)^2(\mathbb{E}[F(x_{T-2})] - F(x^*)) + ((1 - \mu\varpi K\alpha) + 1)\varpi\alpha^2 KQ(K, B_{\text{low}}, \varrho, \varpi)$$

$$\cdots$$

$$\le (1 - \mu\varpi K\alpha)^{T-1}(\mathbb{E}[F(x_1)] - F(x^*)) + \sum_{\tau=0}^{T-1}(1 - \mu\varpi K\alpha)^\tau\varpi\alpha^2 KQ(K, B_{\text{low}}, \alpha, \varpi)$$

$$= (1 - \mu\varpi K\alpha)^{T-1}\Theta(x_1) + \frac{1 - (1 - \mu\varpi K\alpha)^T}{\mu\varpi K\alpha}\varpi\alpha^2 KQ(K, B_{\text{low}}, \alpha, \varpi)$$

$$\le (1 - \mu\varpi K\alpha)^{T-1}\Theta(x_1) + \frac{\alpha}{\mu}Q(K, B_{\text{low}}, \alpha, \varpi),$$

which completes the proof. $\qquad\square$

### D.7  PROOF OF THEOREM 3.6

*Theorem 3.6.* Let $\hat{t} = t - 1$. Restricting $q_t^i \in [p_i/2, 3p_i/2]$ yields $\mathbb{P}\{|q_t^i - p_i| \le p_i/2\} \ge 1 - \min\{2e^{-\hat{t}p_i^2/2}, 4(1 - p_i)/(\hat{t}p_i)\}$ by the Hoeffding's and Chebyshev's inequalities. Then

$$\left|\frac{1}{q_t^i} - \frac{1}{p_i}\right| = \left|\frac{q_t^i - p_i}{q_t^i p_i}\right| \le \frac{2}{p_i^2}|q_t^i - p_i|$$

holds with probability not less than $1 - \min\{2e^{-\hat{t}p_i^2/2}, 4(1 - p_i)/(\hat{t}p_i)\}$. Noting that under $q_t^j \in [p_i/2, 3p_i/2]$, $\frac{2}{p_i^2}|q_t^i - p_i| \le \mathcal{G}t^{-a/2}$ (i.e., $|q_t^i - p_i| \le \frac{\mathcal{G}}{2}p_i^2 t^{-a/2}$) implies $|(q_t^i)^{-1} - p_i^{-1}| \le \mathcal{G}t^{-a/2}$, and that

$$\mathbb{P}\left\{|q_t^i - p_i| \le \frac{\mathcal{G}}{2}p_i^2 t^{-a/2}\right\} \ge 1 - \min\left\{2e^{-\frac{\mathcal{G}^2 p_i^4}{2}\hat{t}t^{-a}}, \frac{4(1 - p_i)}{\mathcal{G}^2 p_i^3\hat{t}t^{-a}}\right\},$$

where we use the Hoeffding's and Chebyshev's inequalities again. Let $\mathcal{A} := \{|(q_t^i)^{-1} - p_i^{-1}| \le \mathcal{G}t^{-a/2}\}$, $\mathcal{B} := \{q_t^i \in [p_i/2, 3p_i/2]\}$, and $\mathcal{C} := \{|q_t^i - p_i| \le \frac{\mathcal{G}}{2}p_i^2 t^{-a/2}\}$. The desired result follows $\mathcal{B} \cap \mathcal{C} \subseteq \mathcal{A}$ and $\mathbb{P}\{\mathcal{B} \cap \mathcal{C}\} \ge 1 - \mathbb{P}\{\mathcal{B}^c\} - \mathbb{P}\{\mathcal{C}^c\}$. $\qquad\square$

# E  Supplementary Proofs

## E.1  Proof of Theorem 2.1

**Lemma E.1.** *Let $x_1, x_2, \ldots, x_N$ be independent Bernoulli random variables with $p_i > 0$, i.e., $x_i \sim \text{Bernoulli}(p_i)$. Then,*

$$\mathbb{E}\left[\frac{1}{1 + \sum_{i=1}^N x_i}\right] = \int_0^1 \prod_{i=1}^N (1 - p_i + p_i t) \mathrm{d}t.$$

*Proof.* Let $S = \sum_{i=1}^N x_i$. Considering that for any $a > 0$, it follows $\frac{1}{a} = \int_0^\infty e^{-at} \mathrm{d}t$. Picking $\alpha = 1 + S > 0$ yields

$$\frac{1}{1 + \sum_{i=1}^N x_i} = \frac{1}{1 + S} = \int_0^\infty e^{-t} e^{-St} \mathrm{d}t.$$

Taking expectation for both sides of the equality above, we have

$$\mathbb{E}\left[\frac{1}{1 + \sum_{i=1}^N x_i}\right] = \mathbb{E}\left[\int_0^\infty e^{-t} e^{-St} \mathrm{d}t\right] = \int_0^\infty e^{-t} \mathbb{E}[e^{-St}] \mathrm{d}t,$$

where the second equality is due to that $e^{-St}$ is a discrete random variable. Since $x_i$ is independent and $S = \sum_{i=1}^N x_i$, it follows $\mathbb{E}[e^{-St}] = \prod_{i=1}^N \mathbb{E}[e^{-x_i t}]$. Noting that $\mathbb{E}[e^{-x_i t}] = p_i e^{-t} + (1 - p_i)$, we obtain $\mathbb{E}[e^{-St}] = \prod_{i=1}^N (p_i e^{-t} + (1 - p_i))$. Finally, let $u = e^{-t}$. Then $\mathrm{d}u = -e^{-t} \mathrm{d}t$, $u \to 1$ as $t \to 0$, and $u \to 0$ as $t \to \infty$. Hence,

$$\int_0^\infty e^{-t} \mathbb{E}[e^{-St}] \mathrm{d}t = \int_0^1 \prod_{i=1}^N (1 - p_i + p_i u) \mathrm{d}u,$$

which completes the proof. $\qquad\square$

Now we are ready to prove Theorem 2.1.

*Proof of Theorem 2.1.* At the $t$-th outer iteration, $\mathcal{S}_t$ denotes the indices set of agents who send their gradient streams to the server. Let $x_i = \begin{cases} 1 & i \in \mathcal{S}_t, \\ 0 & i \notin \mathcal{S}_t. \end{cases}$ Then

$$\mathbb{E}\left[\sum_{i \in \mathcal{S}_t} \frac{1}{|\mathcal{S}_t|} \text{grad} f_i(x)\right] = \sum_{i=1}^N \text{grad} f_i(x) \mathbb{E}\left[\frac{x_i}{\sum_{i=1}^N x_i}\right]. \tag{E.1}$$

Noting that $\mathbb{E}\left[\frac{x_i}{\sum_{i=1}^N x_i}\right] = \mathbb{E}\left[\mathbb{E}\left[\frac{x_i}{\sum_{i=1}^N x_i}\right] \Big| x_i\right] = p_i \mathbb{E}\left[\frac{1}{1 + \sum_{j \neq i}^N x_j}\right]$. Since $x_j \sim \text{Bernoulli}(p_j)$ is independent, by Lemma E.1, we have $\mathbb{E}\left[\frac{1}{1 + \sum_{j \neq i}^N x_j}\right] = \int_0^1 \prod_{j \neq i}^N (1 - p_j + p_j t) \mathrm{d}t$. Plugging these intermediate results into (E.1) leads to the desired result. $\qquad\square$

## E.2  Proof of the Claim in Remark 3.4

In general, it is difficult to verify directly whether the objective function satisfies the PL (in the Euclidean setting) or RPL (in the Riemannian setting) property. There are some stronger but useful sufficient conditions that imply PL or RPL condition. Specifically, in the Euclidean setting, a strongly convex function satisfies the PL condition (Bottou et al., 2018). Similarly, in the Riemannian setting, the geodesic strong convexity of real-valued functions implies the RPL property (Boumal, 2023). However, geodesic strong convexity usually requires the use of exponential mapping and its inverse, whose the closed-form expression is not available in some manifolds, e.g., the Stiefel manifold. In the next theorem, we use a more general notion of the strong convexity of real-valued functions—strong retraction-convexity, in the Riemannian setting than geodesic strong convexity and claim that a strongly retraction-convex function also satisfies RPL condition.

**Theorem E.1.** *Suppose that function $q : \mathcal{M} \to \mathbb{R}$ is twice continuously differentiable and $\mu$-strongly retraction-convex with respect to the retraction $\mathrm{R}$ on $\mathcal{W} \subseteq \mathcal{M}$, which is a totally retractive neighborhood of $x^*$, a minimizer of $q$ on $\mathcal{W}$. Then,*

$$q(x) - q(x^*) \le \frac{1}{2\mu}\|\mathrm{grad}q(x)\|^2,$$

*that is, $q$ satisfies the RPL condition on $\mathcal{W}$.*

*Proof.* From the proof of Huang et al. (2015, Lemma 3.2), the $\mu$-strongly retraction-convexity of $q$ implies that

$$q(y) - q(x) \ge \langle \mathrm{grad}q(x), \eta \rangle + \frac{\mu}{2}\|\eta\|^2, \tag{E.2}$$

for any $x \in \mathcal{W}$, $\eta \in \mathrm{T}_x\mathcal{M}$, and $y = \mathrm{R}_x(\eta) \in \mathcal{W}$. Define $q_x(\eta) = q(x) + \langle \mathrm{grad}q(x), \eta \rangle + \frac{\mu}{2}\|\eta\|^2$ with $\eta \in \mathrm{T}_x\mathcal{M}$, which is $\mu$-strongly convex with respect to $\eta$ (in classical), implying that the unique minimizer of $q_x$ is given by $\eta^* = -\frac{1}{\mu}\mathrm{grad}q(x)$. Thus, $\min_{\eta \in \mathrm{T}_x\mathcal{M}} q_x(\eta) = q_x(\eta^*) = q(x) - \frac{1}{2\mu}\|\mathrm{grad}q(x)\|^2$. It follows from (E.2) that

$$q(x^*) \ge q(x) + \langle \mathrm{grad}q(x), \eta \rangle + \frac{\mu}{2}\|\eta\|^2 \ge q_x(\eta^*) = q(x) - \frac{1}{2\mu}\|\mathrm{grad}q(x)\|^2,$$

which completes the proof. $\qquad\square$

