# OpenReview forum: "Riemannian Federated Learning via Averaging Gradient Streams"
_ICLR.cc/2026/Conference — ICLR 2026 Poster_

### Official Review · Reviewer_iW4m · 2025-10-28

**Soundness:** 3
**Presentation:** 2
**Contribution:** 3
**Rating:** 6
**Confidence:** 3

**Summary:**

The authors propose RFedAGS, a federated learning algorithm optimized for optimization on Riemannian manifolds under realistic scenarios of partial client participation and non-IID data. The method introduces a server aggregation strategy called Averaging Gradient Streams (AGS), which averages transported gradient information instead of local model parameters. This approach addresses the non-linearity of manifolds and avoids costly exponential map computations. The authors theoretically demonstrate global and sublinear convergence under decaying step sizes, as well as sublinear/linear convergence to a neighborhood under fixed step sizes. These results are supported by an assumption that holds with high probability as training progresses. Extensive experiments on both synthetic and real-world manifold-structured datasets showcase RFedAGS’s ability to achieve superior accuracy and faster convergence compared to existing Riemannian FL baselines.

**Strengths:**

- The paper introduces an aggregation mechanism, Averaging Gradient Streams (AGS), which avoids direct parameter averaging on curved manifolds by accumulating and transporting gradient updates in a common tangent space. This design elegantly resolves the geometric inconsistency inherent in prior Riemannian FL methods and extends federated optimization to non-Euclidean domains under partial client participation—a setting unexplored before.

- The work demonstrates theoretical depth and rigor, providing global and sublinear convergence guarantees under decaying step sizes and linear convergence under Riemannian PL/strong convexity conditions. The proofs are comprehensive, grounded in standard FL and manifold optimization assumptions, and even include a non-trivial probabilistic justification (Assumption 3.8) for estimating client participation probabilities.

- Despite the technical complexity, the paper is logically structured, moving smoothly from motivation to algorithm design, theory, and experiments. The authors provide sufficient mathematical background, clear notation, and consistent use of terminology. While additional intuition could help non-specialists, the exposition is coherent and reproducible, with algorithmic and experimental details clearly documented.

- The proposed RFedAGS algorithm broadens the applicability of federated learning to manifold-constrained models (e.g., SPD matrices, hyperbolic embeddings, Stiefel subspaces), an area of high emerging interest.

**Weaknesses:**

- While the paper is theoretically rigorous, its dense and mathematically demanding nature may hinder its accessibility to a broader audience at ICLR. Consequently, many critical ideas, such as the geometric intuition behind averaging gradient streams and how vector transport preserves consistency, are primarily presented in formal notation.

- Although the paper claims to handle arbitrary participation and non-IID data, the experiments do not explicitly vary these factors to show robustness. It remains unclear how the algorithm performs under different participation rates or degrees of client heterogeneity.

- The convergence proof relies on Assumption 3.8, which establishes a bound on the estimation error between the true and empirical participation probabilities. While this assumption is theoretically justified, the paper lacks a clear description of how these probabilities are computed in practice and how sensitive the algorithm is to their inaccuracies.

**Questions:**

1. The convergence proofs rely heavily on Assumption 3.8, which bounds the deviation between estimated and true participation probabilities. However, it remains unclear how $q_{i,t}$ is actually computed during training.
- Are these probabilities updated as empirical participation frequencies over rounds, or are they fixed a priori?
- How sensitive is RFedAGS to inaccurate or time-varying participation estimates (e.g., if some clients drop out permanently)?

2. While the theory emphasizes arbitrary participation and heterogeneous data, the experiments do not explicitly test these conditions.
- Could the authors provide additional experiments that vary (a) the proportion of participating clients per round and (b) the degree of data heterogeneity across clients?
- How does RFedAGS perform compared to baselines as participation becomes sparse or data distributions diverge?

3. The proposed AGS framework involves transporting and averaging gradients in the manifold’s tangent space, which may introduce additional computational overhead compared to standard Riemannian FedAvg.
- How does this affect runtime and communication efficiency when the number of clients or model dimensionality scales up?
- Are there specific manifolds (e.g., Stiefel or SPD) where vector transport becomes a bottleneck?

---

> ### Author Response · Authors · 2025-11-20
> **Part I of the response**
>
> Thank you very much for your detailed review and valuable comments. We answer each of your concerns in the following.
>
> **W1. While the paper is theoretically rigorous, its dense and mathematically demanding nature may hinder its accessibility to a broader audience at ICLR. Consequently, many critical ideas, such as the geometric intuition behind averaging gradient streams and how vector transport preserves consistency, are primarily presented in formal notation.**
>
> **A1.** Thank you for your comment. To ensure the rigor and completeness of this paper, we have reviewed the necessary basic knowledge about Riemannian geometry and optimization in Appendix B. Besides, in the main body (see Section 2.1), we show the geometry interpretation of (TM) and our proposed (AGS) to conveniently understand.
>
> **Q1. The convergence proofs rely heavily on Assumption 3.8, which bounds the deviation between estimated and true participation probabilities. However, it remains unclear how $q_{i,t}$ is actually computed during training.**
>
> - **Q1.1. Are these probabilities updated as empirical participation frequencies over rounds, or are they fixed a priori?**
>
> - **A1.1** $q_{i,t}$ is updated as empirical participation frequencies over rounds. In Assumption 2.1, we assume that each agent independently participates in communication with a fixed probability independent of other agent. This enable us to approximate the true probability by frequency as discussed in Section 3.4.
>
> - **Q1.2. How sensitive is RFedAGS to inaccurate or time-varying participation estimates (e.g., if some clients drop out permanently)?**
>
> - **A1.2.** Empirically, we observe that the performance of our RFedAGS using frequencies is very close to that of using true probabilities even early iteration. This is observed from Figures 6 and 7 (in Appendix A.1.1), where the curves **Scheme II-True-0.3** and **Scheme II-True-0.5** exactly use true probabilities while the curves **Scheme II-Freq-0.3** and **Scheme II-Freq-0.5** use frequencies to estimate the probabilities. We observe that the **red** and **yellow** curves (corresponding to using true probabilities and frequencies, respectively) behave similarly. This also highlights the robustness of our method. For time-varying participation probabilities case, it is an interesting and more challenging work. The theoretical convergence of Euclidean FL with time-varying participation probabilities is still not well understood.
>
> **Q2. While the theory emphasizes arbitrary participation and heterogeneous data, the experiments do not explicitly test these conditions.**
>
> - **Q2.1. Could the authors provide additional experiments that vary (a) the proportion of participating clients per round and (b) the degree of data heterogeneity across clients?**
>
> - **A2.1.** We provide a new experiment where we enlarge the level of data heterogeneity and the participation sparsity, respectively. See the table below or Figure 12 in the revision for details.
>
> - **Q2.2. How does RFedAGS perform compared to baselines as participation becomes sparse or data distributions diverge?**
>
> - **A2.2.** It follows from the table below (or, see Figure 12 in the revision) that (i) for all of algorithms, as data distributions diverge, the performance becomes poor; besides, at the same level of data heterogeneity, RFedAGS consistently outperforms compared to the other algorithms. (ii) as participation becomes sparse, the performance of all algorithms becomes poor; on the other hand, at the same participation ratio, our RFedAGS consistently performs compared to other algorithms.
>
> | Algorithms\ heterogeneity or ratio | iid                  | niid-slight          | niid-heavy           | 0.5                  | 0.4                  | 0.3                  |
> |------------------------------------|----------------------|----------------------|----------------------|----------------------|----------------------|----------------------|
> | RFedAGS                            | $4.45\times 10^{-5}$ | $7.92\times 10^{-4}$ | $2.34\times 10^{-3}$ | $7.58\times 10^{-4}$ | $1.00\times 10^{-3}$ | $1.04\times 10^{-3}$ |
> | RFedAvg                            | $4.06\times 10^{-4}$ | $1.20\times 10^{-2}$ | $1.87\times 10^{-2}$ | $1.16\times 10^{-2}$ | $1.13\times 10^{-2}$ | $1.33\times 10^{-2}$ |
> | RFedSVRG                           | $1.17\times 10^{-3}$ | $1.40\times 10^{-2}$ | $1.67\times 10^{-2}$ | $1.49\times 10^{-2}$ | $4.43\times 10^{-2}$ | $1.00$               |
> | RFedProj                           | $1.74\times 10^{-4}$ | $1.88\times 10^{-3}$ | $7.01\times 10^{-3}$ | $1.59\times 10^{-3}$ | $3.17\times 10^{-3}$ | $7.20\times 10^{-3}$ |

---

> > ### Author Response · Authors · 2025-11-20
> > **Part II of the response**
> >
> > **Q3. The proposed AGS framework involves transporting and averaging gradients in the manifold's tangent space, which may introduce additional computational overhead compared to standard Riemannian FedAvg.**
> >
> > - **Q3.1.** How does this affect runtime and communication efficiency when the number of clients or model dimensionality scales up?
> >
> > - **A3.1** We provide table below (which also is added in the revision; see Table 2), which shows the computation and communication complexity when the manifolds are compact Riemannian submanifolds embedded in $\mathbb{R}^{d\times p}$. As shown in the table, the server in RFedAvg requires an additional $\mathbf{ir} \times N$ flops compared to RFedAGS. Meanwhile, RFedAGS requires $K$ additional vector transport evaluations compared to RFedAvg. When $K$ is small and $N$ is large, RFedAvg may require more CUP time than RFedAGS. These discussions are verified by Figure 2. On the other hand, we would like to point out that number of clients, local dataset size, and manifold dimension have a linear relationship with the total computation complexity, so their increase will not cause the total computation complexity to increase sharply.
> >
> > |          | LICpA                                                           | SCC                                 | CC     | TCC                                                                            |
> > |----------|-----------------------------------------------------------------|-------------------------------------|--------|--------------------------------------------------------------------------------|
> > | RFedAGS  | $\mathbf{r}K + \mathbf{v}(K-1) + \mathbf{g}BK+2dpK$             | $\mathbf{r}+dpN$                    | $2dpN$ | $\mathbf{r}(K+1)+\mathbf{v}(K-1)+\mathbf{g}BK + dp(2K+N)$                      |
> > | RFedAvg  | $\mathbf{r}K+\mathbf{g}BK+dpK$                                  | $(\mathbf{ir}+dp)N+\mathbf{r}$      | $2dpN$ | $(\mathbf{ir}+dp)N+\mathbf{r}$                                                 |
> > | RFedSVRG | $\mathbf{r}K + \mathbf{v}K + \mathbf{g}BK + \mathbf{g}S + 3dpK$ | $(\mathbf{ir} + 2dp)N + \mathbf{r}$ | $4dpN$ | $(\mathbf{ir}+2dp)N + \mathbf{r}(K+1) + \mathbf{v}K + \mathbf{g}(BK+S) + 3dpK$ |
> > | RFedProj | $\mathbf{p}(K+2)+\mathbf{g}BK+dp(4K+3)$                         | $\mathbf{p}+ dp(N+2)$               | $2dpN$ | $\mathbf{p}(K+3) + \mathbf{g}BK + dp(4K+N+5)$                                  |
> >
> > Note: LICpA, SCC, CC, and TCC denote the local iteration complexity per agent, server computational complexity, communication complexity, and total computational complexity, respectively. Note that TCC=LICpA+SCC.
> >
> > - **Q3.2. Are there specific manifolds (e.g., Stiefel or SPD) where vector transport becomes a bottleneck?**
> >
> > - **A3.2.** For Riemannian submanifolds embedded in Euclidean spaces, there is a **cheap** vector transport, i.e., vector transport by projection. In these cases, vector transport is not the bottleneck; see Figure 2 for example, where RFedAGS costs less time than others. Obviously, there are some manifolds, whose vector transport is relatively expensive. See Figure 4, for example, where the manifold is the SPD manifold, we use parallel translation as the vector transport, and but ever here as well RFedAGS performed similarly to RFedAvg in time but gave better optimality gap.

---

> > > ### Author Response · Authors · 2025-11-25
> > >
> > > **Dear Reviewer iW4m**
> > >
> > > Once again, we sincerely appreciate the time and effort you have taken to review our work. We have submitted our point-by-point responses and truly value your consideration during the ongoing discussion phase. We are happy to provide any additional details that may assist in your evaluation.
> > >
> > > Best regards

---

### Official Review · Reviewer_mABS · 2025-10-30

**Soundness:** 3
**Presentation:** 3
**Contribution:** 2
**Rating:** 4
**Confidence:** 2

**Summary:**

They propose RFedAGS, a Riemannian Federated Learning algorithm that aggregates gradient streams instead of model parameters to handle manifold curvature and arbitrary partial participation. The method introduces a new aggregation correction AGS-AP, supported by convergence proofs under generalized retraction and bounded vector transport assumptions. Experimental results on several manifold-based tasks  show consistent improvement over prior Riemannian FL baselines.

**Strengths:**

1. The algorithm avoids the computational burden of exponential map inverses and parallel transport used in earlier Riemannian FL methods.

2. It is the first Riemannian FL method proven to work under arbitrary partial participation.

**Weaknesses:**

1. Theoretical clarity and novelty: While the proposed framework claims to generalize existing Riemannian FL methods by relaxing the requirements on retraction and vector transport, the theoretical advancement remains unclear. Specifically, the main difficulty in proving convergence under assumptions like 3.1, 3.2, and 3.5 is not explicitly articulated. The authors should clarify why convergence analysis becomes more challenging under generalized retraction and bounded vector transport, and in what way their proof techniques go beyond those established in prior works. In other words, the paper should highlight which parts of the analysis cannot be handled by the existing Riemannian FL theoretical tools and why this generalization is nontrivial.

2. Significance of AGS-AP extension: The transition from AGS-RS to AGS-AP appears to be a relatively straightforward correction that compensates for non-uniform participation probabilities by reweighting expectations. While this adjustment enables handling arbitrary participation, it is not evident that it introduces fundamentally new theoretical challenges. The proposed fix seems more like an incremental adaptation rather than a substantial methodological contribution. The authors should therefore elaborate on why the treatment of partial participation in the Riemannian context poses unique analytical obstacles that cannot be addressed by simply adapting existing Euclidean analyses with weighted expectations.

In summary, the current formulation does not convincingly demonstrate that the proposed theoretical extensions constitute a significant leap beyond existing works. The authors are encouraged to explicitly contrast their convergence analysis with prior proofs, detailing where prior methods would fail or become inapplicable under their new setting.

**Questions:**

1. Mislabels algorithm in line 438.

2. In A.1.4 it seems like the effect of heterogeneity is almost unseen as the convergence improves consistently when K increases. Is it possible to show results for K > 10? Since the algorithm is not designed to mitigate heterogeneity, there should be a certain level of performance degradation observed with extremely large K.

**Details Of Ethics Concerns:**

No concerns.

---

> ### Author Response · Authors · 2025-11-20
> **Part I of the response**
>
> Thank you very much for your detailed review and valuable comments. We answer each of your concerns in the following.
>
> **W1. Theoretical clarity and novelty: While the proposed framework claims to generalize existing Riemannian FL methods by relaxing the requirements on retraction and vector transport, the theoretical advancement remains unclear. Specifically, the main difficulty in proving convergence under assumptions like 3.1, 3.2, and 3.5 is not explicitly articulated. The authors should clarify why convergence analysis becomes more challenging under generalized retraction and bounded vector transport, and in what way their proof techniques go beyond those established in prior works. In other words, the paper should highlight which parts of the analysis cannot be handled by the existing Riemannian FL theoretical tools and why this generalization is nontrivial.**
>
> **A1.** Existing Riemannian FL algorithms can be divided into two categories: (1) ones based on orthogonal projection [1-2] and (2) ones based (TM) [3-6]. (1) the first class algorithms are restricted on the compact Riemannian submanifolds embedded in Euclidean spaces, however our framework proposed in this paper is designed for general manifolds. Thus the analysis therein is not suitable for our proposed RFedAGS. (2) whether it is the second class algorithms or our algorithm,  the two key steps in analysis are bounding the two terms $\mathbb{E}[\left<\mathrm{grad}F(x _ t),\mathrm{R} _ {x _ t} ^ {-1}(x _ {t+1})\right>]$ and $\mathbb{E}[\Vert\mathrm{R} _ {x _ t}^{-1}(x _ {t+1})\Vert ^ 2]$ (retraction $\mathrm{R}$ is replaced with exponential mapping $\mathrm{Exp}$ in the second calss algorithms).
> For the second class algorithms, it follows form (TM) that $\mathrm{Exp} _ {x _ t} ^ {-1}(x _ {t+1})=\frac{1}{S}\sum _ {j\in \mathcal{S} _ t}\mathrm{Exp} _ {x _ t} ^ {-1}(x _ {t,K} ^ j)$, which still makes these two terms difficult to be bounded when $K>1$ due to the effects of curvature of manifolds.
> Beyond that, to bound the first term mentioned above, a consequent key step is to bound $\mathbb{E}[\Vert\mathrm{R} _ {x _ t} ^ {-1}(x_{t,k}^j)\Vert ^ 2]$ which states the "distance" between the $k$-th local update $x_{t,k}^j$ and the $t$-th outer iterate $x_t$.
> Existing Riemannian FL theoretical tools all do not address these issues for $K>1$.
> Instead, our proposed (AGS) makes use of vector transport to avoid the computation of $\mathrm{Exp}^{-1}$ (even $\mathrm{R} ^ {-1}$) in (TM), maintains linearity with respect to local stochastic gradients (which is consistent with that in the Euclidean setting), and thus enables analysis for $K>1$.
>
>
> **W2. Significance of AGS-AP extension: The transition from AGS-RS to AGS-AP appears to be a relatively straightforward correction that compensates for non-uniform participation probabilities by reweighting expectations. While this adjustment enables handling arbitrary participation, it is not evident that it introduces fundamentally new theoretical challenges. The proposed fix seems more like an incremental adaptation rather than a substantial methodological contribution. The authors should therefore elaborate on why the treatment of partial participation in the Riemannian context poses unique analytical obstacles that cannot be addressed by simply adapting existing Euclidean analyses with weighted expectations.**
>
> **A2.** We would like to emphasize that such a situation where agents' availability and response speeds are hardly predictable is more practical in the FL setting. Existing Riemannian FL algorithms do not have theoretical guarantees in this setting (even in the random sampling setting). Our (AGS-AP) extension enables proposed RFedAGS to be the **first one** in such a setting.
> It is noted that the theoretical challenges of this paper do not lie in introducing this extension, but in analyzing under the algorithm with (AGS), as pointed out in **A1**. Once the analysis challenges (mentioned in **A1**) are overcome, the analysis using the (AGS-AP) extension becomes relatively easy.
>
> [1] Zhang J, Hu J, So A M C, et al. Nonconvex federated learning on compact smooth submanifolds with heterogeneous data[J]. Advances in Neural Information Processing Systems, 2024.
>
> [2] Wang H, Pan Z, He C, et al. Federated Learning on Riemannian Manifolds: A Gradient-Free Projection-Based Approach[J]. arXiv, 2025.
>
> [3] Jiaxiang Li and Shiqian Ma. Federated learning on Riemannian manifolds[J]. Applied Set-Valued Analysis and Optimization, 2023.
>
> [4] Zhenwei Huang, Wen Huang, Pratik Jawanpuria, and Bamdev Mishra. Federated learning on Riemannian manifolds with differential privacy. arXiv, 2024.
>
> [5] He Xiao, Tao Yan, and Kai Wang. Riemannian SVRG using Barzilai-Borwein method as second-order approximation for federated learning. Symmetry, 2024
>
> [6] He Xiao, Tao Yan, and Shimin Zhao. Riemannian SVRG with Barzilai-Borwein scheme for federated learning. Journal of Industrial and Management Optimization, 2025.

---

> > ### Author Response · Authors · 2025-11-20
> > **Part II of the response**
> >
> > **Q1. In A.1.4 it seems like the effect of heterogeneity is almost unseen as the convergence improves consistently when K increases. Is it possible to show results for K > 10? Since the algorithm is not designed to mitigate heterogeneity, there should be a certain level of performance degradation observed with extremely large K.**
> >
> > **A1.** Thank you for your comment. We have performed new experiments with K = 14, 20; see updated Figures 11 and 13, or tables provided below. As stated in Item 1 of Theorem 3.4, at the initial state, the larger $K$ is the faster RFedAGS converges. But for larger $K$ since more noises are introduced, the second term (constant with respect to $T$) at the right-hand side of (3.3) is larger, which may lead to the more inaccuracy of the solutions. From the tables provided below or Figures 11 and 13, we can observe the consistent results with theoretical analysis.
> >
> > Table for **Figure 11**
> > | Figure 11 | IID                   | NIID-slight          | NIID-heavy           | IID                  | NIID-slight          | NIID-heavy           | IID                  | NIID-slight          | NIID-heavy           | IID                  | NIID-slight          | NIID-heavy           |
> > |-----------|-----------------------|----------------------|----------------------|----------------------|----------------------|----------------------|----------------------|----------------------|----------------------|----------------------|----------------------|----------------------|
> > | Iter \ K    | 8                     | 8                    | 8                    | 10                   | 10                   | 10                   | 14                   | 14                   | 14                   | 20                   | 20                   | 20                   |
> > | 0         | 37.99                 | 37.99                | 37.99                | 37.99                | 37.99                | 37.99                | 37.99                | 37.99                | 37.99                | 37.99                | 37.99                | 37.99                |
> > | 50        | 34.48                 | 34.48                | 34.45                | 29.15                | 29.14                | 29.11                | 9.68                 | 9.65                 | 9.45                 | $3.66\times 10^{-1}$ | $3.66\times 10^{-1}$ | $3.55\times 10^{-1}$ |
> > | 100       | 2.82                  | 2.81                 | 2.77                 | $2.80\times 10^{-1}$ | $2.80\times 10^{-1}$ | $2.79\times 10^{-1}$ | $3.00\times 10^{-3}$ | $5.25\times 10^{-3}$ | $8.57\times 10^{-3}$ | $2.02\times 10^{-4}$ | $4.23\times 10^{-3}$ | $1.24\times 10^{-2}$ |
> > | 150       | $2.37\times 10^{-2}$  | $2.50\times 10^{-2}$ | $2.56\times 10^{2}$  | $7.33\times 10^{-4}$ | $1.71\times 10^{-3}$ | $4.45\times 10^{-3}$ | $1.24\times 10^{-4}$ | $2.41\times 10^{-3}$ | $9.19\times 10^{-3}$ | $1.38\times 10^{-4}$ | $3.14\times 10^{-3}$ | $6.66\times 10^{-3}$ |
> > | 200       | $2.56\times 10^{-4}$  | $1.80\times 10^{-3}$ | $2.65\times 10^{-3}$ | $8.96\times 10^{-5}$ | $1.63\times 10^{-3}$ | $4.00\times 10^{-3}$ | $9.99\times 10^{-5}$ | $1.77\times 10^{-3}$ | $4.25\times 10^{-3}$ | $1.40\times 10^{-4}$ | $2.50\times 10^{-3}$ | $7.00\times 10^{-3}$ |
> > | 250       | $5.94 \times 10^{-5}$ | $1.67\times 10^{-3}$ | $3.63\times 10^{-3}$ | $7.09\times 10^{-5}$ | $1.36\times 10^{-3}$ | $3.58\times 10^{-3}$ | $1.11\times 10^{-4}$ | $1.91\times 10^{-3}$ | $4.29\times 10^{-3}$ | $1.21\times 10^{-4}$ | $2.20\times 10^{-3}$ | $4.14\times 10^{-3}$ |
> > | 300       | $6.02\times 10^{-5}$  | $9.55\times 10^{-4}$ | $2.43\times 10^{-3}$ | $8.79\times 10^{-5}$ | $1.84\times 10^{-3}$ | $3.31\times 10^{-3}$ | $1.16\times 10^{-4}$ | $2.88\times 10^{-3}$ | $6.38\times 10^{-3}$ | $1.81\times 10^{-4}$ | $3.70\times 10^{-3}$ | $1.10\times 10^{-2}$ |
> >
> > Table for **Figure 13**
> > | K \ Iter | 0     | 30                    | 60                    | 90                    | 100                   |
> > |----------|-------|-----------------------|-----------------------|-----------------------|-----------------------|
> > | 10       | 70.17 | $4.50\times 10^{-4}$  | $1.30\times 10^{-9}$  | $1.47\times 10^{-13}$ | $3.09\times 10^{-13}$ |
> > | 14       | 70.17 | $7.34\times 10^{-6}$   | $2.53\times 10^{-12}$ | $8.76\times 10^{-13}$ | $4.05\times 10^{-13}$ |
> > | 20       | 70.17 | $1.41\times 10^{-8}$  | $2.40\times 10^{-13}$ | $2.59\times 10^{-13}$ | $3.09\times 10^{-13}$ |
> > | 30       | 70.17 | $7.13\times 10^{-12}$ | $5.44\times 10^{-13}$ | $3.16\times 10^{-13}$ | $2.47\times 10^{-12}$ |

---

> > > ### Author Response · Authors · 2025-11-25
> > >
> > > **Dear Reviewer mABS**
> > >
> > > Thank you again for the time and effort you have put into reviewing our paper. We have submitted our rebuttal and clarifications, and we sincerely appreciate your consideration during the ongoing discussion phase. Please let us know if any further information from our side would be helpful.
> > >
> > > Best regards

---

### Official Review · Reviewer_mU3X · 2025-10-31

**Soundness:** 3
**Presentation:** 2
**Contribution:** 3
**Rating:** 4
**Confidence:** 3

**Summary:**

This paper proposes RFedAGS, a new Riemannian federated learning algorithm that replaces parameter averaging with gradient-stream aggregation to preserve linearity in tangent spaces and improve computational efficiency. RFedAGS proposes an efficient retraction-based aggregation which eliminates the need for the computationally burdensome inverse exponential map or parallel transports required by prior Riemannian FL methods, and theoretically establishes global (and sublinear/linear neighborhood) convergence under various step size regimes. Extensive experimental results are provided across synthetic and real-world tasks, demonstrating RFedAGS’ advantages over competing Riemannian FL baselines under general participation and data heterogeneity.

**Strengths:**

This paper proposes and analyzes RFedAGS, a Riemannian federated learning algorithm that introduces a new and efficient server aggregation scheme based on averaging gradient streams.

The method is designed to effectively handle both partial client participation and data heterogeneity. Theoretical analysis establishes that RFedAGS achieves global convergence and a sublinear convergence rate under decaying step sizes, and further converges sublinearly or linearly to a neighborhood of a stationary point or solution when using fixed step sizes. Extensive experiments on both synthetic and real-world datasets demonstrate the strong empirical performance and stability of the proposed approach.

**Weaknesses:**

1. Limited novelty. The key idea—aggregating gradient flows in tangent space—is conceptually straightforward once the FedAvg update is projected to a manifold setting.
2. The paper lacks a argument for why RFedAGS offers a distinct or superior geometric interpretation.
3. Limited Scope of Baselines: Although several strong Riemannian FL baselines are included (RFedAvg, RFedSVRG, RFedProj) for targeted tasks, more recent algorithms are not considered, e.g., Wang et al., 2025 [1].
4. Some results tied too closely to specific manifolds. Experiments and implementation notes (Appendix A.3) are mostly focused specific manifolds. Broader applicability to more exotic or high-dimensional manifolds remains an open question
5. Although the authors claim computational efficiency (no need for exponential/logarithmic maps), no quantitative results support this.

[1] Wang H, Pan Z, He C, et al. Federated Learning on Riemannian Manifolds: A Gradient-Free Projection-Based Approach[J]. arXiv preprint arXiv:2507.22855, 2025.

**Questions:**

1. Can the authors provide empirical or theoretical discussion regarding the scalability of the method as the number of agents, local dataset size, or manifold dimension increases?
2. Could the method be efficiently applied to other manifolds? Are there limitations?
3. Could the method be compared with recent or advanced Riemannian federated learning algorithms (e.g., Wang et al., 2025 [1]) ?
4. The paper claims computational efficiency due to the removal of exponential/logarithmic maps, yet provides no quantitative analysis.
Could the authors offer detailed communication and computation cost metrics per round, beyond total CPU time, to support this claim?

---

> ### Author Response · Authors · 2025-11-20
> **Part I of the response**
>
> Thank you very much for your insightful comments and questions. Our response to your questions and comments are as follows.
>
> **W1. Limited novelty. The key idea---aggregating gradient flows in tangent space---is conceptually straightforward once the FedAvg update is projected to a manifold setting.**
>
> **A1.** Although our key idea builds up from FedAvg (Euclidean), it is **not a trivial generalization** of the Euclidean counterpart.
>
> The proposed aggregation (AGS) is not simply projecting the FedAvg update, but drawing insight on the nature of the update of FedAvg---averaging all of local stochastic gradients. To achieve this goal, the AGS uses the standard Riemannian optimization tool---vector transport instead of projection.
>
> In fact, there exists different versions of Riemannian generalization of the aggregation. See also our Table 1 in the paper for comparisons. However, we emphasize that **not all of them** can yield nice theoretical results and convincing numerical performance. Our contribution to the best of our knowledge provides the most general (relaxed) take, e.g., partial participation, non-iid data, use of retraction, and use of bounded  vector transport.
>
> **W2. The paper lacks an argument for why RFedAGS offers a distinct or superior geometric interpretation.**
>
> **A2.** Due to the limitations of space, in the last manuscript we did not present the geometric interpretation. We have now added the part in the revision; see Figure 1 (in the revised version). For convenience, we restate that here.
>
> From the perspective of geometry, the (TM) "projects" the final inner iterates $x_{t,K}^j$ back to the tangent space at $x_t$, then averages them and finally retracts the average into the manifold. While, in (AGS), the intermediary negative mini-batch gradients $-\frac{1}{B _ t}\sum _ {b\in \mathcal{B} _ {t,k}^j}\mathrm{grad}f _ i(x _ {t,k} ^ j;\xi _ {t,k,b} ^ j)$ are transported to the tangent space at $x_t$ in some way, then averages them and finally retracts the average into the manifold.
>
> The (TM) actually is an approximation of the weighted averages of inner iterates $x_{t,K}^j$. When the degree of heterogeneity across clients are large, the inner $x_{t,K}^j$ is closer to the minimizer of local function $f_j$, and their average may be far away from the minimizer of the global function. While the proposed (AGS) leverages the gradient information drawn from clients to generate global direction and thus helps to alleviate this bias.
>
>
> **W3. Some results tied too closely to specific manifolds. Experiments and implementation notes (Appendix A.3) are mostly focused specific manifolds. Broader applicability to more exotic or high-dimensional manifolds remains an open question.**
>
> **A3.** We have already conducted experiments on **5 problems**, which are over the **Stiefel** manifold (Section 4, PCA), the **hyperbolic** manifold (Section 4, HSP), the **SPD** manifold (Section 4, FMC), the **sphere** manifold (Appendix A.1), and the **Grassmann** manifold (Appendix A.2). Those five manifolds are commonly-encountered and widely-used in many important applications.

---

> > ### Author Response · Authors · 2025-11-20
> > **Part II of the response**
> >
> > **Q1. Can the authors provide empirical or theoretical discussion regarding the scalability of the method as the number of agents, local dataset size, or manifold dimension increases?**
> >
> > **A1.** Thanks for your comment. We have conducted a new experiment to explore the scalability of the RFedAGS as the number of agents, local dataset size, or manifold dimension increases; see the table provided below or **Figure 16** in Appendix A.3.1 of the revision.
> >
> > In the second row of the table below, we fix the local dataset size and the manifold dimension and enlarge the number of agents. In the fourth row, we enlarge local dataset size and fixed the other two factors. In the last row, we enlarge the manifold dimension and fix the other two factors. In summary, it can be observed from the table that the RFedAGS can all solve these problems of such scale, showing the scalability of RFedAGS.
> > We would like to point out that as shown by Table 2 in the revision, number of agents, local dataset size, and manifold dimension have a linear relationship with the total computation complexity, so their increase will not cause the total computation complexity to increase sharply.
> >
> > | -                 | $(d,r)=(100,5)$                          | $N=60$           | $N=80$           | $N=100$           | $N=150$          | $N=200$           |
> > |-------------------|------------------------------------------|------------------|------------------|-------------------|------------------|-------------------|
> > | $S = 1000$        | rel\_error($\times 10^{-3}$)/CPU time (s) | 2.80/10.30       | 1.81/12.12       | 1.21/12.99        | 1.25/14.06       | 0.58/15.18        |
> > | -                 | $(d,r)=(100,5)$                          | $S=400$          | $S=800$          | $S=1200$          | $S=1600$         | $S=2000$          |
> > | $N=100$           | rel\_error($\times 10^{-3}$)/CPU time (s) | 3.44/6.59        | 4.31/10.79       | 6.57/14.83        | 6.69/18.34       | 5.40/21.03        |
> > | -                 | -                                        | $(d,r)=(1000,5)$ | $(d,r)=(2000,5)$ | $(d,r)=(2000,10)$ | $(d,r)=(4000,5)$ | $(d,r)=(4000,10)$ |
> > | $(N,S)=(50,1000)$ | rel\_error($\times 10^{-2}$)/CPU time (s) | 1.74/13.31       | 1.56/24.14       | 1.27/49.80        | 1.61/41.06       | 1.40/101.83       |
> >
> > Note: relative error is defined as $(F(x _ t)-F ^ *)/F ^ *$ and the CPU time is equal to $\sum _ {t=1} ^ T(S _ t+\max _ iA _ {i,t})$ with $S _ t$ and $A _ {i,t}$ being the time consumed by the server and agent $i$ at the $t$-th round.
> >
> >
> >
> > **Q2. Could the method be efficiently applied to other manifolds? Are there limitations?**
> >
> > **A2.** Appendix discusses experiments on a number of applications/manifolds. The proposed method can efficiently work in general manifolds, not limited to Riemannian submanifolds embedded in Euclidean spaces, e.g., the Grassmann manifold. Our numerical experiments include various manifolds, including sphere manifold, Stiefel manifold, hyperbolic manifold, SPD manifold, and Grassmann manifold.
> >
> > **Q3. Could the method be compared with recent or advanced Riemannian federated learning algorithms (e.g., Wang et al., 2025 [1])?**
> >
> > **A3.** We were not aware of this paper at the submission time (Wang et al. paper was in arXiv in July and the ICLR deadline was in Sep). Thank you for bringing this to attention.
> >
> > In the revision, we have now added the comparison of RFedAGS with the algorithm in Wang et al., 2025 [1] (called ZO-RFedProj); see Figure 2 or the table below.
> > We point out that in our work, the problems we encounter are first-order accessible, while ZO-RFedProj is designed to the situation where the exact first-order information (i.e., gradient) is not available. In the latter case, the authors of [1] proposed an estimator to approximate the gradient and integrate the estimator into the RFedProj [2]. Due to the existence of the estimator error, it can be expected that the performance of ZO-RFedProj is poorer than that of RFedProj.
> > Therefore, as expected, ZO-RFedProj does not perform as good as our algorithm and actually is the worst one compared to other algorithms that uses first-order information.
> >
> > |           |                                           | RFedAGS    | RFedAvg    | RFedSVRG    | RFedProj   | ZO-RFedProj    |
> > |-----------|-------------------------------------------|------------|------------|-------------|------------|----------------|
> > | Synthetic | rel\_error($\times 10^{-3}$)/CPU time (s) | 8.66/0.62  | 74.66/1.90 | 118.29/2.34 | 47.30/0.55 | 248.44/7.07    |
> > | CIFAR10   | rel\_error($\times 10^{-3}$)/CPU time (s) | 0.49/15.28 | 0.87/16.17 | 1.00/34.653 | 0.76/20.21 | 203.94/659.666 |

---

> > > ### Author Response · Authors · 2025-11-20
> > > **Part III of the response**
> > >
> > > **Q4. The paper claims computational efficiency due to the removal of exponential/logarithmic maps, yet provides no quantitative analysis. Could the authors offer detailed communication and computation cost metrics per round, beyond total CPU time, to support this claim?**
> > >
> > > **A4.** Thank you for your suggestion.
> > > We would like to emphasize that the baselines RFedAvg and RFedSVRG require the **exponential** mapping, its inverse, and parallel translation. In the Stiefel manifold, for instance, the exponential mapping involves matrix exponential calculation which is computationally expensive, and the inverse of exponential mapping has not a closed-form expression and only iterative methods are developed to compute it, which makes the computational cost unacceptable. Instead, our RFedAGS requires **retraction and vector transport**. For most of commonly encountered manifolds, the two tools are computationally cheap.
> > >
> > > For the purpose of comparison, we provide a table below (which is added into the revision; see Table 2) to quantitatively demonstrate the computation and communication cost per round taking the compact Riemannian submanifold embedded in $\mathbb{R}^{d\times p}$ as an example, where we use retraction, its inverse, and vector transport to replace exponential mapping, its inverse, and parallel translation.
> > > The communication complexity of RFedAGS, RFedAvg, and RFedProj is the same, and a half of that of RFedSVRG.
> > > In terms of computational complexity, as shown in the table, the servers in RFedAvg and RFedSVRG require an additional $\mathbf{ir} \times N$ flops compared to RFedAGS and RFedProj since $\mathbf{r}\approx \mathbf{p}$ in our experiments. Meanwhile, RFedAGS has approximately the same LICpA as RFedSVRG but requires $K$ additional vector transport evaluations compared to RFedAvg.
> > > Consequently, the CPU time of RFedSVRG is expected to be consistently higher than those of RFedAGS and RFedAvg, regardless of the value of $N$. When $K$ is small and $N$ is large, RFedAvg may require more CPU time than RFedAGS.
> > > Compared to RFedProj, the proposed RFedAGS requires $K$ additional vector transport evaluations in local updates. When a lower-complexity vector transport (e.g., vector transport by projection) is used, RFedAGS may require less CPU time than RFedProj in each outer iteration even if $p$ is not large. This is verified by Figure 2.
> > >
> > > |          | LICpA                                                           | SCC                                 | CC     | TCC                                                                            |
> > > |----------|-----------------------------------------------------------------|-------------------------------------|--------|--------------------------------------------------------------------------------|
> > > | RFedAGS  | $\mathbf{r}K + \mathbf{v}(K-1) + \mathbf{g}BK+2dpK$             | $\mathbf{r}+dpN$                    | $2dpN$ | $\mathbf{r}(K+1)+\mathbf{v}(K-1)+\mathbf{g}BK + dp(2K+N)$                      |
> > > | RFedAvg  | $\mathbf{r}K+\mathbf{g}BK+dpK$                                  | $(\mathbf{ir}+dp)N+\mathbf{r}$      | $2dpN$ | $(\mathbf{ir}+dp)N+\mathbf{r}$                                                 |
> > > | RFedSVRG | $\mathbf{r}K + \mathbf{v}K + \mathbf{g}BK + \mathbf{g}S + 3dpK$ | $(\mathbf{ir} + 2dp)N + \mathbf{r}$ | $4dpN$ | $(\mathbf{ir}+2dp)N + \mathbf{r}(K+1) + \mathbf{v}K + \mathbf{g}(BK+S) + 3dpK$ |
> > > | RFedProj | $\mathbf{p}(K+2)+\mathbf{g}BK+dp(4K+3)$                         | $\mathbf{p}+ dp(N+2)$               | $2dpN$ | $\mathbf{p}(K+3) + \mathbf{g}BK + dp(4K+N+5)$                                  |
> > >
> > > In the table above, LICpA, SCC, CC, and TCC denote the local iteration complexity per agent, server computational complexity, communication complexity, and total computational complexity, respectively. Note that TCC=LICpA+SCC.
> > >
> > > [1] Wang H, Pan Z, He C, et al. Federated Learning on Riemannian Manifolds: A Gradient-Free Projection-Based Approach[J]. arXiv preprint arXiv:2507.22855, 2025.
> > >
> > > [2] Zhang J, Hu J, So A M C, et al. Nonconvex federated learning on compact smooth submanifolds with heterogeneous data[J]. Advances in Neural Information Processing Systems, 2024.

---

> > > > ### Author Response · Authors · 2025-11-25
> > > >
> > > > **Dear Reviewer mU3X**
> > > >
> > > > Once again, we sincerely appreciate the time and effort you have taken to review our work. We have submitted our point-by-point responses and truly value your consideration during the ongoing discussion phase. We are happy to provide any additional details that may assist in your evaluation.
> > > >
> > > > Best regards

---

### Official Review · Reviewer_zBLo · 2025-11-01

**Soundness:** 3
**Presentation:** 3
**Contribution:** 3
**Rating:** 6
**Confidence:** 3

**Summary:**

This paper proposes a new federated optimization algorithm for learning over Riemannian manifolds. The proposed RFedAGS allows efficient and theoretically sound updates even under partial participation and non-IID data. The authors provide convergence guarantees and validate RFedAGS on synthetic and real datasets, showing consistent improvements over baselines.

**Strengths:**

* The proposed aggregation mechanism is novel and easy to follow.
* The paper provides comprehensive convergence analysis.
* The main experiments effectively demonstrate the proposed method’s effectiveness.

**Weaknesses:**

* While I understand the reasonableness of $G$, I am wondering what the value of $G$ would be when the true probabilities are not available to the server in the experiments.
* How are the data partitioned across clients? How many total clients are included in the experiments, and what is the client participation ratio?
* The ablation study is somewhat limited, and the sensitivity of several important parameters is missing—for example, different participation ratios, varying numbers of local steps, and comparisons between using approximate probabilities and true probabilities.
* The assumption of Lipschitz continuity for each $f_i$ seems a bit strong, although it may be necessary for the Riemannian SGD convergence analysis. I am also curious whether this Lipschitz continuity can be empirically verified in the experiments.

**Questions:**

See questions in the weaknesses part.

---

> ### Author Response · Authors · 2025-11-20
>
> Thank you very much for your insightful comments and questions. Our response to your questions and comments are as follows.
>
> **W1. While I understand the reasonableness of $G$, I am wondering what the value of $G$ would be when the true probabilities are not available to the server in the experiments.**
>
> **A1.** The precise value of $G$ is not easily available if the true probabilities are not available. However, we want to emphasize that the constant $G$ is **only used in theory**, but not used as an input of the proposed algorithm. We are more concerned with the existence of this constant than with obtaining its precise value. Theorem 3.6 **guarantees** the existence of $G$ under some reasonable assumptions. In the experiments, we did not need the precise value of $G$ and the proposed algorithm RFedAGS works well in various scenarios.
>
> **W2. How are the data partitioned across clients? How many total clients are included in the experiments, and what is the client participation ratio?**
>
> **A2.** In Appendix A.3, we give the details of the experiment settings. Specifically, for the MNIST and CIFRA10 datasets, we partition the data for clients following the way (Pathological Non-IID) in [1]. Doing so makes the number of each tag different for clients, and thus the local datasets are non-I.I.D across clients.
> For PCA problem, the number of clients is 40 in the synthetic data case, and that is 50 in the MNIST data case.
> For the HSP problem, due to the nature of the dataset of mammals subtree of WordNet itself (total 1180 samples), the number of clients is set as 9. For the FMC problem, the number of clients is set as 50.  For all of the experiments in Section 4, we set the true probability $p_i$ following uniformly distribution $U(0,1)$. Thus, the client participation ratio is $0.5$ in expectation since $\frac{\sum_{i=1}^N\mathbb{E}[p_i]}{N}=\frac{1}{2}$.
>
> **W3. The ablation study is somewhat limited, and the sensitivity of several important parameters is missing---for example, different participation ratios, varying numbers of local steps, and comparisons between using approximate probabilities and true probabilities.**
>
> **A3.** In Appendix A, we already show a number of ablation studies. Below, we reproduce those again.
>
> - **different participation ratio:** In Appendix A1.2, we conduct experiments to test if frequencies approximating probabilities is workable, and the impact of different participation ratio. Here we let each agent has the same true probability $p_i=\rho$. This case indeed reduces to the random sampling case, and it is expected that the performance is consistent with that of random sampling. The results are indeed so.
>
> - **varying numbers of local steps:** In Appendix A1.4, we test the impact of different number of local updates $K$ on the performance of RFedAGS. The results show that more $K$ leads to faster convergence at the initial stage and introduces more  noise to the final solutions, which is consistent with the theoretical finding (Theorem 3.4).
>
> - **comparisons between using approximate probabilities and true probabilities:** In Appendix A1.1, the results shows the performance of using frequencies is very close to that of using true probabilities.
>
> **W4. The assumption of Lipschitz continuity for each $f_i$ seems a bit strong, although it may be necessary for the Riemannian SGD convergence analysis. I am also curious whether this Lipschitz continuity can be empirically verified in the experiments.**
>
> **A4.** The assumption of Lipschitz continuity is a **standard requirement** for Euclidean/Riemannian optimization [2-4]. The commonly encountered problems are smooth and thus satisfy the assumption when restricted on a compact subset, which naturally hold for compact manifolds such as the Stiefel manifold. Therefore, in experiments, if the generated iterates stay in a compact subset of the manifold and the objective function is smooth, then we can empirically claim that the function is Lipschitz continuity in the compact subset. Moreover, in [3], an upper bound of Lipschitz constant is obtained for some problems under reasonable assumptions, e.g., principal eigenvector computation over sphere manifold, Frechet mean computation over SPD manifold, Wasserstein barycenter over SPD manifold, and hyperbolic structured prediction.
>
> [1] McMahan B, Moore E, Ramage D, et al. Communication-efficient learning of deep networks from decentralized data. Artificial intelligence and statistics. PMLR, 2017.
>
> [2] Jianyu Wang and Gauri Joshi. Cooperative SGD: A unified framework for the design and analysis of local-update SGD algorithms. Journal of Machine Learning Research, 2021.
>
> [3] Han, A., Mishra, B., Jawanpuria, P. et al. Differentially private Riemannian optimization. Mach Learn 113, 1133-1161 (2024).
>
> [4] Hosseini, R., Sra, S. An alternative to EM for Gaussian mixture models: batch and stochastic Riemannian optimization. Math. Program, 2020.

---

> > ### Author Response · Authors · 2025-11-25
> >
> > **Dear Reviewer zBLo**
> >
> > Thank you again for the time and effort you have put into reviewing our paper. We have submitted our rebuttal and clarifications, and we sincerely appreciate your consideration during the ongoing discussion phase. Please let us know if any further information from our side would be helpful.
> >
> > Best regards

---

### Meta-Review · Area_Chair_XxLT · 2026-01-07

**Summary:**

The paper studies Riemannian Federated Learning (FL) and proposes a new algorithm with convergence analysis under partial participation and data heterogeneity. It provides a meaningful advancement to the field of Riemannian FL.

**Reviewer Concerns:**

The reviewers had concerns on the novelty, the need for some additional results from experiments, and other technical details. The authors provided a substantial rebuttal addressing these concerns. I believe that all the concerns have been addressed satisfactorily.

**Reviewer Scores:**

I think all the reviewers would have a positive opinion after the rebuttal. They would give scores of 6 or higher.

---

### Decision · Program_Chairs · 2026-01-26

Accept (Poster)